# *SYNERGISTIC ON AUXIN AND CYTOKININ 1* positively regulates growth and attenuates soil pathogen resistance

Andrej Hurný[1], Candela Cuesta [1,2,15], Nicola Cavallari[1,15], Krisztina Ötvös[1,3], Jerome Duclercq [4], Ladislav Dokládal[5,6], Juan Carlos Montesinos[1], Marçal Gallemí[1], Hana Semerádová[1], Thomas Rauter[1,7], Irene Stenzel[8], Geert Persiau[9,10], Freia Benade[11], Rishikesh Bhalearo[12], Eva Sýkorová [5], András Gorzsás [13], Julien Sechet [14], Gregory Mouille[14], Ingo Heilmann [8], Geert De Jaeger[9,10], Jutta Ludwig-Müller [11] & Eva Benková[1✉]

Plants as non-mobile organisms constantly integrate varying environmental signals to flexibly adapt their growth and development. Local fluctuations in water and nutrient availability, sudden changes in temperature or other abiotic and biotic stresses can trigger changes in the growth of plant organs. Multiple mutually interconnected hormonal signaling cascades act as essential endogenous translators of these exogenous signals in the adaptive responses of plants. Although the molecular backbones of hormone transduction pathways have been identified, the mechanisms underlying their interactions are largely unknown. Here, using genome wide transcriptome profiling we identify an auxin and cytokinin cross-talk component; *SYNERGISTIC ON AUXIN AND CYTOKININ 1* (*SYAC1*), whose expression in roots is strictly dependent on both of these hormonal pathways. We show that SYAC1 is a regulator of secretory pathway, whose enhanced activity interferes with deposition of cell wall components and can fine-tune organ growth and sensitivity to soil pathogens.

[1] Institute of Science and Technology, Klosterneuburg, Austria. [2] Departamento de Biología de Organismos y Sistemas, Universidad de Oviedo, Oviedo, Spain. [3] Bioresources Unit, Center for Health & Bioresources, AIT Austrian Institute of Technology, Tulln, Austria. [4] Unité 'Ecologie et Dynamique des Systèmes Anthropisés' (EDYSAN UMR CNRS 7058 CNRS), Université du Picardie Jules Verne, UFR des Sciences, Amiens, France. [5] Institute of Biophysics, The Czech Academy of Sciences, Královopolská 135, 61265 Brno, Czech Republic. [6] Mendel Centre for Plant Genomics and Proteomics, CEITEC, Masaryk University, Brno, Czech Republic. [7] Molecular Biology and Biochemistry, Gottfried Schatz Research Center, Medical University of Graz, Neue Stiftingtalstraße 6/6, 8010 Graz, Austria. [8] Department of Cellular Biochemistry, Institute for Biochemistry and Biotechnology, Martin-Luther-University Halle-Wittenberg, Halle, Germany. [9] Department of Plant Biotechnology and Bioinformatics, Ghent University, Ghent, Belgium. [10] VIB Center for Plant Systems Biology, Ghent, Belgium. [11] Institut für Botanik, Technische Universität Dresden, Dresden, Germany. [12] Umeå Plant Science Centre, Department of Forest Genetics and Plant Physiology, Swedish University of Agricultural Sciences, S-901 83 Umeå, Sweden. [13] Department of Chemistry, Umeå University, Linnaeus väg 6, SE-901 87 Umeå, Sweden. [14] Institut Jean-Pierre Bourgin, INRAE, AgroParisTech, Université Paris-Saclay, 78000 Versailles, France. [15] These authors contributed equally: Candela Cuesta, Nicola Cavallari. ✉email: eva.benkova@ist.ac.at

Plants are sessile organisms, so throughout evolution, this lack of mobility has been compensated for by a unique survival strategy—an exceptional developmental plasticity of the plant body. Plants are able to rapidly modulate their growth and whole architecture in order to efficiently use local resources and adapt to fluctuating environmental conditions. Plant hormones are essential mediators and endogenous transducers of these environmental inputs. Hormonal pathways connected via multiple levels of interactions form powerful regulatory networks that sensitively react to changes in the environment and drive the relevant adaptive responses. Currently, a hormonal network formed by the two developmentally essential hormones, auxin and cytokinin, is among the best characterized endogenous regulatory systems[1,2]. Antagonistic inputs of auxin and cytokinin balance proliferation and differentiation of meristematic cells to maintain the root and shoot apical meristem activity[3,4], as well as defining the branching pattern of both roots and shoots[5,6]. Conversely, cell division and growth in plant tissue cultures[7], or rapid elongation growth of roots, is under the synergistic control of cytokinin and auxin[8]. Recently, core pathways mediating hormone perception and signal transduction have been uncovered[9–13] and a complex regulatory network interconnecting these two pathways has been identified. Auxin positively feedbacks on cytokinin biosynthesis through the direct transcriptional control of *ISOPENTENYL TRANSFERASE* (*IPT*) genes, mediated via the AUXIN RESPONSE FACTOR 19 (ARF19)[8]. The auxin pathway can directly stimulate the cytokinin signaling pathway through ARF5-mediated direct transcriptional control of *CYTOKININ RESPONSE FACTOR 2* (*CRF2*)[14]. Cytokinin coordinates the distribution of auxin by regulating the expression of influx and efflux carriers of *Aux/LAX*, and the *PIN* family; both transcriptionally and posttranslationally[3,15–18]. Although these findings have uncovered just a part of this multilevel hormonal network, the complexity of the mechanisms underlying the coordination of plant development is obvious. Such a hormonal network is a guarantor of plant developmental plasticity and adaptability in response to environmental inputs[19]. For example, modulation of organ growth kinetics is one of the most efficient and powerful mechanisms plants employ to rapidly react to environmental changes; such as water and nutrient availability, biotic, and abiotic stresses[20–22].

Although the contribution of auxin and cytokinin to the regulation of organ growth is well established[1,23], the molecular mechanisms integrating the inputs of both pathways, or the downstream components, are still largely unknown. Here, we identified a previously undescribed hub of auxin–cytokinin crosstalk. We show that auxin and cytokinin converge at the regulation of *SYNERGISTIC ON AUXIN AND CYTOKININ 1* (*SYAC1*), encoding for a protein of unknown function. SYAC1 is a component of the secretory pathway and when overexpressed can impact on the cell wall composition. Modulation of SYAC1 activity affects growth of plant organs such as roots, hypocotyls, and interferes with apical hook development. Growth of plant organs is tightly linked with defense mechanisms protecting plants against pathogens[24,25], with hormones playing an essential function in coordination of these regulatory pathways[26]. Noteworthy, we found that in addition to the control of root growth, SYAC1 impacts root sensitivity to soil pathogens such as *Plasmodiophora brassicae*.

## Results

### Auxin and cytokinin control expression of *SYAC1* in root.
To search for molecular components and mechanisms of auxin–cytokinin crosstalk, we performed genome wide transcriptome profiling in *Arabidopsis thaliana* roots after hormonal treatment.

The transcriptome analysis was performed on 5-day-old seedlings exposed to auxin (1 μM 1-naphthaleneacetic acid; NAA), cytokinin (10 μM N6-benzyladenine), and both hormones simultaneously for 3 h. As the original focus of the project was on genes involved in root branching, the transcriptome profiling was performed on pericycle tissue after sorting cells expressing a green fluorescent protein (GFP) reporter in J1201 reporter lines. *SYNERGISTIC ON AUXIN AND CYTOKININ 1* (*SYAC1*, *AT1G15600*), which encodes a protein of unknown function, was among the top candidate genes selected for their expression being synergistically upregulated by simultaneous hormonal treatment when compared with the expected additive effect of both hormones applied separately. After a 3 h of treatment with either auxin or cytokinin increased *SYAC1* expression (2.47- and 1.53-fold, respectively, $n = 3$ each) was detected, whereas application of both hormones simultaneously resulted in 16.36-fold ($n = 3$) higher expression compared with the untreated control (Supplementary Fig. 1a). The *SYAC1* expression profile in roots was further validated by quantitative real-time (RT-qPCR) (Fig. 1a). Further to this, we found a significant increase of *SYAC1* transcription only 30 min after application of both hormones when compared with untreated roots (Fig. 1b), thus indicated that *SYAC1* is among the early response genes rapidly induced by auxin and cytokinin. Lack of either cytokinin or auxin perception mediated through CRE1-12/AHK4, AHK3 and TIR1, AFB2 receptors, respectively, severely attenuated transcription of *SYAC1* in response to dual auxin and cytokinin treatment (Supplementary Fig. 1b); suggesting that both cytokinin and auxin signaling cascades contribute to synergistic regulation of *SYAC1* transcription.

To examine the spatio-temporal pattern of *SYAC1* expression and its responsiveness to hormones in roots, the *SYAC1* promoter was cloned with either β-*Glucuronidase* (*GUS*), nuclear localized *GFP* reporters, or *SYAC1* genomic coding sequence fused to *GFP*. The basal expression of *pSYAC1:GUS*, *pSYAC1:nlsGFP*, and *pSYAC1:gSYAC1-GFP* reporters was under the threshold of detection, however, exposure to cytokinin for 6 h enhanced reporter signal in the quiescent center (QC) and columella initials (CI) (Fig. 1c; Supplementary Fig. 1c–e). When treated with auxin the activity of *pSYAC1* in the QC and provasculature of the root apical meristem could be detected (Fig. 1c; Supplementary Fig. 1c). The promoter activity of *pSYAC1* was substantially enhanced by the simultaneous application of both hormones, and remarkably, a strong reporter signal was detected in the differentiation and rapid elongation zone, a pattern not observed in roots exposed to either of the hormones separately (Fig. 1c and Supplementary Fig. 1c–e). As the initial concentrations of auxin (1 μM) and cytokinin (10 μM) used for transcriptome profiling within 3 h were relatively high, we tested the sensitivity of the *pSYAC1* reporter line to varying concentrations of both hormones. We confirmed that *pSYAC1* sensitively responded to simultaneous application of both hormones, and that their application at concentrations of 0.25 μM each is sufficient to trigger reporter expression in the root differentiation and elongation zones (Supplementary Fig. 1f). These results demonstrate that the *SYAC1* promoter is under the tight control of the combined and synergistic action of auxin and cytokinin. When each hormone was applied individually, *SYAC1* expression was activated in cells known to exhibit either auxin or cytokinin response maxima, such as QC/CI[27] or cells of the provasculature[28], respectively. Intriguingly, *SYAC1* transcription in the root differentiation and elongation zone was fully dependent on the simultaneous enhancement of both auxin and cytokinin.

As application of cytokinin and auxin might lead to the dysregulation of other hormonal pathways, in particular that of ethylene[29], we also examined the sensitivity of *SYAC1* to this

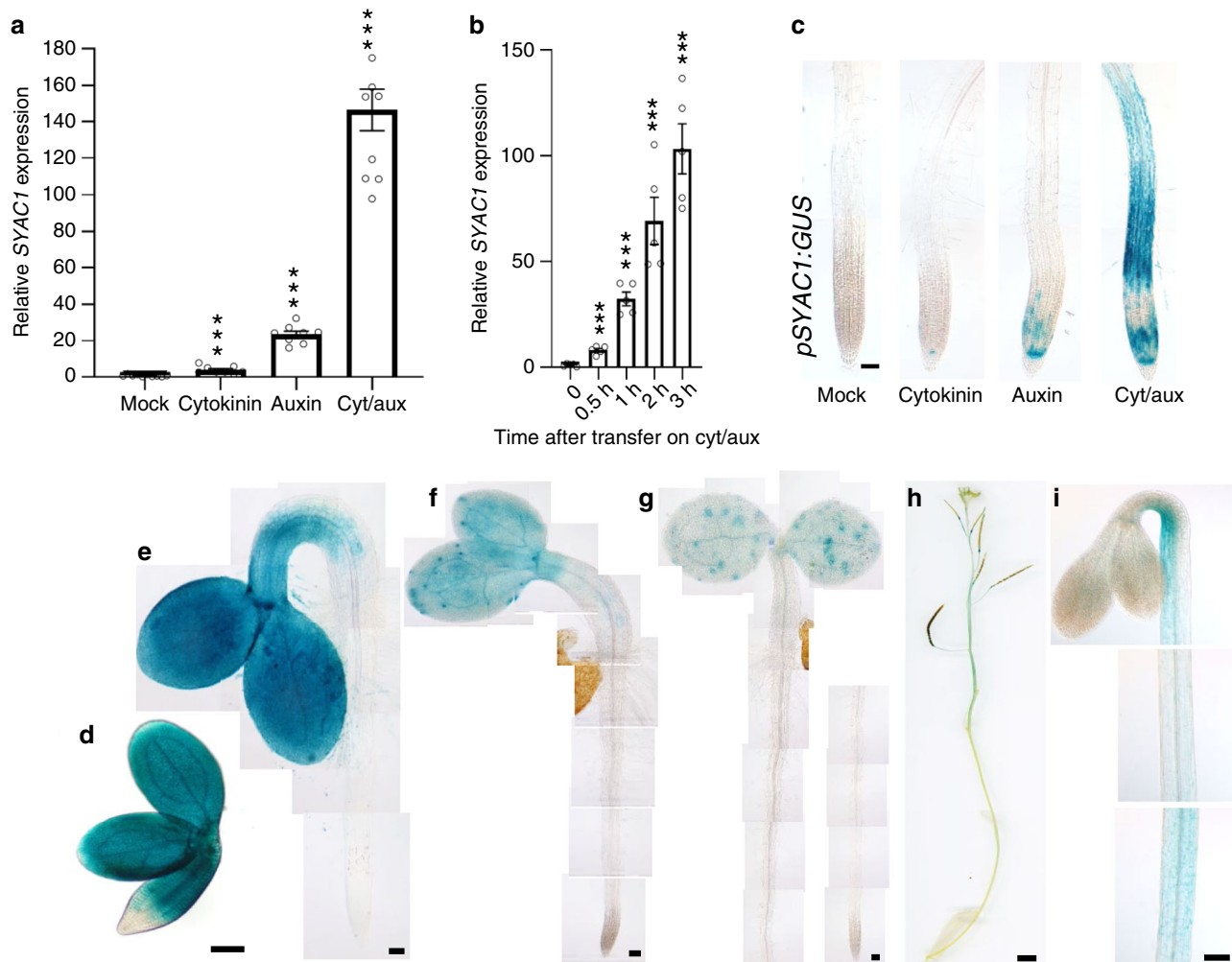

**Fig. 1 SYAC1 expression in *Arabidopsis* and in response to hormonal treatments. a, b** Expression of *SYAC1* in 5-day-old *Arabidopsis* roots analyzed by RT-qPCR. Seedlings were treated with cytokinin (10 µM) and auxin (1 µM) and both hormones together for 3 h (**a**) or both hormones together for indicated time intervals (**b**). Significant differences to mock treated roots are indicated as ***$P < 0.001$ (*t*-test, $n = 5$–8 biological replicates with three technical replicates each, average ± SE). **c–i** *SYAC1* expression monitored using *pSYAC1:GUS* reporter. Roots treated with cytokinin (10 µM) and auxin (1 µM) and both hormones together for 6 h (**c**), and untreated mature embryo (**d**), 2-, 3- and 4-day-old seedling (**e–g**); 8-week-old shoot (**h**), and dark grown hypocotyl and apical hook of 3-day-old seedling (**i**). Scale bar 50 µm (**c**, **e–g**), 200 µm (**d**), 500 µm (**h**), and 100 µm (**i**). 1-Naphthaleneacetic acid and N6-benzyladenine used as auxin and cytokinin, respectively.

hormone. No enhancement of *pSYAC1:GUS* expression was detected in roots treated with either 1-aminocyclopropane-1-carboxylic acid (ACC, a precursor of ethylene biosynthesis) only or in combination with either cytokinin, auxin, or both hormones together (Supplementary Fig. 1g). Likewise, treatment with other hormones such as abscisic acid (ABA), methyl jasmonate (MeJA), brassinosteroids (BR), gibberellins (GA₃), or inhibitor of GA biosynthesis paclobutrazol (PAC) did not trigger or interfere with *SYAC1* expression (Supplementary Fig. 1h). Taken together, the expression analysis confirms *SYAC1* as a common target of the auxin and cytokinin pathways acting in roots.

**Spatio-temporal pattern of SYAC1 expression in planta**. To explore the growth and developmental processes in which *SYAC1* might be involved, we monitored its expression at different stages of *Arabidopsis thaliana* development. Strong *SYAC1* promoter activity was detected in the embryonic hypocotyl and cotyledons, but not in the embryonic root (Fig. 1d). During germination, in 2-day-old seedlings, the *pSYAC1:GUS* activity remained strong in cotyledons and the upper part of the hypocotyl, however, in the

lower hypocotyl its expression ceased almost completely with commencing rapid elongation growth (Fig. 1e). As growth of hypocotyl and cotyledons progressed, in 3- to 4-day-old seedlings, *SYAC1* promoter activity in these organs gradually attenuated (Fig. 1f, g) and in ~3-week-old seedlings expression was under detection limit (Supplementary Fig. 1i). In adult 8-week-old *Arabidopsis* plants, *pSYAC1:GUS* signal was observed in the upper part of the stem and the abscission zone of siliques (Fig. 1h; Supplementary Fig. 1j). In etiolated seedlings, the *SYAC1* promoter activity was concentrated in short cells at the inner (concave) side of the apical hook, whereas no signal was detected in expanded cells at outer side of the hook (Fig. 1i). Based on these data, SYAC1 function seems not to be limited to plant roots and its expression pattern spatio-temporally largely correlates with processes involving the control of elongation growth.

**SYAC1 impacts on elongation growth of plant organs**. To gain insights into the developmental function of *SYAC1*, we performed a detailed phenotypic analysis of plants with either a reduced or an enhanced activity of this gene. In the available mutant alleles,

the T-DNA is inserted either in the middle of the 3′ untranslated region (*syac1-1*, *syac1-2*, and *syac1-3*) or in the middle of the second intron (*syac1-4*) (Supplementary Fig. 2a). As in the *syac1-3* allele expression of the gene is not fully suppressed (Supplementary Fig. 2b), we obtained an additional *syac1-5* mutant line using the CRISPR/Cas9 approach. In the *syac1-5* plant, the CRISPR/Cas9 cassette introduces an extra thymine at 90 bps after the ATG, which results in a STOP codon after 33 amino acids in the SYAC1 coding sequence (Supplementary Fig. 2a). In addition, to investigate the impact of increased *SYAC1* expression on plant development, the transgenic lines *SYAC1-HAox*, *HA-SYAC1ox*, *SYAC1-GFPox*, and *GFP-SYAC1ox* carrying *SYAC1* fused to either the *-HA tag* or a *GFP* reporter under the control of the 35S promoter were generated and enhanced expression was confirmed using RT-qPCR and western blot approaches (Supplementary Fig. 2c, d).

Given the observed pattern of *SYAC1* expression, we focused on growth processes involving the tightly controlled cell expansion, such as apical hook development, hypocotyl elongation, and primary root growth. Specific expression at the concave side of the apical hook prompted us to more closely investigate *SYAC1* function in this developmental process. In control *Arabidopsis* seedlings, shortly after germination (about 15–20 h), the hypocotyl progressively bent to establish an apical hook with an angle around 180° (formation phase, F). This bend was stabilized during the maintenance phase (M) and subsequently, about 60 h after germination, a progressive opening of the hook occurred (opening phase, O) (Fig. 2a)[30,31]. The overexpression of *SYAC1* prevented the formation of the apical hook bend, severely interfering with apical hook development. In contrast, in *syac1-3* and *syac1-5*, the formation phase occurred at a similar rate to the wild-type controls, but the maintenance phase was shortened and the opening of the hook started earlier at 35 h after germination. Introduction of *pSYAC1:gSYAC1-GFP* into the *syac1-3* background rescued this defect and prolonged the maintenance phase until 60 h after germination, as observed in wild-type seedlings (Fig. 2a). Apical hook development is the result of tightly orchestrated differential growth along the apical–basal axis of the hypocotyl. Since the *SYAC1* expression maximum occurs in the shorter, concave side of the apical hook curvature (Fig. 1i), these data suggest that local accumulation of SYAC1 restricts expansion of cells locally at the inner side of hook and thereby coordinates the timely transition of the closed apical hook to the opening phase. Disruption of this endogenous expression pattern in *SYAC1ox* leads to inhibition of cell expansion on both sides of the hypocotyl, which prevents the formation of the apical hook. Hence, SYAC1 might play an important role in the regulation of differential growth, possibly by fine tuning cell elongation. Consistent with this notion, modulation of *SYAC1* activity affected growth of hypocotyls. In 4-day-old dark-grown etiolated seedlings hypocotyls were significantly longer in both *syac1-3* and *syac1-5* alleles, whereas *SYAC1* overexpression resulted in severe reduction of hypocotyl length when compared with the wild-type control (Fig. 2b). Since hypocotyl growth in darkness is largely driven by cell elongation rather than cell proliferation[32], the hypocotyl growth defects observed in *syac1* mutants and *SYAC1ox* further support the SYAC1 function in regulation of cell elongation.

Analysis of root growth did not reveal any significant alterations in *syac1-3* and *syac1-5* compared with the wild type when grown on either control or hormone supplemented media (Fig. 2c, d). We therefore tested whether SYAC1 might operate in root growth adaptation to transient hormonal fluctuations. Five-day-old *syac1-3* and *syac1-5* seedlings revealed significantly reduced sensitivity to transient increases of auxin and cytokinin when compared with wild type (Fig. 2e). Hence, we hypothesized

that under constitutive hormonal treatment conditions other proteins might compensate for the absence of SYAC1. An in silico search for *SYAC1* related genes in the *Arabidopsis* genome identified a family of eight similar (40–60%) homologous genes of which seven are located as a cluster on chromosome 1 (Supplementary Fig. 2e). Among these, we found that *BROTHER OF SYAC1* (*BSYAC1*), a close homolog of *SYAC1*, is also synergistically regulated by auxin and cytokinin (Supplementary Fig. 2f), and thus presumably partially compensates for the loss of *syac1* activity. By contrast, the overexpression of *SYAC1* significantly reduced root and cell length when compared with wild type (Fig. 2c; Supplementary Fig. 2g). Monitoring root growth revealed that estradiol-induced expression of *SYAC1* triggered a steep deceleration in root growth (Fig. 2f), indicating that SYAC1 effectively feeds back onto the kinetics of root elongation. Taken together, these results suggest that SYAC1 might act as a developmentally specific regulator of elongation growth, whose activity is involved in coordination of specific phases of apical hook development as well as the growth of other organs, such as hypocotyls and roots.

**SYAC1 localizes to secretory pathway compartments**. To explore cellular function of SYAC1, we next compared its subcellular localization in *Arabidopsis* root cells with specific reporters for cellular compartments. In the estradiol inducible line, 6 h after induction SYAC1-GFP is restricted to small compartments in the cell interior. Measurement of Pearson correlation coefficient revealed a high SYAC1 colocalization pattern with Golgi and *trans*-Golgi (TGN) compartments labeled by the anti-SEC21 (0.57 ± 0.01; $n = 24$) and anti-ECH (0.51 ± 0.02; $n = 36$) antibody, respectively. This subcellular localization was further confirmed by colocalization with anti-ARF1 (0.45 ± 0.02; $n = 34$) and anti-SYP61 (0.49 ± 0.02; $n = 34$) antibodies, which label both Golgi and TGN. A significant colocalization was also observed with the pre-vacuolar/endosomal compartments (PVC), labeled with a mixture of anti-ARA7 and anti-RHA1 (0.52 ± 0.02; $n = 33$), or anti-VSR (0.34 ± 0.03; $n = 13$) antibodies. In contrast, almost no colocalization was observed between SYAC1 and anti-BIP2 (0.04 ± 0.04; $n = 12$) and anti-PIN2 (0.03 ± 0.04; $n = 21$) antibodies, which label the ER and the plasma membrane, respectively (Fig. 3a, b). Accordingly, the SYAC1-GFP signal in *SYAC1-GFPox* line exhibited strong colocalization with markers for Golgi (anti-SEC21; 0.55 ± 0.02; $n = 34$), TGN (anti-ECH; 0.60 ± 0.02; $n = 41$), both of them together (anti-ARF1; 0.55 ± 0.02; $n = 41$ and anti-SYP61; 0.40 ± 0.02; $n = 40$) and PVC (anti-ARA7/anti-RHA1; 0.44 ± 0.02; $n = 39$; anti-VSR; 0.47 ± 0.02; $n = 31$) but almost no colocalization with markers for ER (anti-BIP2; 0.01 ± 0.03; $n = 22$) and the plasma membrane (anti-PIN2; 0.02 ± 0.02; $n = 35$) (Supplementary Fig. 3a, b). To further validate the immunocolocalization results, we crossed the *GFP-SYAC1ox* line with the multicolor "Wave" marker set[33] for analysis of plant endomembrane compartments. We confirmed colocalization of SYAC1 with markers for Golgi (wave 18R; 0.53 ± 0.03; $n = 16$ and wave 127R; 0.42 ± 0.02; $n = 15$), Golgi and endosomes (wave 25R; 0.69 ± 0.03; $n = 20$, and wave 29R; 0.35 ± 0.03; $n = 12$), Golgi and TGN (*SYP61:SYP61-CFP*; 0.45 ± 0.02; $n = 28$), TGN and early endosomes (wave 13R; 0.27 ± 0.06; $n = 6$) as well as for endosomes/recycling endosomes (wave 34R; 0.31 ± 0.05; $n = 9$ and wave 129R; 0.33 ± 0.02; $n = 18$). In agreement with immunocolocalization, SYAC1 displayed only minor colocalization with markers for ER/plasma membrane (wave 6R; 0.06 ± 0.02; $n = 9$), plasma membrane (wave 131R; 0.02 ± 0.02; $n = 21$ and wave 138R; 0.02 ± 0.03; $n = 12$) and vacuoles (wave 9R; 0.03 ± 0.02; $n = 12$) (Supplementary Fig. 3c, d). These results strongly support that SYAC1 largely resides in the Golgi, TGN, and endosomal and PVC compartments.

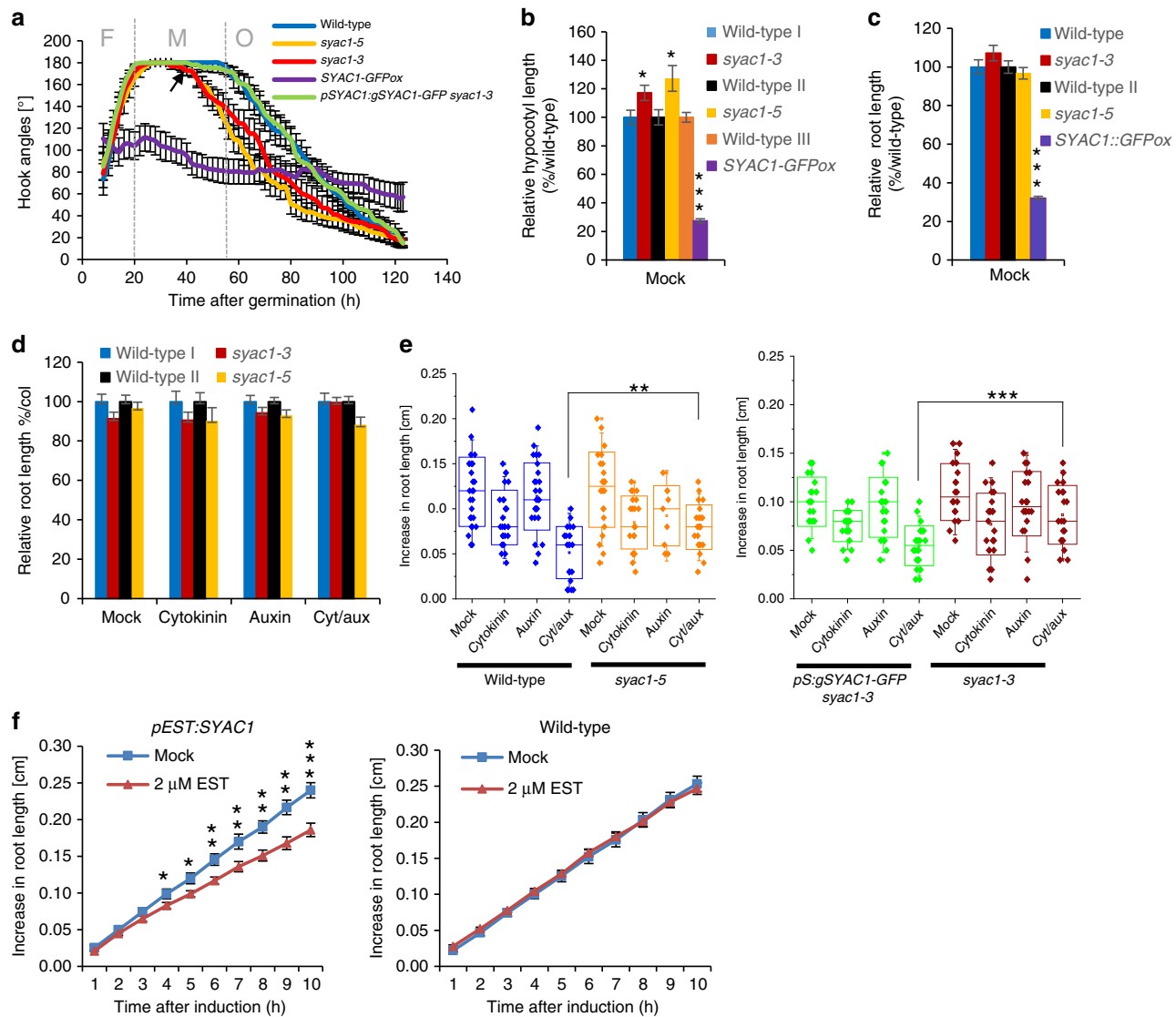

**Fig. 2 Impact of modulated *SYAC1* expression on seedling growth and sensitivity to plant hormones. a** Real time monitoring of formation (F), maintenance (M), and opening (O) phase during apical hook development in wild type, *syac1-3*, *syac1-5*, *SYAC1ox* overexpressor, and *pSYAC1:gSYAC1-GFP syac1-3* seedlings. *syac1-3* and *syac1-5* exhibit premature transition from M to O phase when compared with wild-type control (black arrow indicates time of transition from M to O in *syac1*) and *SYAC1ox* fails to form an apical hook (*n* = 9–11; average ± SE). Analyses of hypocotyl (**b**) and root (**c**) length in 4-day-old seedlings grown in darkness (*n* = 14–20; ±SE) (**b**) and 5-day-old seedlings grown on the light (*n* = 10; ±SE) (**c**). Analysis of root sensitivity to long-term (**d**) and transient (**e**) treatments with hormones. Wild-type and *syac1* mutant seedlings were grown for 5 days on cytokinin (10 μM), auxin (1 μM), and on both hormones together (*n* = 10; average ± SE) (**d**) or 5-day-old seedlings were transferred for 6 h on hormone free or with auxin (0.05 μM), cytokinin (0.1 μM), or auxin plus cytokinin supplemented media (*n* = 10–20) (**e**). Wild type I and III represent respective control to *syac1-3* and *SYAC1-GFPox* lines, respectively. Wild type II was isolated from *syac1-5* heterozygote population (**b**–**e**). In the boxplots, center lines show the medians; box limits indicate the 25th and 75th percentiles as determined by Origin software; whiskers extend 1.5 times the interquartile range from the 25th and 75th percentiles, individual data points are represented by dots (**e**). **f** Transient induction of *SYAC1* expression by estradiol (EST) triggers rapid deceleration of root growth. Five-day-old *pEST:SYAC1* and wild-type seedlings transferred to mock or EST containing medium (*n* = 9–10; average ± SE). Significant differences are indicated as \**P* < 0.05, \*\**P* < 0.01, and \*\*\**P* < 0.001 (*t*-test) (**b**–**f**), compared with mock treated control (**b**, **c**, **f**) or to respective treatment of wild type (**d**, **e**). 1-Naphthaleneacetic acid and N6-benzyladenine used as auxin and cytokinin, respectively.

**SYAC1 is a component of the ECHIDNA/Yip complex**. To further assess molecular function of SYAC1 we identified its molecular interactors using a tandem affinity purification (TAP) assay with SYAC1 used as bait. Proteins including the integral membrane YIP1 family protein (YIP5b; *At3g05280*), β-ketoacyl reductase 1 (KCR1; *At1g67730*), an ubiquitin receptor protein (DSK2; *At2g17200*), and prohibitin 4 (PHB4; *At3g27280*) were recovered by this approach (Supplementary Data 1 and 2). As YIP5b is a member of the YIP (for YPT/RAB GTPase Interacting Protein) family in *Arabidopsis thaliana* that forms a TGN-

localized complex with YIP4a (*At2g18840*) and YIP4b (*At4g30260*) homologs and Echidna (ECH; *At1g09330*) integral membrane protein[34,35], we included them in our subsequent detailed interaction studies. A Yeast two-hybrid assay (Y2H) revealed a strong interaction between SYAC1 and all three YIP family members (Fig. 4a). Moreover, SYAC1 interacted with ECH, but only weakly with KCR1 and not at all with the DSK2 and PHB4 proteins (Fig. 4a). The Y2H results were further validated *in planta* using a bimolecular fluorescence complementation (BiFC) assay. SYAC1 tagged with the C-terminus of Enhanced Yellow Fluorescent

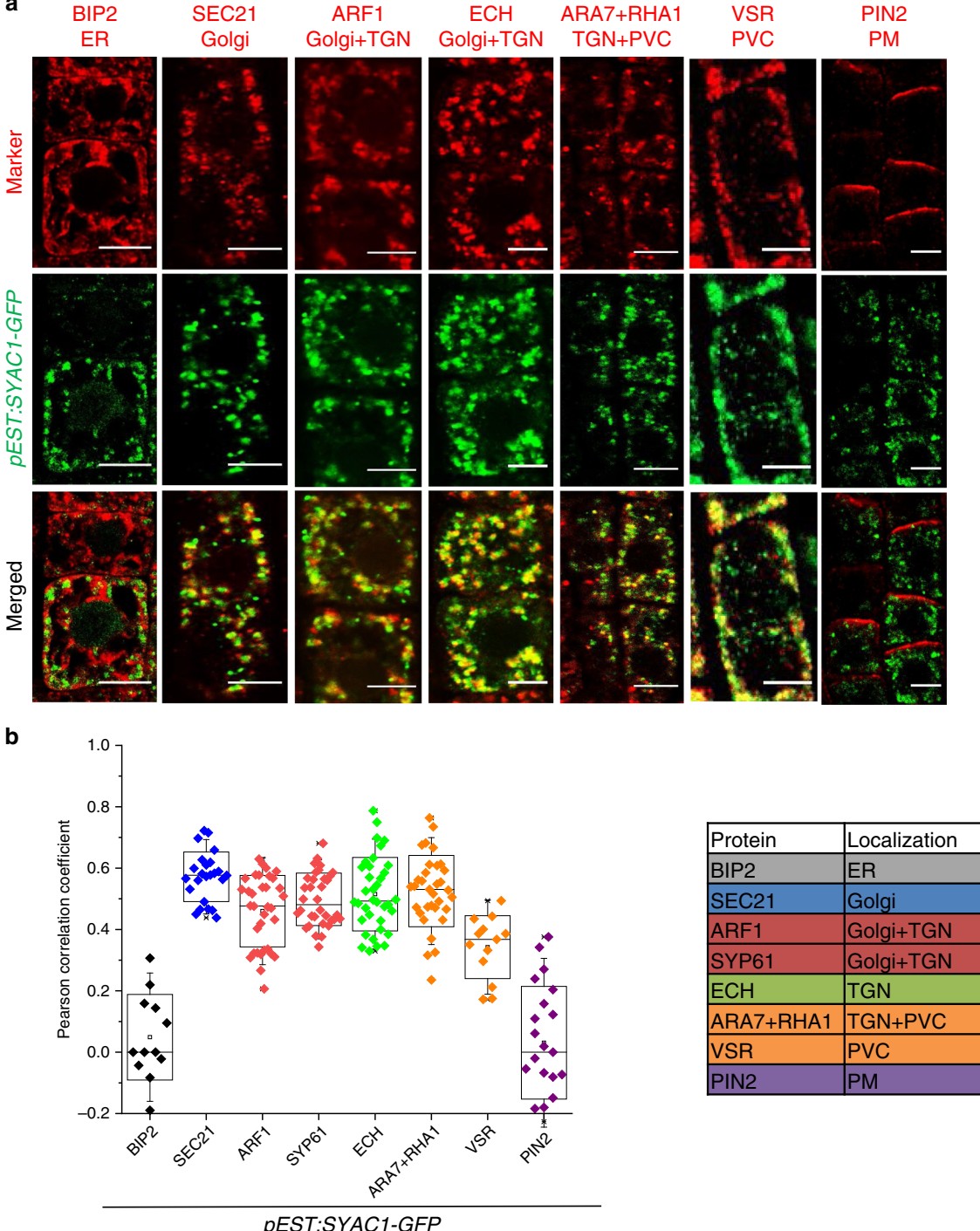

**Fig. 3 SYAC1 colocalizes with Golgi/TGN/endosomal/PVC markers.** Co-imunolocalization of SYAC1-GFP with markers for different subcellular compartments (**a**) and quantification of colocalization using Pearson correlation coefficient (**b**). Five-day-old *pEST:SYAC1-GFP* seedlings grown on mock medium treated with 5 μM estradiol for 6 h used in co-imunolocalization experiments ($n = 10$ roots with 1–5 epidermal cells each). Co-imunolocalization performed using antibodies against anti-GFP and subcellular markers. In the boxplots, center lines show the medians; box limits indicate the 25th and 75th percentiles as determined by Origin software; whiskers extend 1.5 times the interquartile range from the 25th and 75th percentiles, individual data points are represented by dots. ER Endoplasmic reticulum, TGN *trans*-Golgi network, PVC prevacuolar compartment, PM plasma membrane. Scale bar 5 μm.

Protein (EYFP), and YIP5b, YIP4a, YIP4b, ECH, KCR1, DSK2, and PHB4 tagged with the N-terminus of EYFP, were transiently expressed in an *Arabidopsis* root suspension culture. Yellow fluorescence was detected in protoplasts overexpressing SYAC1 in combination with YIP5b, YIP4a, YIP4b, and ECH, indicating the close physical proximity of these proteins in vivo. By contrast, no

EYFP reconstitution was detected in cells overexpressing SYAC1 with KCR1, and PHB4 (Fig. 4b), respectively, in agreement with the result of the Y2H assay. Finally, the interaction between SYAC1 and YIP4a and between SYAC1 and ECH was also confirmed by a co-immunoprecipitation (Co-IP) assay (Fig. 4c). The results from TAP, BiFC, and Co-IP assays revealed SYAC1 has

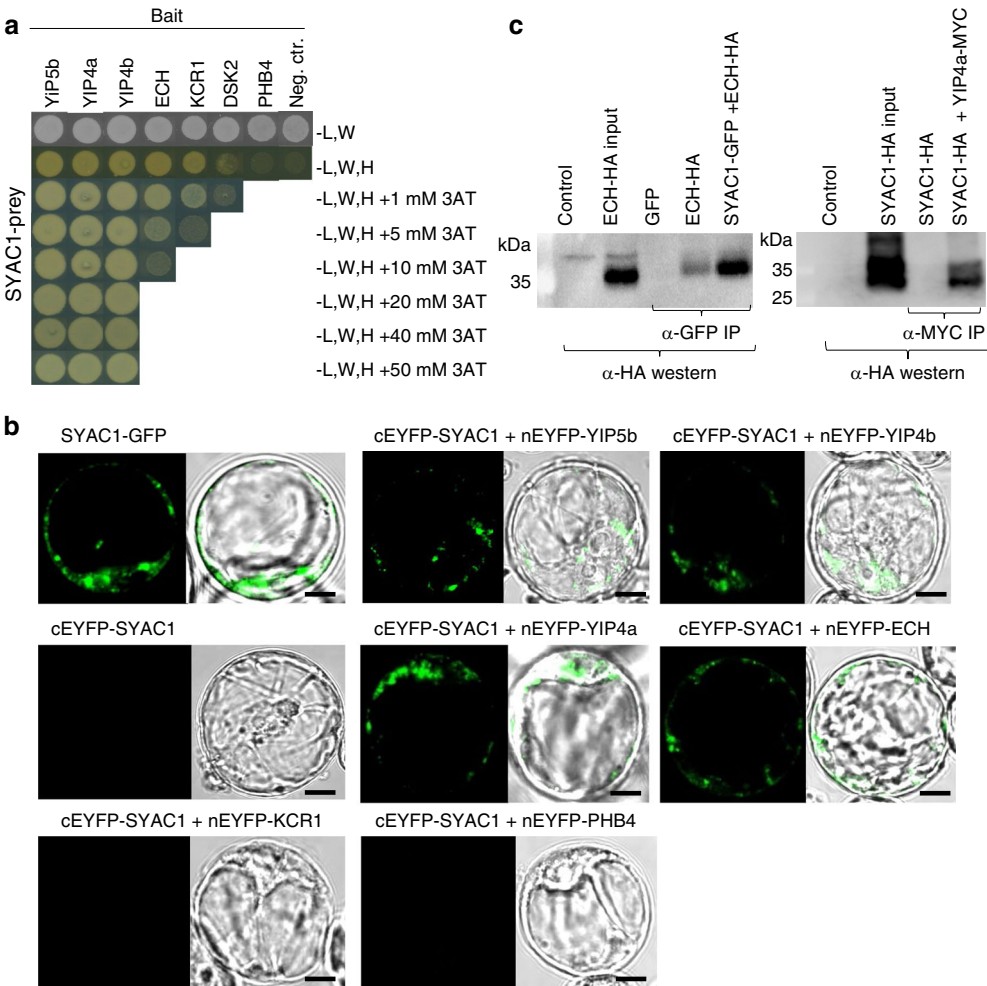

**Fig. 4 SYAC1 interacts with YIP4a, YIP4b, YIP5b, and ECH protein. a** Y2H assay confirms SYAC1 (prey, cloned in pDEST-GADT7) interaction with YIP4a, YIP4b, YIP5b, and ECH. Weaker interaction detected for KCR1 and DSK2, whereas no interaction found between SYAC1 with PHB4 protein. Yeast cells were grown on SD-LWH minimal media without histidin (H), leucin (L) and tryptophan (W), supplemented with 3-amino-1,2,4-triazole (3AT). Empty pDEST-GBKT7 (bait) vector was used as negative control. **b** Bimolecular fluorescence complementation (BiFC) assay in *Arabidopsis* root cell culture protoplasts reveals interaction between SYAC1 and YIP4a, YIP4b YIP5b, ECH, no interaction detected with KCR1, and PHB4 proteins. SYAC1-GFP and SYAC1-cEYFP (C-terminal part of EYFP) were used as a positive and negative control, respectively. Scale bar 10 μm. **c** Co-immunoprecipitation (Co-IP) assay of SYAC1-GFP with ECH-HA and SYAC1-HA with YIP4a-MYC transiently expressed in *Arabidopsis* root cell culture protoplasts. The anti-GFP and anti-MYC antibodies immunoprecipitates were analyzed in a western blot assay with anti-HA antibodies.

interactions with YIP5b, YIP4a, YIP4b, and ECH protein, and suggest it may function in the protein complex involved in maintaining the functionality of the secretory pathway[35].

**SYAC1 affects activity of subcellular trafficking machinery.** SYAC1 localization in Golgi/TGN/endosomal/PVC compartments and the interaction with ECH/YIPs pointed toward a potential function in the secretory pathway[35]. The secretory pathway is of vital importance for all eukaryotic cells, since it manufactures, stores and distributes macromolecules, lipids and proteins as cargo to intracellular and extracellular locations[36]. To assess the involvement of SYAC1 in the regulation of secretion, we performed a transient expression assays in *Arabidopsis* mesophyll protoplasts and evaluated the impact of SYAC1-HAox or HA-SYAC1ox on the secretory index of the α-Amylase (Amy) reporter. Amy is a protein that without any intrinsic sorting signal is secreted extracellularly and can be detected by its endogenous enzymatic activity[37]. The secretion index (SI) was determined by quantifying the ratio of the α-Amylase activity in the medium and in the cells. The expression of the SYAC1 protein decreased the SI from $0.70 \pm 0.04$ ($n = 4$) in control

sample to $0.55 \pm 0.02$ (SYAC1-HAox; $n = 4$) and $0.45 \pm 0.01$ (HA-SYAC1ox; $n = 4$), which suggests a function of SYAC1 as a negative regulator of the anterograde secretory route to the cell surface (Fig. 5a). Due to the colocalization of SYAC1 with markers for PVC compartments, we decided to explore the involvement of SYAC1 in transport to the vacuoles. To do this, an α-Amylase with a vacuolar sorting signal (Amy-Spo) was co-transfected with either *SYAC1-HA* or *HA-SYAC1* encoding plasmids. The SI of α-Amylase (Amy-Spo) was increased from $0.07 \pm 0.01$ ($n = 4$) in the control sample to $0.29 \pm 0.01$ (SYAC1-HAox; $n = 4$) and $0.28 \pm 0.03$ (HA-SYAC1ox; $n = 4$), which suggests that SYAC1 might also interfere with transport to vacuoles leading to more α-Amylase secretion out of the cells (Fig. 5a). The effect of SYAC1 on α-Amylase containing an ER retention signal (Amy-HDEL), which redirects the protein back to the ER was tested. Co-transfection of SYAC1 significantly decreased the SI in protoplasts with leaky retention of α-Amylase from $0.34 \pm 0.01$ ($n = 4$) in the control sample to $0.24 \pm 0.01$ (SYAC1-HAox; $n = 4$) and $0.26 \pm 0.04$ (HA-SYAC1ox; $n = 4$) (Fig. 5a). Altogether these results support the conclusion that SYAC1 coordinates cargo trafficking toward the extracellular space and vacuoles.

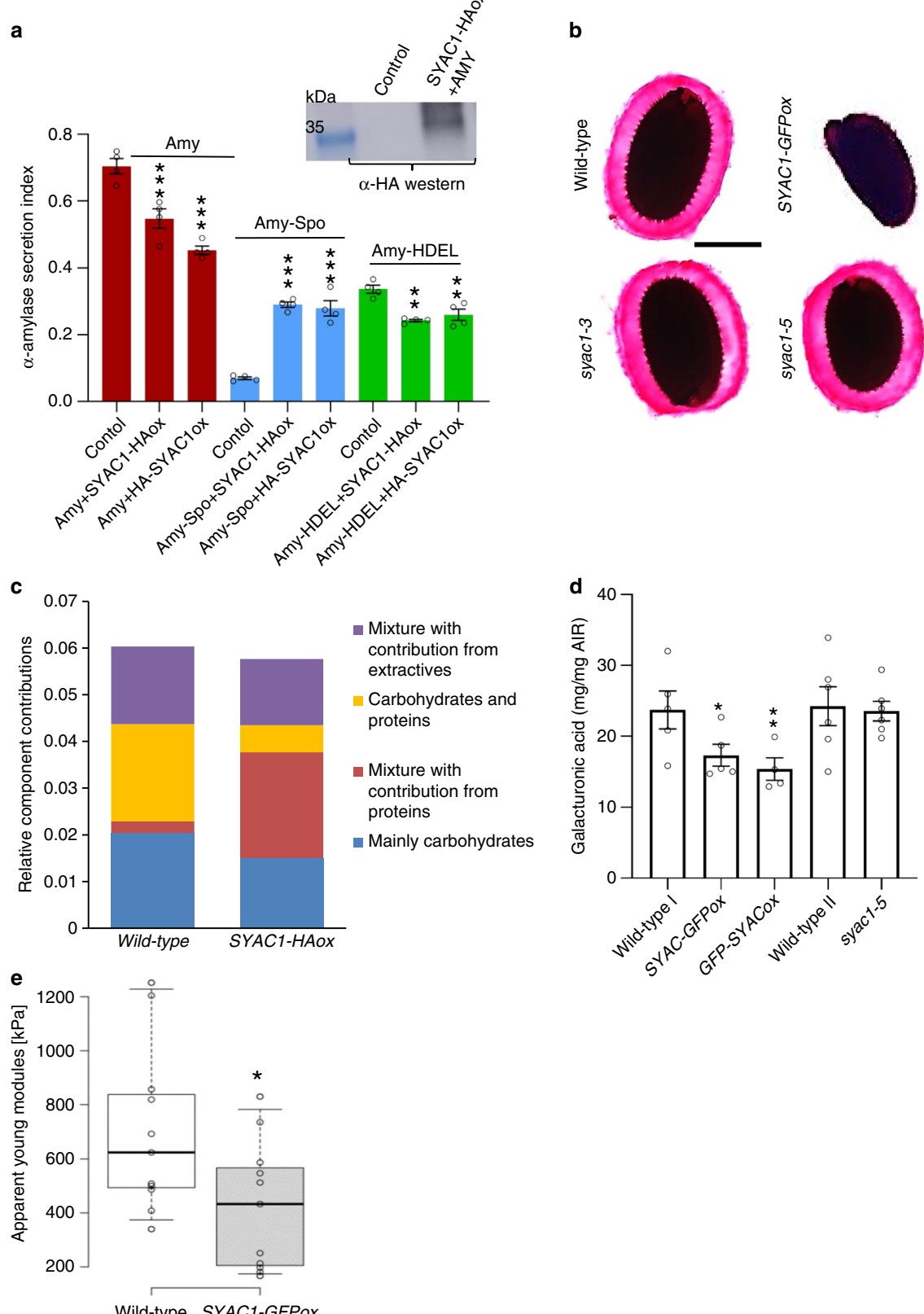

**SYAC1 fine tunes the cell wall composition and mechanics**. In plants, new cell wall components such as pectins and hemicellulose are proposed to be delivered to the extracellular matrix via the secretory pathway[38]. SYAC1 reduction of α-Amylase secretion, along with its Golgi/TGN/endosomal localization and interaction

with YIPs and Echidna proteins, a previously found regulatory components of secretory pathway[35], motivated us to explore the role of SYAC1 in the control of soluble cell wall polysaccharides (pectin and hemicellulose) secretion. Investigating the seed coat epidermis, whose TGN is highly specialized for pectic mucilage

**Fig. 5 SYAC1 regulates composition of cell wall and alters its physical properties. a** *SYAC1* affects α-Amylase secretion index determined by quantifying ratio of the α-Amylase activity in the medium and in the cells. Transient co-expression of *SYAC1* with α-*Amylase* (Amy) and its derivatives carrying different C-terminal sorting motifs, including vacuolar sorting (Amy-spo) and ER retention (Amy-HDEL) motif. (*n* = 4 biological and 2 technical replicates; average ± SE). Expression of SYAC1 fusion constructs confirmed by western blot analysis (insets) using anti-HA specific antibodies. **b** Ruthenium red–stained seed coat mucilage after imbibition of wild-type, *SYAC1-GFPox*, *syac1-3*, and *syac1-5* seeds. Representative images shown (~100 seeds stained per line). Scale bar 200 μm. **c** FT-IR measurements in 4-day-old etiolated hypocotyls show alterations in cell wall composition in *SYAC1-HAox* lines. **d** The amount of galacturonic acid extracted from 4-day-old etiolated hypocotyls (*n* = 4–6, average ± SE). Wild type I represents respective control to *SYAC-GFPox* and *GFP-SYAC1ox*. Wild type II was isolated from *syac1-5* heterozygote population. **e** The apparent Young modules measured by atomic force microscopy (AFM) in 4-day-old etiolated hypocotyls of wild type and *SYAC1-GFPox*. In the boxplots, center lines show the medians; box limits indicate the 25th and 75th percentiles as determined by Origin software; whiskers extend 1.5 times the interquartile range from the 25th and 75th percentiles, individual data points are represented by dots, *n* = 10–14. Significant differences are indicated as *$P < 0.05$, **$P < 0.01$, and ***$P < 0.001$ (*t*-test).

secretion[39], using ruthenium red staining assay revealed that mucilage release from mature seeds was greatly reduced in *SYAC1-GFPox* seeds, relative to wild type (Fig. 5b, Supplementary Fig. 4a). This is in accordance with the anticipated function of SYAC1 as an inhibitor of polysaccharide secretion and with the previously described *ech-1* and *yip4ayip4b* mutants[35]. In *syac1-3* or *syac1-5* mutants, no dramatic change in mucilage secretion could be detected, presumably due to the lack of *SYAC1* expression in seed coat epidermal cells (Supplementary Fig. 4b).

The delivery of new cell wall components during pollen tube growth is a particularly active process, and the main secreted cell wall component in the pollen tube apex is pectin. *SYAC1* expression in tobacco pollen tubes severely affected the apical accumulation of pectin (Supplementary Fig. 4c–h) and resulted in an increased proportion of pollen tubes with growth defects (Supplementary Fig. 4i–k). This observation supports a role for SYAC1 in modulating pectin distribution, presumably by influencing the activity of the secretory pathway. As in *Arabidopsis* root cells the localization of plasma membrane proteins such as PIN1 and PIN2 was not affected, we conclude that SYAC1 might preferentially regulate specific branches of the secretory pathway (Supplementary Fig. 5a). Taken together, the data from altered pectin mucilage and pectin secretion in seeds and pollen tubes overexpressing *SYAC1*, respectively, support a function of SYAC1 in modulating the delivery of cell wall matrix polysaccharides.

To further assess the impact of SYAC1 on cell wall composition, hypocotyls of etiolated seedlings were inspected using Fourier transform-infrared (FT-IR) microspectroscopy. FT-IR analysis revealed that enhanced *SYAC1* expression in plant cells substantially alters the composition of cell walls, which is manifested by a significantly reduced proportion of carbohydrates (Fig. 5c; Supplementary Fig. 5b). To further dissect the qualitative changes in the cell wall, analyses of pectin content and xyloglucans, two major components of cell walls, with modified *SYAC1* expression were performed using hypocotyls of 4-day-old seedlings[40]. A quantitative analysis of galacturonic acid, a key structural component of pectin, did not reveal any significant changes in the hypocotyls of the *syac1* loss of function mutant when compared with controls, consistent with only modest defects on *syac1* hypocotyl growth (cf. Fig. 2b). By contrast, the amount of galacturonic acid extracted from hypocotyls of seedlings with enhanced expression of *SYAC1* was significantly reduced when compared with controls (Fig. 5d). The pectic polysaccharides are secreted in a methylesterified form[41]. In correlation with a reduced amount of pectin in the cell wall, methanol content was decreased in hypocotyls of seedlings overexpressing *SYAC1* (Supplementary Fig. 5c). Unlike pectin, no changes in the amount of fucose, galactose, and xylose, which are indicators of xyloglucan content in the cell wall, were detected in seedlings with either loss or enhanced expression of *SYAC1* (Supplementary Fig. 5d). Together these results suggest that SYAC1 is involved in the control of pectin allocation to the cell wall.

Alterations in cell wall composition might ultimately result in changes in cell wall physical properties. Analyses of etiolated hypocotyls using atomic force microscopy (AFM) revealed a significantly reduced apparent Young modulus in the extracellular matrix of *SYAC1* overexpressor line when compared with control (Fig. 5e). The observed reduction of stiffness of cell walls might affect the elasticity of hypocotyl tissues with increased expression of *SYAC1*, and thus impact on cell expansion, susceptibility of plant tissues to shearing, breaking, but also to attack of pathogens[42–44]. These results support the conclusion that SYAC1 might act as a regulator of pectin distribution, presumably through control of the TGN-mediated trafficking, and ultimately affect the composition and physical properties of cell walls.

**SYAC1 might act as a modulator of YIP/ECH complex activity**. Our screen for interacting partners revealed that SYAC1 can interact with YIPs and ECH, components of a protein complex required for the proper operation of the secretory pathway[35]. Intriguingly, compromised functionality of the YIP/ECH complex leads to cellular and developmental defects reminiscent of these caused by enhanced activity of SYAC1. Similarly to *SYAC1ox*, *yip4a*, *yip4b*, and *ech* loss of function mutants displayed a deficiency in the secretion of pectins, as well as defects in the growth of roots and hypocotyls, and in apical hook development[35]. To explore SYAC1 function as a potential attenuator of YIP/ECH complex activity, we tested whether relief of the SYAC1 mediated inhibition of the secretory pathway might alleviate growth defects caused by defects of the YIP/ECH complex. To test this hypothesis, the *syac1-3* allele was introduced into the *yip4a, yip4b* mutant background, and the resulting combined genotype was phenotyped. Importantly, the *syac1-3 yip4a yip4b* triple mutant displayed significantly improved growth of hypocotyls and shoot organs when compared with the *yip4a, yip4b* double mutant, indicating that SYAC1 might indeed act as a negative regulator of the YIPs/ECH complex (Supplementary Fig. 6a–c). Based on these observations we conclude (i) that the constitutively expressed YIPs and ECH[35] act as generic factors required for the secretion of cell wall components to maintain cell expansion and (ii) that the spatio-temporally controlled expression pattern of SYAC1 serves as a developmentally specific regulator of the YIP/ECH complex to fine tune secretory pathway activity and thereby plant organ growth.

**SYAC1 impacts on root sensitivity to soil pathogens**. In our experimental conditions the basal expression of *SYAC1* in roots is very low and its activation requires simultaneous activation of the auxin and cytokinin pathways, thus raising a question about the function of SYAC1 in this organ. A possible explanation concerns the heterogeneous soil environment, where roots are exposed to a large variety of biotic and abiotic factors. Rhizospheric microbes

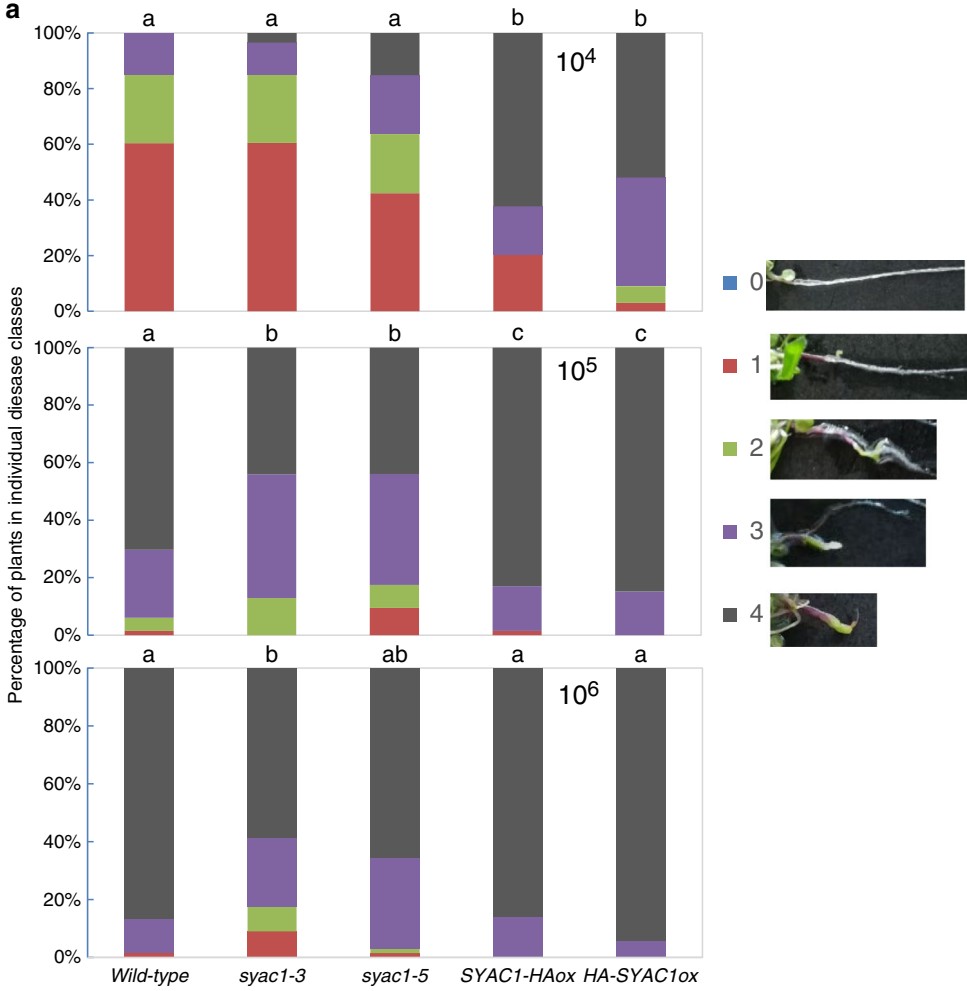

**Fig. 6 Loss of SYAC1 activity increases tolerance to *Plasmodiophora brassicae* infection. a** *syac1-3* and *syac1-5* mutant alleles exhibit reduced sensitivity to fungal infection when compared with wild-type control. In contrast, the overexpression of *SYAC1ox* results in oversensitivity to the pathogen. Five-scale classification was used to evaluate disease severity: 0 (no symptoms), 1 (very small galls mainly on lateral roots and that do not impair the main root), 2 (small galls covering the main root and few lateral roots), 3 (medium to large galls, also including the main root; plant growth might be impaired), and 4 (severe galls on lateral root, main root, or rosette; fine roots completely destroyed; plant growth is reduced). Inoculation was performed with $10^4$, $10^5$, and $10^6$ spore concentration. Significant differences between datasets are indicated by different letters. Data were statistically analyzed using the Kruskal–Wallis test.

are among prominent biotic factors which have developed different strategies, including the modification of phytohormone responses, to penetrate, colonize, and hijack nutrients from host plants[26]. Importantly, the early steps of pathogen infection can be associated with modulation of auxin and cytokinin levels in roots, for instance for the pathogenic protist *Plasmodiophora brassicae*, the causal agent of clubroot disease in cruciferous plants including *Arabidopsis*[45–48]. To examine whether SYAC1, which we identified as an auxin–cytokinin crosstalk component, might be involved in host - *Plasmodiophora brassicae* interaction, we analyzed effects of the pathogen on *SYAC1* expression, and tested the susceptibility of plants with altered *SYAC1* expression to the pathogen. *pSYAC1:GUS* plants at 20 and 28 days after inoculation (dai) displayed blue-stained cells at shoot/root junction as well as islands of GUS positive cells in root branches, whereas no *SYAC1: GUS* activity was detected in noninfected plants (Supplementary Fig. 7a, b). A similar pattern of promoter activity, as that triggered by pathogen infection, was detected in 34-day-old plants treated with auxin and cytokinin for 6 h (Supplementary Fig. 7a). Next, root and shoot phenotypes of plants with modulated expression

of *SYAC1* were analyzed after inoculation with the pathogenic protist at three different spore concentrations ($10^6$, $10^5$, and $10^4$ spores mL$^{-1}$). While a high inoculation pressure should identify tolerant plants, low spore concentrations will reveal oversensitive phenotypes[49,50]. To characterize disease progression we used five categories[50]: 0 denotes no infection, 1 almost no infection, 2 and 3 intermediate infection phenotypes, and 4 a root completely transformed into a clubroot. We observed that both *syac1-3* and *syac1-5* mutant alleles exhibited increased tolerance to the pathogen when compared with the wild-type controls, unlike plants overexpressing *SYAC1*; which exhibited hypersensitivity to pathogen infection (Fig. 6a; Supplementary Fig. 7c). Both root and shoot phenotypes of infected plants are in agreement with these observations. In both mutant plant lines, more roots were classified into lower disease classes at high and medium spore concentrations than in wild-type controls, whereas at the low spore concentration no differences were observed (Fig. 6a). By contrast, all *SYAC1* overexpressors showed more roots in class 4 when compared with the wild type at the lowest spore concentration, thus indicating hypersensitivity to the pathogen. The

patterns determined in roots were also apparent for above-ground tissues, with rosettes of *syac1-3* and *syac1-5* being less affected compared with wild type at higher spore concentrations, and *SYAC1* overexpressors displaying symptoms of hypersensitivity (Supplementary Fig. 7c). Accordingly, the fresh weight of *SYAC1ox* plants at $10^5$ and $10^4$ spores mL$^{-1}$ concentrations was reduced when compared with wild type. In contrast, infected *syac1-3* and *syac1-5* plants were affected less in their fresh weights compared with the overexpressor plants, but similar as wild type (Supplementary Fig. 7d). Together these results indicate that SYAC1-mediated control of cell wall composition in roots might also be involved in root–pathogen interaction.

## Discussion

Auxin and cytokinin play important regulatory roles in various aspects of plant development. The current largely accepted view is that auxin acts mostly antagonistically with cytokinin to control developmental processes[3,51]. In root development, this antagonism is based on the competition between auxin as a promotor of cell division, and cytokinin as a promotor of cell differentiation, with both inputs contributing to the regulation of root meristem size[17,52]. In addition, to specify the root stem-cell niche during embryogenesis, auxin represses cytokinin action[51]. However, this antagonistic interaction between auxin and cytokinin does not occur in all developmental contexts: for instance in the control of cell division in plant suspension cultures, or in the shoot apical meristems, auxin acts synergistically with cytokinin[4,7]. Hence, the concept of yin–yang is probably more accurate, as auxin and cytokinin act together dynamically, with roles that can be paradoxically antagonistic or supportive, to provide robustness to developmental processes[1]. Recently, molecular principles of hormone perception and signal transduction have been deciphered[9–11,13,53–58]. However, the identity of factors and pathways that integrate and transduce inputs between signaling cascades into a proper developmental output is still largely unknown.

Here, we identified a molecular component of the auxin–cytokinin crosstalk *SYNERGISTIC ON AUXIN AND CYTOKININ 1 (SYAC1)*, whose expression in roots is tightly controlled by the auxin–cytokinin balance. Under our experimental conditions, expression of *SYAC1* in roots is suppressed and its activation requires levels of both hormones above a certain minimal threshold. Hence, in cells of the root meristem which typically exhibit a higher endogenous activity of either auxin[27,59] or cytokinin[5] supplementation of their respective hormonal counterpart is sufficient to stimulate gene expression. Intriguingly, in the zone of differentiation and rapid elongation, *SYAC1* expression is dependent on the simultaneous action of both hormonal pathways. Enhanced transcription detected rapidly after only 30 min of the provision of both hormones and the contribution of auxin and cytokinin receptors to synergistic regulation of *SYAC1* transcription, indicate that the gene might be among early common targets of both hormonal pathways. Unlike in roots, in the hypocotyl and cotyledons of germinating seedlings *SYAC1* transcript can be detected without the need for exogenously added hormones. Different requirements for *SYAC1* expression between roots and shoots might reflect distinct configurations of auxin and cytokinin hormonal pathways in these two organs. As a consequence, factors required for *SYAC1* expression might be under suppression in the roots while remaining active in the shoots. In support of such a scenario, transcriptome profiling revealed differences in root and shoot responses to cytokinin, and prolonged exposure of roots to increased cytokinin levels led to an activation of gene clusters typically active only in the shoot[60].

Plant cells are surrounded by complex cell walls, which must remain highly dynamic and adapt to the changing requirements of plants during growth while still providing mechanical support. Furthermore, as a direct contact with the extracellular environment, cell walls serve an important protective function[61]. Pectins, as one of the essential structural components, determine biophysical cell wall properties[62] thus have a significant impact on fundamental plant processes such as elongation growth of plant organs and their adaptation to mechanical stress, abscission of leaves, fruits, seeds or flower organs, as well as a protective role during pathogen infection[63–67]. Intriguingly, recent studies have revealed that pectins are often localized in spatially distinct patterns and these nonuniform pectin distributions might contribute to important aspects of their regulatory function[62].

The expression and functional characterization of SYAC1 suggests that it might be an important regulatory component in the determination of spatio-temporal patterns of pectin distribution and can thus steer the growth and the development of plant organs. A reduced proportion of carbohydrates detected by FT-IR and a decreased amount of galacturonic acid, but unaffected levels of xyloglucans in cell walls of seedlings with enhanced expression of *SYAC1* support a potential function of this protein as a regulator of pectin allocation to the cell wall.

A distinct pattern of pectin distribution in plant cell walls is the result of cell and tissue specific regulation of pectin biosynthesis, its delivery to the cell wall, the control of methyl-esterification and acetylation status of pectin, or its degradation[65,68]. The subcellular localization of SYAC1 and its physical interaction with the ECH/YIP complex, previously linked with the transport of components to the cell wall[30], points toward the SYAC1 involvement in the secretory pathway. A significant decrease of the α-Amylase SI by SYAC1 together with detailed *in planta* observations that reveal negative impact of SYAC1 on release of pectic mucilage from the seed coat, and defective accumulation of pectin at the tip of pollen tubes expressing *SYAC1* support such a scenario. However, taking into account the complexity of the regulatory networks shaping plant cell walls in different tissues and organs, the possibility cannot be excluded that SYAC1 feedbacks onto the activity of factors involved in other processes than the delivery of components to the cell wall. Further studies on SYAC1's potential role in regulating the activity and subcellular localization of proteins involved in the biosynthesis of polysaccharides that comprise pectin, as well as their processing by methylesterification, acetylation or degradation[44,62,69] need to be performed.

Importantly, the pattern of *SYAC1* expression and phenotypic alterations observed in *Arabidopsis* seedlings with modulated activity of SYAC1 support a function of SYAC1 as a regulatory component that might contribute to fine-tuning of pectin allocation to cell walls in a developmentally controlled and tissue-specific manner. *SYAC1* expression is high in the embryonic hypocotyls and in cotyledons, but steadily decreases as seedlings start to germinate. In hypocotyls, the *SYAC1* reporter signal remains strong in short cells close to the apex and is eliminated from elongated cells toward the hypocotyl base. Discrete pattern of the *SYAC1* expression in the upper part of the stem and in the abscission zone of siliques were detected in adult *Arabidopsis* plants. In etiolated seedlings *SYAC1* expression appears concentrated in short cells at the inner and excluded from elongated cells at the outer side of the apical hook. Growth defects such as altered elongation of hypocotyls and roots, or the apical hook phenotype as result of modulated SYAC1 activity accords well with the expression pattern and cellular function of the gene as a regulatory component of pectin allocation to the cell wall[66,67,70].

We propose that SYAC1 contributes to determining a zone of reduced cell expansion by modulating cell wall composition and thereby fine tunes the overall pattern of organ growth kinetics. We hypothesize that constitutively expressed YIPs and ECH

might act as generic factors required for the secretion of cell wall components to maintain cell expansion, and that SYAC1 influences the YIP/ECH complex as a developmentally specific regulator to fine tune pectin distribution pattern and thereby steer plant organ growth.

Why the expression of SYAC1 in roots is normally strongly suppressed and can be unlocked only by the extraneous addition of hormones permitting the synergistic interaction of auxin and cytokinin remains an intriguing question. Restricting the expression of SYAC1 in embryonic roots when compared with embryonic hypocotyls and cotyledons might be part of a developmental mechanism, which coordinates the typical pattern of early germination, namely the emergence of roots prior to the outgrowth of shoot organs.

In roots of germinating seedlings, SYAC1 expression remains low, but can increase when the levels of auxin and cytokinin rise. This indicates that under optimal conditions, the expression of SYAC1 is suppressed and thus does not limit root growth. However, in heterogeneous soil environments roots might be challenged by various abiotic stresses such as excess of aluminum[8] and copper[71], or interactions with rhizospheric microbes[26,46,48], which have been shown to affect the auxin–cytokinin balance and thereby modulate root growth and development. SYAC1 might be a downstream effector by which these stresses lead to decreased root growth.

In soil, a large spectrum of microorganisms can associate with plant roots and the ability of the root system to limit hostile or to promote beneficial interactions with the microbiome is essential for plant survival[26,72]. Microbes penetrate into root systems and trigger major growth and developmental modifications by interfering with the balance of hormonal regulatory networks in the plant. The biotrophic pathogen *Plasmodiophora brassicae*, the causal agent of the clubroot disease in cruciferous plants such as *Brassica napus* and *Arabidopsis thaliana*, is a well-described plant pathogen, which rewires the auxin–cytokinin crosstalk upon infection and increases the levels of both hormones[46,48]. During early phases of the infection these modulations of hormonal activities have been correlated with remodeling and loosening of the cell wall[46], indicating that the plant cell wall might form an important physical barrier to restrain pathogen invasion. The chemical composition of the cell wall, in particular an increased pectin content and lignification, have been implicated in the plant resistance to clubroot disease[73]. Consistent with these reports, enhanced activity of SYAC1, which leads to reduced secretion of pectin and decreased cell wall stiffness, that might increase susceptibility of tissues to breaking, significantly increased the sensitivity of plants to *Plasmodiophora brassicae* when compared with control plants. In contrast, *syac1* loss of function mutants exhibited higher tolerance to the pathogen infection. Hence, the tight regulation of SYAC1, which normally limits its expression in the root, might also interact with pathways controlling root sensitivity to soil pathogens.

In summary, our work reveals an unexpected mechanism by which auxin and cytokinin regulate plant growth and development. We show that SYAC1 is a point of convergence for both hormonal pathways, which is involved in regulation of the cell wall composition and fine-tuning the elongation growth of plant organs. This mechanism might be particularly important in heterogeneous environments, where auxin and cytokinin could act as specific readouts of environmental signals and via SYAC1 rapidly coordinate plant organ growth and adaptive responses.

## Methods

**Plant material and growth conditions**. The *syac1-3* (GABI-KAT 760F05, Col-0, SUL^R) T-DNA insertion line was obtained from the GABI KAT seed collection. Genotyping primers are listed in Supplementary Table 1. The *syac1-5* CRISPR line

was prepared in collaboration with the VBCF Protein Technologies Facility (www.vbcf.ac.at) (see below). The transgenic fluorescent-protein marker lines in Col-0 background have been described elsewhere: mCherry tagged wave line 6, 9, 13, 18, 25, 29, 34, 127, 129, 131, 138[33], SYP61:SYP61-CFP[34]. The *echidna* mutant has been described in ref. [74] and *yip4a-2 yip4b-1* in ref. [35]. *cre1-12; ahk2-2; ahk3-3; cre1-12ahk2-2; cre1-12ahk3-3; ahk2-2,ahk3-3*[75] *tir1-1,afb2-3; tir1-1,* and *afb3-4*[54,76,77]. Seeds of *Arabidopsis* were plated and grown on square plates with solid half strength Murashige and Skoog (MS) medium (Duchefa) supplemented with $0.5\,g\,L^{-1}$ MES, $10\,g\,L^{-1}$ Sucrose, 1% agar, and pH adjusted to 5.9. The plates were incubated at 4 °C for 48 h to synchronize seed germination, and then vertically grown under a 16:8 h day/night cycle photoperiod at 21 °C. Cytokinin and auxin treatments were performed with the N6-benzyladenine cytokinin derivative (Sigma, B3408) and NAA (Sigma, N0640), respectively. Short treatments (6 h) for GUS/GFP expression were performed with 10 μM cytokinin and 1 μM auxin (unless indicated differently). For root growth transient assay 0.1 μM cytokinin and 0.05 μM auxin was used. Gibberellin treatment was performed with 10 μM Gibberellic acid (GA₃) (Sigma, G7645), MeJA treatment with 10 m mM MeJA (Sigma, 392707), ABA treatment with 10 μM ABA (Sigma, A1049) and BR treatment with 1 μM Epibrassinolide (Sigma, E1641). Overall, 10 μM PAC (Sigma,46046) was used as a gibberellin biosynthesis inhibitor. Estradiol treatment was performed with β-Estradiol (Sigma, E8875). All experiments were performed 2–3 times.

**Cloning and generation of transgenic lines**. All cloning procedure was conducted by using Gateway™ (Invitrogen) technology; with the sequences of all used vectors available online (https://gateway.psb.ugent.be/). For promoter analysis of *SYAC1*, an upstream sequence of 2522 bp was amplified by PCR and introduced into the *pDONRP4-P1R* entry vector. Then transcriptional lines (*pSYAC1:GUS, pSYAC1: nlsGFP*) were created: for *pSYAC1:GUS*, an LR reaction with *SYAC1* promoter in *pDONRP4-P1R*, *pEN-L1-S-L2,0*, and *pK7m24GW,0* vectors was performed. For *pSYAC1:nlsGFP* line, an LR reaction with SYAC1 promoter in *pDONRP4-P1R*, *pEN-L1-NF-L2,0*, and *pB7m24GW,0* was performed. To generate overexpressor and inducible lines (SYAC1-GFPox, SYAC1-HAox, HA-SYAC1ox, pEST:SYAC1-GFP, pEST:SYAC1), SYAC1 ORF sequence with or without STOP codon was amplified and fused through a linker (four glycines and one alanine) to GFP or HA tag. The fragments were first introduced into pDONR221, and then into pB2GW7,0 (overexpressor lines), p2GW7,0 (protoplast expression assays), pMDC7 (estradiol inducible line). For GFP-SYAC1ox transgenic line SYAC1 ORF was amplified, introduced to pDONR221 and to the pB7FWG2.0 destination vector. To generate translational fusion line pSYAC1:gSYAC1-GFP, SYAC1 promoter was amplified together with the genomic fragment of the SYAC1 gene, cloned into pDONRP4-P1R and together with pEN-L1-F-L2,0 introduced into pB7m24GW,3. Cloning primers are listed in Supplementary Table 1. All transgenic plants were generated by the floral dip method [78] in Columbia (Col-0) background and transformants were selected on plates with appropriate antibiotic.

**Generation of CRISPR/Cas9 line**. Design of the gRNA for SYAC1 gene, molecular cloning and plant transformation was done in collaboration with VBCF Protein Technologies Facility (www.vbcf.ac.at). Design, specificity and activity of gRNA: GATGGTCAGCAACCACACGA was performed using online available tools: http://cbi.hzau.edu.cn/cgi-bin/CRISPR and http://www.broadinstitute.org/rnai/public/analysis-tools/sgrna-design. gRNA was cloned into pGGZ003 CRISPR/Cas9 destination vector. Transformants resistant to BASTA antibiotic were selected, genomic sequence of SYAC1 amplified with CRISPR Fw and Rv primers (see Supplementary Table 1 below) and sequenced. Individual mutant lines with single base pair insertion in coding sequence (90 bps after the ATG at the place of gRNA binding) were selected. These plants were then propagated to the next generation to obtain homozygote lines. Lines outcrossed of CRISPR/Cas9 cassette were confirmed by PCR for loss of BASTA coding sequence with specific primers (see Supplementary Table 1 below). Only plants without BASTA gene (part of CRISPR/Cas9 vector) were propagated to the next generation. Sensitivity of selected plants to BASTA was confirmed and plants were resequenced to confirm the point mutation.

**Identification of SYAC1 by transcriptome profiling**. SYAC1 was recovered from transcriptome profiling aiming at identification of genes involved in regulation of root branching by auxin and cytokinin. Seven-day-old *Arabidopsis* seedlings of Gal4-GFP enhancer trap line J0121, a marker for xylem pole pericycle[79], were treated with either auxin (1 μM NAA), 10 μM cytokinin (N6-benzyladenine) or both hormones applied simultaneously for 3 h. Fluorescence activated cell sorting (FACS) was performed according to ref. [80]. Approximately 5000 J0121 seeds (per replicate) were sterilized and plated on high growth rate media (0.087% MS medium, 4.5% sucrose) in 16-h light/8-h dark photoperiod at 21 °C. To allow rapid harvesting, seeds were arranged in rows on square plates at a density of ~500 seeds per row on top of nylon mesh (Nitex 03 100/47, Sefar America, Bricarcliff Manor, New York). The mesh with 7-day-old seedlings was transferred on high growth rate media containing the hormonal concentrations indicated. After 3 h, roots were cut off about 1 cm from their tip. Dissected roots were placed in protoplasting solution B [Solution B = (Solution A + 1.5% cellulase, 0.1% pectolyase)] inside 70 μm cell strainers placed in small Petri dishes and incubated for 1 h at room temperature

with agitation. Protoplasted cells were collected from Petri dishes and concentrated by spinning down (at ~800 RCF). The supernatant was aspirated and the cell pellet was resuspended in 1.5 mL of Solution A (600 mM mannitol, 2 mM $MgCl_2$, 0.1% BSA 2 mM $CaCl$, 2 mM MES, 10 mM KCl, pH 5.5). The cell suspension was then filtered through a 40 μm cell strainer. GFP expressing cells were isolated on a fluorescence activated cell sorter (either a Cytomation MoFlo or a Becton Dickinson FACSVantage) fit with a 100 μm nozzle at a rate of 2000–5000 events per second. We mainly used a fluid pressure of 30 psi. Protoplasts from non-GFP expressing Columbia wild-type plants were used as a negative control for establishing sorting criteria based on the following cell properties: (i) a cluster of live protoplasts with intact membranes was selected based on a high forward to side scatter ratio. (ii) GFP positive cells were selected by their emission intensity in the green channel (~530 nm) above negative controls. Cells were sorted directly into lysis buffer (Qiagen RLT buffer), mixed and immediately frozen at −80 °C for later RNA extraction. An autofluorescence filter was established by eliminating cells that fluoresced at equal intensity in the green and orange (~575 nm) channels. Standard Affymetrix protocols were then used for amplifying, labeling and hybridizing RNA samples[80]. RNA was extracted using the RNeasy Plant Mini Kit (Qiagen). A DNase treatment with the RNase-free DNase Set (Qiagen) was carried out for 15 min at 25 °C. Total RNA concentration was determined using a Nanodrop ND-1000 spectrophotometer. All RNA samples were rejected if they did not reach a minimum concentration of 100 ng μL$^{-1}$, a 260 nm/280 nm ratio between 1.8 and 2.0, and an RNA integrity number superior to 7.5, measured with an Agilent 2100 Bioanalyzer (Agilent, USA). *Arabidopsis* Tiling 1.0 R arrays (Affymetrix) were hybridized at the VIB Nucleomics Core (www.nucleomics.be) according to the manufacturer's instructions. Data were normalized from CEL files using the robust multiarray average algorithm implemented in the Bioconductor package Affy (v1.24.2)[81]. The probe annotation was obtained from athtiling1.0rcdf[82]. Differential expression analysis was determined using the empirical Bayes (*eBayes*) function from the *Limma* package (v2.14.0) in R v2.8.0[83]. *P* values were calculated and then transformed into false discovery rates, or *Q* values according to the method described by Storey and Tibshirani[84] as implemented in the R package *qvalue*.

**RNA extraction, RT, and qPCR.** Total RNA was extracted from roots of 5-day-old plants under all conditions (untreated, 1 μM auxin, 10 μM cytokinin and both together for 3 h) using the RNeasy Plant Mini kit (Qiagen). Overall, 1 μg total mRNA was used to generate cDNA using the iScript™ cDNA Synthesis Kit (BioRad). *SYAC1* expression was quantified with specific primer pair (see Supplementary Table 1 below). Three hundred and eighty-four(384) well plates (Roche) were loaded using a JANUS Automated Workstation (PerkinElmer) with a 5 μL reaction containing 2.5 μL Luna® Universal qPCR Master Mix (New England BioLabs). qPCRs were performed using the LightCycler 480 (Roche). Samples ($n \geq 3$) were measured in technical triplicates, and the expression of *PP2A* or *EEF1A* (*AT1G13320*; *AT5G60390*; see Supplementary Table 1 below) was used as a reference[85]. Data were analyzed using the LightCycler 480 Software (Roche).

**Phenotypic analysis.** For root and hypocotyl length analyses, seedlings were photographed and lengths were measured with ImageJ software version 1.52 (https://imagej.nih.gov/ij/). About 10–30 seedlings were processed and three independent experiments were performed. *t*-test was used for statistics.

**Analysis and statistics of the apical hook development.** Development of seedlings was recorded at 1-h intervals for 5 days at 21 °C with an infrared light source (880 nm LED; Velleman, Belgium) by a spectrum-enhanced camera (EOS035 Canon Rebel Xti; 400DH) with built-in clear wideband-multicoated filter and standard accessories (Canon) and operated by EOS utility software. Angles between hypocotyl axis and cotyledons were measured by ImageJ software version 1.52. At least ten seedlings with synchronized germination were processed and the experiment was repeated three times with similar results. For more details see ref. [86].

**Histochemical and histological analysis.** To detect GUS activity, mature embryos, seed coats, seedlings, and mature plants were incubated in reaction buffer containing 0.1 M sodium phosphate buffer (pH 7), 1 mM ferricyanide, 1 mM ferrocyanide, 0.1% Triton X-100, and 1 mg mL$^{-1}$ X-Gluc for 12 h in dark at 37 °C. Afterward, chlorophyll was removed by destaining in 70% ethanol. Seedlings were cleared[87] by incubation in a solution containing 4% HCl and 20% methanol for 10 min at 65 °C, followed by 10 min of incubation in 7% NaOH/60% ethanol at room temperature. Next, seedlings were rehydrated by successive incubations in 60, 40, 20, and 10% ethanol for 15 min, followed by incubation (15 min up to 2 h) in a solution containing 25% glycerol and 5% ethanol. Finally, material was mounted in 50% glycerol. GUS expression was monitored by differential interference contrast microscopy (Olympus BX53).

Immunolabeling in roots (4- to 5-day-old seedlings) was performed using an automated system (Intavis in situ pro) according to published protocol[88]. Roots were fixed in 4% paraformaldehyde for 1 h in vacuum at room temperature. Afterward, seedlings were incubated for 30–45 min in PBS (2.7 mM KCl, 137 mM NaCl, 4.3 mM $Na_2HPO_4$ $2H_2O$, and 1.47 mM $KH_2PO_4$, pH 7.4) containing 2% driselase (Sigma), and then in PBS supplemented with 3% NP40 and 20% DMSO.

After blocking with 3% BSA in PBS, samples were incubated with primary antibody for 2 h. Antibody dilutions were rabbit anti-BIP2 (1:200) (Agrisera AS09481), rabbit anti-SEC21 (1:800) (Agrisera AS08327), rabbit anti-ARF1 (1:600) (Agrisera AS08325), rabbit anti-SYP61 (1:200)[89], rabbit anti-ECH (1:600) (kindly provided by R.P. Bhalerao, Umea Plant Science Centre), rabbit anti-ARA7 + RHA1 1:1 (1:100)[90], rabbit anti-VSR (1:600) (kindly provided by Liwen Jiang, The Chinese University of Hong Kong), rabbit anti-PIN1 (1:1000)[91], rabbit anti-PIN2 (1:1000) (provided by C. Luschnig, University of Natural Resources and Life Sciences, Vienna), and mouse anti-GFP (1:600) (Sigma G6539). Secondary antibody incubation was carried on for 2 h. Anti-mouse-Alexa 488 (Life Technologies, 1252783) and Cy3-conjugated anti-rabbit antibody (Sigma, C2306) were diluted 1:600 in blocking solution. Samples were mounted in solution containing 25 mg mL$^{-1}$ DABCO (Sigma) in 90% glycerol, 10% PBS, pH 8.5. Signal was monitored using a confocal laser scanning microscope (LSM 700, Zeiss). Images were analyzed by using ImageJ software version 1.52.

**Colocalization analysis.** Pearson's correlation coefficient (R) was used for colocalization analyses: the analysis is based on the pixel intensity correlation over space and was performed using Image J software version 1.52. After splitting the two channels, region of interest (ROI) was identified. For our analysis, one cell was considered as one ROI; in every root ~1–5 cells (five ROIs) were measured and a minimum of ten roots were used. Colocalization plug-in using an automatic threshold was used to obtain Rcoloc value, which represent Pearson's correlation coefficient.

**Phylogenetic analysis.** Sequences from the *Arabidopsis thaliana* SYAC1 protein family (*AT1G15590, AT1G15600, AT1G15610, AT1G15620, AT1G15625, AT1G15630, AT1G15640,* and *AT1G44010*) were aligned using ClustalW[92]. Phylogenetic tree was generated with MEGA7 (Molecular Evolutionary Genetics Analysis Version 7.0 for Bigger Datasets; www.megasoftware.net) using Neighborjoining methods with a complete deletion mode. Bootstrap tests were performed with 1000 replications. Poisson correction distance was adopted and rates among sites were set as uniform.

**Confocal imaging and image analysis.** Zeiss LSM 700 confocal scanning microscope using either ×20 or ×40 (water immersion) objectives were employed to monitor expression of fluorescent reporters. GFP (YFP) and Cy3 signals were detected either at 488 nm excitation/507 nm emission or 550 nm excitation/570 nm emission wavelength, respectively. Quantification of immunodetected PIN1 and PIN2 expression in root meristems was performed by measurement of membrane signal in cortex and epidermal cells, respectively. Signal in approximately ten cells in a minimum of ten roots was measured using ImageJ software version 1.52. For cell size measurement, roots of 5-day-old wild type and *SYAC1ox* plants were stained for 5 min with FM4-64 dye (Thermo Fisher, T13320) followed by image acquisition. Cell size was measured by ImageJ software version 1.52. Significant differences were evaluated using *t*-test; $n = 15–20$ roots, two biological replicates.

**Transient expression in root suspension culture protoplasts.** The transient expression assays were performed on 4-day-old *Arabidopsis* root suspension culture by PEG mediated transformation. Protoplasts were isolated in enzyme solution (1% cellulose (Serva), 0.2% macerozyme (Yakult), in B5—0.34 M glucose-mannitol solution (2.2 g MS with vitamins, 15.25 g glucose, 15.25 g mannitol, pH to 5.5 with KOH) with slight shaking for 3–4 h, and afterward centrifuged at $800 \times g$ for 5 min. The pellet was washed and resuspended in B5 glucose-mannitol solution to a final concentration of $4 \times 10^6$ protoplasts/mL. DNAs were gently mixed together with 50 μL of protoplast suspension and 60 μL of PEG solution (0.1 M $Ca(NO_3)_2$, 0.45 M mannitol, 25% PEG 6000) and incubated in the dark for 30 min. Then 140 μL of 0.275 M $Ca(NO_3)_2$ solution was added to wash off PEG, wait for sedimentation of protoplasts and remove 240 μL of supernatant. The protoplast pellet was resuspended in 200 μL of B5 glucose-mannitol solution and incubated for 16 h in the dark at room temperature. Transfected protoplasts were mounted on the slides and viewed with Zeiss LSM 700 confocal scanning microscope.

**Transient expression in *Arabidopsis* mesophyll protoplasts.** Mesophyll protoplasts were isolated from rosette leaves of 4-week-old *Arabidopsis* plants grown in soil under controlled environmental conditions in a 16:8 h light/dark cycle at 21 C. Leaves were cutted with razor and protoplasts were isolated in enzyme solution (1,5% cellulose (Serva), 0.4% macerozyme (Yakult), in MES-mannitol solution (0.4 M mannitol, 20 mM KCl, 20 mM MES, 10 mM $CaCl_2$, pH to 5.7 with KOH) with slight shaking for 3–4 h, and afterward centrifuged at $800 \times g$ for 5 min. The pellet was washed in W5 solution (154 mM NaCl, 152 mM $CaCl_2$, 5 mM KCl, 2 mM MES) and resuspended to a final concentration of $4 \times 10^6$ protoplasts/mL in MMg (0.4 M mannitol, 15 mM $MgCl_2$, 4 mM MES). DNAs were gently mixed together with 50 μL of protoplast suspension and 60 μL of PEG solution (0.1 M $CaCl_2$, 0.4 M mannitol, 25% PEG 4000) and incubated in the dark for 10 min. Then 140 μL of W5 solution was added to wash off PEG, wait for sedimentation of protoplasts and remove 240 μL of supernatant. The protoplast pellet was resuspended in 200 μL of W5 solution and incubated for 16 h in the dark at room temperature. More details are described in ref. [93].

**Characterization of SYAC1 effects in tobacco pollen tubes**. Pollen was harvested from 8-week-old tobacco (*Nicotiana tabacum*) plants and resuspended in growth medium (5% (w/v) sucrose, 12.5% (w/v) PEG-6000, 0.03% (w/v) casein hydrolysate, 15 mM MES-KOH pH 5.9, 1 mM CaCl₂, 1 mM KCl, 0.8 mM MgSO₄, 1.6 mM H₃BO₃, 30 μM CuSO₄, 10 μg/mL rifampicin) and filtered onto cellulose acetate filters (Sartorius, Göttingen, Germany). The filters were placed on filter paper (Whatman, Maidstone, UK) moistened with growth medium. Within 5–10 min of harvesting, the collected pollen was transformed by bombardment with plasmid coated 1 μm gold particles using a helium-driven particle accelerator (PDS-1000/He; BIO-RAD, Munich, Germany) using 1350 psi rupture discs and a vacuum of 28 in. of mercury. Gold particles (1.25 mg) were coated with 3–7 μg of plasmid DNA, by precipitating the DNA with 1 M CaCl₂ and 16 μM Spermidine, and washing the gold particles three times with 95% (v/v) ethanol. *SYAC1* ORF sequence without STOP codon was amplified and fused through a linker (four glycines and one alanine) to *mCherry* tag. The fragment was first introduced into *pDONR221* and then into *pLatGW* (Invitrogen) plasmid. Cloning primers are listed in Supplementary Table 1. After bombardment pollen was resuspended in growth medium and grown for 5–8 h in small droplets of media directly on microscope slides as previously described[94]. Fluorescent pollen tubes were analyzed for apical pectin deposition by staining with ruthenium red (Sigma) at a final concentration of 0.01 % (w/v) and imaged under the light microscope within 5–10 min after addition of the dye, as previously described[94]. Microscopic examination was performed using an Axiovert200 epifluorescence microscope fitted with an LD Achroplan ×63/ 0.75 Korr Ph2 (DICIII) objective and an AxioCam HR color camera, and using Axio Vision Rel 4.6 Software (all Carl Zeiss Jena, Germany). Staining patterns were manually examined for deposition thickness, or automatically for the apical occupancy of red dye using Fiji software version 1.52 (https://fiji.sc/). The subcellular localization of SYAC1-mCherry in tobacco pollen tubes was performed using an LSM 880 Airyscan confocal microscope (Carl Zeiss Jena, Germany) and Zen blue software as previously described[95]. Pollen tube morphologies were assessed by scoring the incidence of normal, stunted or growth-arrested (nongerminating) cells among fluorescent pollen tubes[96].

**Co-immunoprecipitation (Co-IP) assays**. For the Co-IP assays, proteins were expressed in root suspension culture protoplasts (see above) and extracted from the cell pellet as described previously[97]. In more detail, *Arabidopsis* cell culture protoplasts were transfected with 7.5 μg of each construct by the polyethylene-glycol (PEG) method (see above). Vectors containing *ECH-HA* and *YIP4a-Myc* were kindly provided by R.P. Bhalerao, Umea Plant Science Centre. For testing protein interactions, co-transfected protoplasts were extracted in lysis buffer containing 25 mM Tris–HCl (pH 7.8), 10 mM MgCl2, 75 mM NaCl, 5 mM EGTA, 60 mM β-glycerophosphate, 1 mM dithiothreitol, 10% glycerol, 0.2% Igepal CA-630, and Protein Inhibitor Cocktail (Sigma-Aldrich, P9599-5ML). The cell suspension was frozen in liquid nitrogen, then thawed on ice and centrifuged for 10 min at 14,000 rpm at 4 °C. In a final volume of 100 μL the resulting supernatant was mixed with 150 mM NaCl and 1.5 μL of anti-Myc antibody (Covance, 9E10) or 1 μg anti-GFP (JL-8, Clontech) or 2 μL of anti-HA antibody (Covance, 16B12) for 2 h at 4 °C on a rotating wheel. Immunocomplexes were captured on 15 μL of Protein G-Magnetic Beads (BIO-RAD), which were previously equilibrated in TBS buffer, for 2 h at 4 °C on a rotating wheel, washed three times in 25 mM sodium phosphate, 5% glycerol, and 0.2% Igepal CA-630 buffer, and eluted by boiling with 40 μL of SDS sample buffer. The presence of immunocomplexes was assessed by probing protein gel blots with either rat anti-HA-peroxidase (3F10, Roche, dilution 1:5000) or anti-Myc antibody (Covance, 9E10) at a 1:2000 dilution.

**Western blot**. Freshly harvested and grinded 5-day-old seedlings or 5 × 10⁵ transfected protoplasts were lysed with 50 μL of extraction buffer (50 mM Tris HCl pH 7.5, 150 mM NaCl, 10 mM MgCl2, 1% (w/v) Triton X-100, 5 mM DTT, PhosSTOP phosphatase inhibitor cocktail (Roche), 1x EDTA free-Complete Protease Inhibitor Cocktail (Roche). After vortexing vigorously for 30 s, the samples were centrifuged at 12,000 × g for 10 min at 4 °C and supernatants were collected. For immunoblot analysis, 25 μL of collected supernatants were separated by SDS-PAGE and subjected to immunoblotting using anti-GFP-HRP antibody (130-091-833, Miltenyi Biotec, dilution 1:5000) and Anti-HA-Peroxidase, (3F10, Roche, dilution 1:5000).

**Bimolecular fluorescence complementation (BiFC) assay**. To generate constructs for BiFC assay, the ORFs for *SYAC1*, *YIP4a*, *YIP4b*, *YIP5b*, *ECH*, *KCR1*, and *PHB4* proteins were cloned into the *pDONRZeo* vector. Next, the ORFs were transferred from their respective entry clones to the gateway vector *pSAT4-DEST-n (174)EYFP-C1* (ABRC stock number CD3-1089) or *pSAT5-DEST-c(175-end)EYFP-C1* (ABRC stock number CD3-1097), which contained the N-terminal 174 amino acids of enhanced yellow fluorescent protein (EYFPⁿ) or the C-terminal 64 amino acids of EYFP (EYFPᶜ), respectively. The fusion constructs encoding cEYFP-SYAC1 and nEYFP-YIP4a, nEYFP-YIP4b, nEYFP-YIP5b, nEYFP-ECH, nEYFP-KCR1 or nEYFP-PHB4 proteins were mixed at a 1:1 ratio and transfection of root suspension culture protoplasts (see above) was performed. *SYAC1* in P2YGW7 was used as a positive control.

**Yeast two-hybrid assays**. Y2H assay was performed using the GAL4-based two-hybrid system (Clontech). Full-length *SYAC1* and *YIP4a*, *YIP4b*, *YIP5b*, *ECH*, *KCR1*, *DSK2*, and *PHB4* ORFs were cloned into *pDEST-GADT7* and *pDEST-GBKT7* (Clontech) to generate the constructs AD-SYAC1 and BD-YIP4a (*YIP4b*, *YIP5b*, *ECH*, *KCR1*, *DSK2*, *PHB4*). The constructs were transformed into the yeast strain PJ69-4A with the lithium acetate method:100 μL of competent cells resuspended in TE/LiAc solution (100 mM LiAc, 10 mM Tris-HCl, 1 mM EDTA, pH 7.5) were mixed with DNA. 600 μL of PEG/LiAc solution (50% PEG, 100 mM LiAc, 10 mM Tris-HCl, 1 mM EDTA, pH 7.5) was added to each reaction and incubated for 30 min. The yeast cells were afterward centrifuged at 800 × g for 5 min, resuspended in 500 μL of TE buffer (10 mM Tris-HCl, 1 mM EDTA, pH 7.5) and grown on minimal medium (–Leu/–Trp). Transformants were plated (minimal medium, –Leu/–Trp/–His without or with increasing concentration of 3-Amino-1,2,4-trizol) to test the protein interactions.

**α-Amylase enzymatic assay**. α-Amylase assays and calculations of the SI were performed as described[37]. Briefly,α-Amylase expression constructs were kindly provided by P. Pimpl and transfections were performed in *Arabidopsis* mesophyll protoplasts (see above). α-Amylase was extracted with extraction buffer (50 mM C₄H₆O₅, 50 mM NaCl, 2 mM CaCl₂, 0.05 % NaN₃, pH to 5.2) and activity was measured with a kit Ceralpha (Megazyme). The reaction was performed in a microtiter plate at 37 °C with 30 μL of extract and 30 μL of substrate. The reaction was stopped by the addition of 150 mL of stop buffer. The absorbance was measured at a wavelength of 405 nm. Experiment was performed four times including two biological replicates with two technical replicates per experiment. The α-Amylase activity in medium (secreted to the apoplast) and in cells (retained in the inner part of the cell) was measured in two to three time points and SI (ratio between α-Amylase in medium an in cells) as a normalized parameter, which allows to compare the different replicates and neutralizes the variability, was calculated. Protoplasts transformed without any plasmid were used for blank measurement.

**AFM measurements and apparent Young's modulus calculations**. The AFM data were collected and analyzed as described elsewhere with minor changes[66]. To examine extracellular matrix properties the turgor pressure was suppressed by seedlings immersion in a hypertonic solution (0.55 M mannitol) for at least 30 min before examination. 4 day-old seedlings grown in darkness were placed on petri dishes filled with 1% agarose and 10% mannitol and immobilized by low melting agarose (0.7% with 10% mannitol). The focus was set on the anticlinal (perpendicular to the organ surface) cell walls and its extracellular matrix. To ensure proper indentations (especially in the bottom of the doom shape between two adjacent cells regions), cantilevers with long pyramidal tip (14–16 μm of pyramidal height, AppNano ACST-10), with a spring constant of 7.8 N/m were used. The instrument used was a JPK Nano-Wizard 3.0 and indentations were kept to <10% of cell height (typically indentations of 500 nN force). Three scan maps per sample were taken over an intermediate region of the hypocotyls, using a square area of 25 × 25 μm, with 16 × 16 measurements, resulting in 1792 force–indentation experiments per sample. The lateral deflection of the cantilever was monitored and in case of any abnormal increase the entire dataset was discarded. The apparent Young's modulus (EA) for each force–indentation experiment was calculated using the approach curve (to avoid any adhesion interference) with the JPK Data Processing software (JPK Instruments AG, Germany). To calculate the average EA for each anticlinal wall, the total length of the extracellular region was measured using masks with Gwyddion 2.45 software (at least 20 points were taken in account). The pixels corresponding to the extracellular matrix were chosen based on the topography map. For topographical reconstructions, the height of each point was determined by the point of contact from the force–indentation curve. A total of 12–14 samples were analyzed. A standard *t*-test was applied to test for differences between genotypes.

**Ruthenium red staining of seeds**. Mature seeds were incubated in 0.01% (w/v) aqueous solution of Ruthenium red for 1 h while gently shaking. Seeds were washed with water, mounted in water and viewed using a DIC Olympus BX53 microscope. Approximately 100 seeds from each genotype were stained with ruthenium red and 30 representative images were recorded.

**Tandem affinity purification (TAP)**. TAP assay was performed in *Arabidopsis* cell suspension culture as described[98] with minor modifications. Briefly, SYAC1 was produced as N-terminally tagged GSᵀᴱⱽ fusion in PSB-D cell culture. Proteins were extracted for 1 h in TAP extraction buffer in which 0.1% NP40 was replaced by 1% digitonin. After 30 min of extraction, 0.1% benzonase was added to the mixture. After a two-step centrifugation, protein complexes were bound to IgG resin for 1 h and nonspecific proteins were washed away. IgG elution was done with TEV protease, and the eluted fraction was bound to streptavidin resin. All wash steps and TEV elution was done in TEV elution buffer in which 0.1% NP40 was replaced by 0.2% digitonin. After the final streptavidin wash step, one additional wash step was performed with TEV buffer without detergent. Finally, proteins were eluted from the streptavidin resin with NuPAGE sample buffer containing 20 mM Desthiobiotin. Eluted proteins were subjected to a short NuPAGE run and in-gel

digested with Trypsin. Protein interactors were identified by mass spectrometry using an LTQ Orbitrap Velos mass spectrometer. Proteins with at least two matched high confident peptides were retained. Background proteins were filtered out based on frequency of occurrence of the copurified proteins in a large dataset containing 543 TAP experiments using 115 different baits[98].

**Cell wall analyses**. Analyses were performed on 4-day-old dark grown hypocotyls using an alcohol-insoluble residue (AIR) prepared as follows. Seeds of *Arabidopsis* were plated and grown on square plates with Milieu *Arabidopsis* medium (Duchefa) supplemented with $0.32 \, g \, L^{-1}$ CaNO$_3$, $10 \, g \, L^{-1}$ Suc, 0.8% agar and pH adjusted to 5.75. The plates were incubated at 4 °C for 48 h to synchronize seed germination, and then vertically grown in darkness at 18 °C. Freshly collected samples were submerged into 96% ethanol, grinded and incubated for 30 min at 70 °C. The pellet was then washed twice with 96% ethanol and once with acetone. The remaining pellet of AIR was dried in a fume hood overnight at room temperature. Dry weight of each sample was measured. After saponification of the AIR (1–4 mg) with 0.05 M NaOH, supernatant containing methyl ester released from the cell wall was separated from the pellet with polysaccharides. Pectins were extracted from the pellet with 1% ammonium oxalate at 80 °C for 2 h[64,65,99–101]. Galacturonic acid was then quantified by colorimetry: pectin extracts were treated with six volumes of sulfuric acid for 15 min at 100 °C, cooled down and mixed with 0.2 V of *meta*-hydroxydiphenyl solution (0.15% in 0.5% NaOH) to read absorbance at 520 nm. Methyl ester was quantified from NaOH supernatant by colorimetry at 420 nm: enzymatic oxidation of methanol with alcohol oxidase was realized in a 0.1 M pH 7.5 phosphate buffer for 20 min at 30 °C before coloration for 10 min at 67 °C in a 10 mM acetyl acetone/24.5 mM acetic acid/1 M ammonium acetate solution in water. The monosaccharide composition of the noncellulosic fraction was determined by hydrolysis of 100 µg AIR with 2 M TFA for 1 h at 120 °C. After cooling and centrifugation, the supernatant was dried under a vacuum, resuspended in 200 µL of water and retained for analysis. All samples were filtered using a 20 µm filter caps, and quantified by HPAEC PAD on a Dionex ICS-5000 instrument (Thermo Fisher Scientific) equipped with a CarboPac PA20 analytical anion exchange column (3 mm × 150 mm) and a PA20 guard column (3 mm × 30 mm)[102]. The following separation conditions were applied: an isocratic gradient of 4 mM NaOH from 0 to 6 min followed by a linear gradient of 4 mM NAOH to 1 mM NaOH from 6 to 19 min. At 19.1 min, the gradient was increased to 450 mM NaOH to elute the acidic sugars.

**Fourier transform-infrared spectroscopy (FT-IR)**. Spectra were recorded from the 4-day-old dark grow hypocotyls sections in transmission mode on a Bruker Tensor 27 spectrometer equipped with a Hyperion 3000 microscopy accessory and a liquid N$_2$ cooled $64 \times 64$ mercury cadmium telluride focal plane array (FPA) detector. The entire setup was placed on a vibration-proof table. Spectra were recorded in the region $900$–$3900 \, cm^{-1}$, with $4 \, cm^{-1}$ spectral resolution and 32 scans co-added in double sided, forward-backward mode. FPA frame rate was 3773 Hz and integration time 0.104 ms, with offset and gain optimized for each sample between 180–230 and 0–1, respectively. A low pass filter and an aperture of 6 mm were used. Four hypocotyls for each line were used and five spectra from each of three different regions were used in the analyses. Background was recorded on a clean, empty spot on the CaF$_2$ carrier (Crystran Ltd, UK) and automatically subtracted. Fourier transformation was carried out using a zero filling factor of two, and Blackman–Harris three-term apodization function. Phase correction was set to the built-in Power mode with no peak search and a phase resolution of 32. White light images were recorded with a Sony ExwaveHAD color digital video camera mounted on the top of the microscope and exported as jpg files. Spectra were recorded using OPUS (version 6.5 and 7, Bruker Optics GmbH, Ettlingen, Germany), cut to the fingerprint region of $900$–$1800 \, cm^{-1}$ and exported as.mat files for subsequent processing and analysis. The exported spectra were pre-processed by an open-source software developed at the Vibrational Spectroscopy Core Facility in Umeå (https://www.umu.se/en/research/infrastructure/visp/downloads/), written in MATLAB (version 2014a-2018b, Mathworks, USA), using asymmetric least squares baseline correction[103]; (lambda: 100,000 and $p = 0.001$), Savitzky–Golay smoothing[104]; using a first-order polynomial, with a frame number of 5; and total area normalization. Multivariate Curve Resolution–Alternating Least Squares analysis was performed on the spectra using 5 components (based on singular value decomposition of the initial dataset). A maximum of 50 iterations and a convergence limit 0 f 0.1 were used, with initial estimates in the spectrum direction (determined automatically by the built-in SIMPLISMA based algorithm) and a noise level of 10% given in the script. Only nonnegativity constraints were used, both in the spectrum and concentration dimensions. The resulting profiles explained 99.84% of the variation. For classification, k-means clustering was performed within this open-source software, using the resolved spectral profiles and MATLAB's built-in algorithm.

**Clubroot infection rating**. All experiments were performed with the *Plasmodiophora brassicae* single-spore isolate e3[105] and *Arabidopsis thaliana* ecotype Columbia was used as the wild type. Resting spores were extracted by the homogenization of mature clubroot galls (stored at $-20$ °C) of Chinese cabbage, followed by filtration through gauze (25-mm pore width) and two centrifugation steps

($2500 \times g$, 10 min). Fourteen-day-old *Arabidopsis* seedlings, which were cultivated in a controlled environment (23 °C, 16-h light, 100 mmol photons/s/m$^2$) using a compost:sand (9:1 v/v) mixture (pH 5.8), were inoculated by injecting the soil around each plant with 1 mL of a resting spore suspension in Na$_2$HPO$_4$ buffer (pH 5.8), with the spore concentration $10^6$, $10^5$, and $10^4$ spores per mL. Controls were the same age and were treated with Na$_2$HPO$_4$ buffer (pH 5.8) (mock) instead of spore suspension. Disease symptoms were assessed at 28 dai. At least 30 *Arabidopsis* plants were analyzed for each line and treatment. The disease severity was assessed qualitatively on the basis of the infection rate and a disease index as described[50] using the following five-scale classification: 0 (no symptoms), 1 (very small galls mainly on lateral roots and that do not impair the main root), 2 (small galls covering the main root and few lateral roots), 3 (medium to large galls, also including the main root; plant growth might be impaired), and 4 (severe galls on lateral root, main root, or rosette; fine roots completely destroyed; plant growth is reduced). Data are displayed as percentage of plants in the individual disease classes since this gives a more detailed view on the differences. Data presented are means of two independent experiments for the $10^6$ and $10^5$ spore concentrations and one for the $10^4$ spore concentration. The qualitative disease assessment data were analyzed using the Kruskal–Wallis test and by subsequently comparing the mean rank differences as described in ref. [50].

Fresh weight of shoots of wild type, *syac1-3* and *syac1-5* and lines overexpressing *SYAC1ox* were measured 28 days after *P. brassicae* inoculation with control (=buffer) treatment (mock), $10^4$ and $10^5$ spore concentration. For the experiment 30–35 plants per line and treatment were analyzed, one point in the box plot graphs represents an average fresh weight from two to three plants.

**SYAC1 expression analysis after infection by *P. brassicae***. Fourteen-day-old *pSYAC1:GUS* transgenic *Arabidopsis* seedlings were inoculated with the spore concentration $10^6$ as described above and GUS expression detected in roots at 20 and 28 dai. As a positive control 34-day-old plants were treated with 1 µM auxin (NAA) and 10 µM cytokinin (N6-benzyladenine) for 6 h. GUS staining performed as described above (Histochemical and histological analysis).

## Data availability
The authors declare that the data supporting the findings of this study are available within the paper, and its Supplementary Information files. A reporting summary for this Article is available as a Supplementary Information files. The datasets generated and analyzed during the current study are available from the corresponding author upon request. The source data underlying Figs. 1a, b; 2a–f; 3b; 4c; 5a, c, d, e; 6a; and 7d, e, as well as Supplementary Figs. 1a, b; 2b–d, g; 3b, d; 4c, d, e, f; 5d, f, i; 6a; and 7d, are provided as a Source Data file. Transcriptome profile data associated with this study has been deposited in the NCBI Sequence Read Archive under accession number GSE146778. The mass spectrometry proteomics data have been deposited to the ProteomeXchange Consortium via the PRIDE partner repository with the dataset identifier PXD018159 [10.6019/PXD018159].

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

## Acknowledgements

We thank Daria Siekhaus, Jiri Friml and Alexander Johnson for critical reading of the manuscript, Peter Pimpl, Christian Luschnig and Liwen Jiang for sharing published material, Lesia Rodriguez Solovey for technical assistance. This work was supported by the Austrian Science Fund (FWF01_I1774S) to A.H., K.Ö., and E.B., the German Research Foundation (DFG; He3424/6-1 to I.H.), by the People Programme (Marie Curie Actions) of the European Union's Seventh Framework Programme (FP7/2007-2013) under REA grant agreement n° [291734] (to N.C.), by the EU in the framework of the Marie-Curie FP7 COFUND People Programme through the award of an AgreenSkills+ fellowship No. 609398 (to J.S.) and by the Scientific Service Units of IST-Austria through resources provided by the Bioimaging Facility, the Life Science Facility. The IJPB benefits from the support of Saclay Plant Sciences-SPS (ANR-17-EUR-0007).

## Author contributions

A.H. and E.B. conceived the project; A.H. performed most of experiments related to functional and phenotype analyses of SYAC1; C.C. contributed to generation of constructs and transgenic lines; N.C. performed expression analysis of SYAC1 in wild-type and mutant background in response to hormones; K.Ö. performed Co-IP experiments; J. D. carried out transcriptome profiling; L.D. supported by E.S. and A.H. performed Y2H and BIFC analyses; J.C.M. contributed to α-amylase assays; M.G. analyzed cell wall mechanical properties; T.R. and H.S. contributed to co-immunolocalisation and phenotype analyses; I.S. and I.H. designed and carried out SYAC1 expression and phenotype analyses in pollen tubes; R.B. contributed to analyses of SYAC1 interaction with ECH/YIP complex; A.G. and A.H. performed FT-IR measurements; J.S. and G.M. analyzed composition of cell walls; G.P. and G.D.J. identified SYAC1 interactors; F.B and J.L-M. tested tolerance of plants to *Plasmodiophora brassicae* infection. A.H. and E.B. wrote the manuscript.

## Competing interests

The authors declare no competing interests.
