## [Peer Review File · Nature Communications]

Reviewers' comments:

Reviewer #1 (Remarks to the Author):

The Manuscript from Hurný and collaborators describes novel and important results that bridge the gap between hormonal control and cell growth in plants, a stream of processes which remains elusive. Here, the authors provide robust data that show the synergistic action of auxin and cytokinin in triggering expression of SYAC1 (SYnergistic on Auxin and Cytokinin), a gene the authors show to repress cell growth but also organ growth such as apical hook development, hypocotyl elongation and root growth. A set of in vitro and in vivo protein-protein interaction assay convincingly shows that the SYAC1 protein interacts with the TGN-localized protein complex ECH/YIP4a/YIP4b which was previously shown to act in protein secretion of pectin, a major component of the cell wall involved in the flexibility of organ growth. Genetic interaction between SYAC1 and YIP4 genes additionally confirmed that they are acting in one pathway. It is particularly interesting that a biotic stress, the infection of the plant by a protist which relies on an auxin-cytokinin crosstalk, is also dependent on SYAC1 function. The manuscript is firmly grounded on an impressive set of genetics (loss of function by T-DNA and CRISPR), cell biology (rescue of SYAC1-GFP in KO lines, subcellular localization) and biochemistry (protein-protein interaction and cell wall analyses) approaches as well as phenotypic analyses (apical hook defects, hypocotyl elongation and root growth defects). As it stands, the manuscript is really strong and I have only few comments to make.

1- Ethylene is not involved in enhancing SYAC1 expression in roots but gibberellins could act as an auxin-cytokinin master regulator. Does gibberellins application or treatment with gibberellins inhibitor have an effect on SYAC1 expression?

2- A role of SYAC1 is described for apical hook development, hypocotyl growth and root growth. In apical hook, expression of SYAC1 is correlated to the place where auxin accumulates and cell elongation is inhibited. Given other data suggesting that SYAC1 is a negative regulator of ECH/YIP-mediated secretion pathway, the results convincingly argue for a role of SYAC1 as negative regulator of cell elongation where auxin accumulates. However, during hypocotyl phototropism auxin accumulates at the opposite side of the illuminated hypocotyl to trigger cell elongation positively. Does SYAC1 also co-accumulate with auxin during hypocotyl phototropism?

3- The link between auxin/cytokinin and SYAC1 is provided at the gene expression level while localization and interaction data are provided at the protein level. A functional pSYAC1:gSYAC1-GFP protein fusion was produced in this study but was too weak to be detected in untreated condition. Does the pSYAC1:gSYAC1-GFP signal significantly raised upon auxin/cytokinin treatment correlatively to the gene expression?

4- Similarly, co-localization data were performed using an estradiol over-expression system that decelerate root growth. If the pSYAC1:gSYAC1-GFP signal raised upon auxin/cytokinin treatment, does SYAC1-GFP co-localize with anti-ECH or anti-SYP61 upon auxin/cytokinin activation?

5- SYAC1 interacts with ECH/YIP complex. Is the ECH localization altered in SYAC1ox?

Reviewer #2 (Remarks to the Author):

The manuscript by Hurny et al. describes the identification of the novel gene Synergistic on auxin and cytokinin 1 regulated (SYAC1), a gene synergistically regulated at the transcriptional level by auxin and cytokinin, and that encodes a protein of unknown function. Several assays are presented to support a role for SYAC1 in the control of root growth, participation in the secretory

pathway, and altered responses to the pathogen *Plasmodiophora brassicae*. However, there is no conclusion on how this gene integrates auxin and cytokinin responses in plants, and while the pathogen assays do indicate altered responses to pathogens in SYAC1 loss-of-function and gain-of-function lines, there is no evidence that SYAC1 regulates growth-defense tradeoffs. These issues, together with a general lack of details of how experiments were performed, as well as lack of controls for some important experiments, detract from the quality of the manuscript and weaken its central hypothesis. Major weaknesses and points in need of clarification are highlight below:

1) In the beginning of the manuscript, the authors indicate that the novel SYAC1 gene was found through genome-wide transcriptome profiling of Arabidopsis roots, though it is unclear where this information came from such as RNAseq data sets or by other means. There are no details about this transcriptomics study in the manuscript. The authors should provide more details, including what type of transcriptomics (RNAseq?), age and type of plant tissue was used, as well as levels of cytokinin and auxin used, type of mock controls, and timing of tissue harvest after hormone application. This basic information is necessary to assess the importance of SYAC1 up-regulation during hormonal responses. For example, fast up-regulation by both hormones is indicative of a primary transcriptional target. As is stands, there is no information on the manuscript about how fast SYAC1 is up-regulated after hormone treatment. Therefore, I cannot agree with the claims of SYAC1 being a novel target (line 124) of auxin and cytokinin pathways as no direct binding between auxin- or cytokinin-regulated transcription factors were shown to directly bind to the promoter region of the gene, though data does support SYAC1 as a downstream response in these hormonal pathways.

2) Line 126: "throughout the lifespan of Arabidopsis". Differently from what the authors describe, the expression of SYAC1 was not monitored throughout life span of the plant, but only from germination to 4 days old. Further, in 2-, 3- and 4-day-old plants, expression was also observed in the cotyledons, therefore the claim that its expression is limited to plant roots (Line 137) is not warranted. The authors should clarify and show what is the expression pattern of SYAC1 in older plants, including in shoots. This is important for their argument that SYAC1 expression is correlated to "processes involving the control of elongation growth" (Line 138). The establishment of the pSYAC1:GUS and pSYAC1:nlsGFP reporter lines allow for visualization of expression pattern in the roots. However, the authors should include a magnified image of the quiescent center and columella initials in Figure 1B to adequately show SYAC1 expression in these cell types.

3) For the SYAC1 CRISPR/Cas line generated in the study: the genotyping data confirming the mutation, homozygosity and lack of CRISPR/Cas cassette should be presented as supplementary information. Furthermore, it would be beneficial if the authors could demonstrate that the introduced mutation does indeed lead to the expression of a truncated protein.

4) In order to address the function of SYAC1 in Arabidopsis, the authors isolated a T-DNA mutant on this gene, as well as created a CRISPR/Cas mutated line (see also comment 3). Given that the expression of this gene was identified as synergistically regulated by auxin and cytokinin in roots, the authors then proceeded to see whether root growth was altered upon auxin and cytokinin treatment in the mutant lines, in comparison to similarly treated wild type plants. Results from this experiment (figure 2D) showed no difference between wild type and mutants, leading the authors to argue that a gene similar in sequence and in pattern of expression (named here BSYCA1, Figure S2) is redundant to SYAC1, and therefore could make up for a loss of function in SYAC1. While this could be a plausible argument, it would be difficult to argue that this gene would not be redundant during transient hormone induction (Figure 2E). Further, any claims of redundancy can only be determined by double mutant analysis, which was not performed in this study. Furthermore, the increased hypocotyl growth phenotype in the *syac1*, *yip4a*, *yip4b* triple mutant compared to the *yip4a*, *yip4b* double mutant (figure S5A-B) argue against a redundant function for BSYAC1, although to be sure, the authors would have to redo these assays using wild type plants as controls, something that is missing in this experiment (see also comment 7).

5) For the experiments depicted in Figures 2D and 2E, what are the levels of auxin and cytokinin used? Similarly to the point raised above, what is the kinetics of the induction (how fast after hormone application)? There is some information on a 6 hour hormone treatment in the Methods section, but it is unclear whether this refers to figures 2D/E. This information is needed to properly understand the data shown (and once available it should be included in the figure legend, as well as the main text). Also with regards to Figure 2, some colors should be changed in Figures 2E and F as some are difficult to see and therefore interpret completely.

6) The yeast two-hybrid and IP experiments shown in figure 4 place SYAC1 in the YIP complex, which is informative, given the fact that SYAC1 encodes a protein of unknown function. However, the Bi-FC experiments used for interaction validation need further explanation. Were 4-days old root suspension cultures really used for protoplast isolation, or were roots of 4-day-old seedlings used? Suspension cultures are usually kept in culture for longer periods. It would be useful if this could be further explained. Also, more information on the tandem affinity purification protocol needs to be provided. What do "extract and binding" refer to (Line 716), and what were the conditions used?

7) Figure 5A, B, C and E show assays used to determine a function of SYAC1 in the secretory pathways. One major point of concern is the fact that these assays rely mostly on data from overexpressing lines, and lack corroborating data from loss-of-function lines. Given that overexpression may lead to gain-of-function phenotypes that do not properly reflect native function, the reliance on overexpressing lines raises the question on whether a function in the secretory pathway is indeed the real function of SYAC1 in plants. What are the phenotypes in these assays (reduced alpha amylase secretion seen in figure 5A; reduced seed mucilage in figure 5B) for the *syac1* mutants? The data in figure 5D indicates that loss-of-function *syac1* plants do not have a phenotype, at least for that assay. Furthermore, some of these assays (seed mucilage assay on 5B; pectin secretion on pollen tubes in tobacco (figure S4B)) also rely on overexpression of SYAC1 in tissues that are different from the tissue where SYAC1 was initially determined by the authors and where a function for this gene was initially proposed (hypocotyl and root cells; figure 1 D-H; see Line 143; pathogen assays in figure 6, see Line 364-367). The main piece of data indicating that SYAC1 has a role in the secretory pathway is shown in figure S5A-B, however the phenotype seems to be subtle, and as noted earlier, those assays are missing wild type plants as controls. Also, photos of seedlings grown on media are difficult to see clearly and could be improved with better lighting to aid reader interpretation.

8) Also, Figure 5B shows reduced mucilage in the SYAC1-GFPox transgenic seed. For comparison, it would be interesting to include similar data for the *syac1-5* line and if there is an increase in mucilage from mature seeds. The total number of seeds observed for the assay in 5B and S4A are not noted in the text, figure legends, or Materials sections. Use of atomic force microscopy (AFM) is indicated in the text but parameters are not included in the Methods section. Analysis of the apparent Young modulus, or the elasticity of SYAC1 transgenic plants should be expanded upon in the text (i.e., reduced modulus indicates reduced hypocotyl elasticity and therefore more susceptible to shearing and breaking) allowing for more obvious connection to biological relevance.

9) For the pathogen assays in figure 6: statistical analyses are needed, especially given the fact that the disease rating is not really quantitative (i.e., it is based on a visual score, rather than on measuring of plant lesions or pathogen growth). The legend on figure 6 needs to be updated as it currently is listed as figure 7. Rosette development indicated in figure S6 is not clear in the presently displayed images as individual rosettes cannot be identified from these images.

10) Also with regards to the pathogen results: The altered phenotypes of SYAC1 loss-of-function and gain-of-function lines in plant growth/secretion (figures 1-5) and pathogen responses (figure 6) led the authors to argue that SYAC1 regulates auxin and cytokinin-regulated growth-defense tradeoffs. This argument does not really stand. There are several examples in the literature of auxin and cytokinin mutants with altered responses to pathogens, and most of them also have

plant growth phenotypes, but not a role in growth-defense tradeoffs. A role in growth-defense tradeoffs would require SYAC1 to be involved in the regulation of the negative effect on plant growth that is observed during activated defense, something that has not been demonstrated in this manuscript.

11) Finally, there are some English mistakes that need to be corrected throughout the manuscript. In the abstract, the authors should consider changing "non-locomotive" to "sessile" or "non-mobile".

Reviewer #3 (Remarks to the Author):

Hurny et al. identified a novel gene SYNERGISTIC ON AUXIN CYTOKININ1 (SYAC1) as one of those synergistically upregulated by a simultaneous application of plant hormones auxin and cytokinin in Arabidopsis roots. The authors also identified interacting partners of SYAC1, and showed that the auxin/cytokinin-inducible SYAC1 modulates secretory pathway for cell wall components in SYAC1 overexpressing plants. SYAC1 was further indicated as crucial for cell growth, as well as pathogen resistance in the Arabidopsis roots.

I think all data are clearly presented and support their conclusions. Even though the authors did not clarify in detail how SYAC1 regulates the secretory pathway, and how the auxin/cytokinin signalling converges on it, the manuscript describes a novel component where the two major plant hormones take part in modulating plant growth. This manuscript advances our knowledge and provides new perspectives on phytohormone-mediated regulation of plant development.

I suggest the authors to consider the following:

- Treating plants with high concentrations of two hormones (auxin and cytokinin) for 6h may result in artefacts, in addition to the actual hormone responses. To investigate whether SYAC1 is a direct target of the two phytohormone signalling pathways, the authors may check the expression level of SYAC1 in hormone signalling mutants, such as *ahk234* or inducible *bdl*-expressing plants, following the hormone treatment.
- The current data convincingly suggest that SYAC1 is involved in plant cell growth, however it would be further supportive if the authors actually measured the cell size of SYAC1 overexpressing plants.
- Description about biochemical properties of SYAC1 would be informative for readers if available. (The authors may describe what kind of domain(s) it has, how big the protein is, and predicted roles based on its properties, and so on in the first section of the results.)
- Figure 2 legend, 4th row: I think the wording "...from O to M phase..." should read "...from M to O phase...". Please check.
- Figure 6 legend: This is indicated as Figure 7 currently, please correct. Also, the number of replicates is not specified, and info on the statistics is incomplete.

Reviewer #4 (Remarks to the Author):

The manuscript entitled "Synergistic on auxin and cytokinin 1 integrates growth and pathogen defense" describes the identification and characterization of an auxin and cytokinin cross-talk component, SYNERGISTIC ON AUXIN AND CYTOKININ1 (SYAC1). Using transcriptomic analysis, the authors identified a gene specifically up-regulated in roots exposed simultaneously to cytokinin

and auxin. Transcriptional SYAC1 activity/levels were low or not detected in untreated roots becoming active by the synergic action of auxin and cytokinins. In embryonic hypocotyl and cotyledons, the transcriptional reporter is endogenously active and remained active in seedlings cotyledons hypocotyl. Interestingly, the expression in etiolated plants is concentrated in short cells at the inner (concave) side of the apical hook, whereas no signal was detected in elongating cells at outer side of the hook. This pattern of expression led the authors to propose that SYAC1 might be regulating cell elongation. Additional evidence supporting such theory was collected using loss of function mutants and over-expression of the protein.

The authors describe the protein SYAC1 to be located at the Golgi and TGN/endosomal organelles. Additionally, SYAC1 was shown to be interacting with ECHIDA/Yip complex by yeast two hybrid, co-immunoprecipitation and by BIFC (transient expression in protoplasts). The ECHIDA/Yip complex has been previously described to be involved in secretion specially in elongating cells. Therefore, the authors propose that SYAC1 is acting as a negative regulator of the ECHIDA/Yip complex activity and therefore restricting cell elongation and growth. Finally, the authors propose that a protist infection that increases auxin-cytokinin levels in roots activates the expression of SYAC1 inhibiting secretion and facilitating infection.

General comments:

1.Regarding figure legends, they need to be more precise. In many cases the legends are confusing (see minor issues). Is also necessary to include in the legends: how many experiments are represented in each graph? The values are product of a single experiment but representative of the replicates? How many times the experiment was replicated? Is difficult to interpret the figures without this information that in many cases is not properly detailed in materials and methods either. Additionally, standard error can sometimes hide big differences between replicates, I recommend replacing the closed bars graphs by individual plot values, like the ones in the open boxes represented in figure 2E.

2.Regarding material and methods, is very frustrating as a reader finding statements like this one: the experiment was performed "as previously described" "reference". There are many protocols absent in the manuscript referenced this way, I strongly recommend the authors to include, at least, a brief description of the protocol.

Major issues:

3.SYAC1A was shown to co-localize with RAB5-like protein markers (RHAI and ARA7). ARA7 is present in late endosomes/PVC but it can be also found at the TGN (Singh, Manoj K., et al. Current Biology 24.12 (2014): 1383-1389.). Therefore, the authors should use additional PVC markers to provide supplementary evidence to confirm PVC localization. Otherwise, is easy to propose wrong conclusions based on miss interpretation of organelle localization.

4.Several conclusions are based in indirect evidence weakening the manuscript:

4.1Evidence is presented suggesting that decrease alpha-amylase activity in the growing media of SYAC1 overexpressing protoplasts indicates a decrease in protein secretion. Nevertheless, it could also be a problem in transformation efficiency, protein synthesis or protein folding that will produce a concomitant decrease alpha-amylase activity in the media. As this is a transient assay, additional controls should be included to quantify total protein and the amount of protein secreted to the media. Additionally, evidence supporting SYAC1 protein over-expression in protoplasts were the alpha-amylase activity is decreased should also be provided (for example western blot).

4.2Additional information supporting the role of SYAC1 inhibiting of secretion was proposed by analyzing SYAC1ox plants. Imbibed seeds exhibit decreased mucilage secretion. Additionally, lower amounts of galacturonic acid can be extracted from cell wall in seedlings. All these evidences suggest a problem in cell wall components secretion, but it represents indirect evidence. The authors need to probe retention of cell wall components in the secretory system. Is important to know if the retention is at the level of the Golgi/TGN consistently with SYAC1 localization. Without this information is difficult to discard a problem in polysaccharides synthesis that also takes place at the Golgi apparatus. Additional evidence on protein or polysaccharides retention at the Golgi or TGN will considerably strengthen the hypothesis and mechanisms proposed.

4.3The authors suggest that SYAC1 over-expression in protoplasts increased secretion of Amy-Spo

indicating an impairment on vacuolar transport and, therefore a role of SYAC1 in vacuolar transport. This evidence is highly indirect, additional experiments in planta are necessary to support this claim. Again, evidence supporting SYAC1 protein over-expression in protoplasts should also be provided as this is a transient assay.

4.4 Transient over-expression of LAT52:SYAC1 in pollen tubes is suggested to decrease pectin deposition at the pollen tube apex. Nevertheless, supplementary figure 4D is not clear. The images seem to be coded in black and white making difficult to visualize the red staining. Essentially it looks like there is more ruthenium red signal in LAT52:SYAC1 pollen tube than in the control. Providing better quality images and including immunolocalizations using JIM5/JIM7 antibodies recognizing de-esterified or partially de-esterified pectin's is also advisable. It is also necessary to provide evidence supporting the over-expression of SYAC1 protein in this transient system.

4.5 PIN2 and PIN1 accumulation in root cells is not affected by SYAC1 over-expression (stable). How about during gravitropic response? This situation could be similar to the hypocotyl hook as the cells on the upper part are elongating in opposition the cells on the lower part of the bending root. Is important to determine SYAC1 function in planta and not only in transient experiments to support the conclusions of the paper.

4.6 Auxin+cytokinins treatments increase SYAC1 promoter activity in the root. Nevertheless, the authors do not show a decrease on protein or polysaccharide secretion in such conditions or a concomitant decrease in cell elongation. It is also necessary to analyze if Auxin+cytokinins treatments on the loss of function SYAC1 mutants to show opposite phenotypes. I realize that the suggested experiments may represent an oversimplification of the hormone cross-talk pathway but is necessary to link the activity of the protein with the increased level of expression to be able to support the conclusions proposed in the manuscript. The authors should provide stronger evidence of such link even by providing a different set of experiments that the authors consider more suitable.

4.7 Plasmodio phorabraceae infection increases auxin-cytokinin levels in roots. The manuscript shows that SYAC1 overexpressing plants are sensitive to the infection and, on the contrary, the mutants are more resistant than wild type plants. The authors propose that increased auxin-cytokinin levels in infected roots will increase the expression of SYAC1. Nevertheless, no evidence is provided indicating that SYAC1 mRNA and protein levels are indeed increased in wild type infected plants. This data will help supporting the proposed hypothesis and will increase the relevance of the infection assay.

Minor issues:

1. GUS staining in roots highlights the expression pattern of the transcriptional reporter in the QC and elongation/differentiation zone. Nevertheless, even do they have the promoter fuse to a nuclear localization signal tag with GFP, the authors fail to show a more precise cellular pattern of expression that will certainly be more informative. Having that information will provide the readers a more precise idea on how restricted in the expression at the QC/columella, elongating cells vs dividing cells among others.

2. qPCR analysis showing the remaining SYAC1 mRNA levels in the insertional mutants is only provided for *syac1-1* and *syac1-3* lines. What about *syac1-2*, *syac1-4* and the CRISP line *syac1-5*?

3. The over-expression lines (SYAC1-HAox, HA-SYAC1ox, SYAC1-GFPox and GFP-SYAC1ox) were shown to have higher mRNA level than wild type plants. However, the concomitant increase in protein levels is not shown. Western blots are recommended.

4. The authors propose that in *syac1-3* and *syac1-5* mutant backgrounds the protein BSYAC1 might be compensating for the SYAC1 loss of function. Is either the BSYAC1 mRNA or protein levels altered in the *syac1-3* and *syac1-5* mutant backgrounds?

5. The co-immunoprecipitation analysis in Figure 4C shows in line 1 the control (that I assume is the untransformed protoplast, needs to be stated in the legend), the ECG-HA input, GFP (alone?) and, ECH-HA and protoplast transfected with both SYAC1-GFP and ECH-HA. The Co-IP was performed using anti GFP antibody. However, background signal observed in protoplast extracts expressing only ECH-HA (and no SYAC1-GFP). Is the signal due to poor washing or unspecific background? It needs to be address in the text. Due to that unspecific signal and the transient nature of the experiment is necessary to include the ECH-HA protein level in the input when both proteins, SYAC1-GFP+ECH-HA, are expressed in the protoplast. In this way, is possible

to unequivocally distinguish co-IP ECH-HA together with SYAC1-GFP from unspecific binding due to stronger expression of ECH-HA by itself. Is also remarkable that ECH-HA detected after the immunoprecipitation presents a higher molecular weight compared to the input signal. Is my understanding from materials and methods that this is an SDS-PAGE so, why there is a molecular weight shift? The text needs to address this.

6. Supplemental figure 4D needs a clearer legend, is difficult for this reviewer to understand what the authors are intending to show.

Response to the Reviewers' comments:

We would like thank all reviewers for their insightful comments and suggestions.

Reviewer #1 (Remarks to the Author):

The Manuscript from Hurný and collaborators describes novel and important results that bridge the gap between hormonal control and cell growth in plants, a stream of processes which remains elusive. Here, the authors provide robust data that show the synergistic action of auxin and cytokinin in triggering expression of SYAC1 (SYnergistic on Auxin and Cytokinin), a gene the authors show to repress cell growth but also organ growth such as apical hook development, hypocotyl elongation and root growth. A set of in vitro and in vivo protein-protein interaction assay convincingly shows that the SYAC1 protein interacts with the TGN-localized protein complex ECH/YIP4a/YIP4b which was previously shown to act in protein secretion of pectin, a major component of the cell wall involved in the flexibility of organ growth. Genetic interaction between SYAC1 and YIP4 genes additionally confirmed that they are acting in one pathway. It is particularly interesting that a biotic stress, the infection of the plant by a protist which relies on an auxin-cytokinin crosstalk, is also dependent on SYAC1 function. The manuscript is firmly grounded on an impressive set of genetics (loss of function by T-DNA and CRISPR), cell biology (rescue of SYAC1-GFP in KO lines, subcellular localization) and biochemistry (protein-protein interaction and cell wall analyses) approaches as well as phenotypic analyses (apical hook defects, hypocotyl elongation and root growth defects). As it stands, the manuscript is really strong and I have only few comments to make.

1- Ethylene is not involved in enhancing SYAC1 expression in roots but gibberellins could act as an auxin-cytokinin master regulator. Does gibberellins application or treatment with gibberellins inhibitor have an effect on SYAC1 expression?

Response: Motivated by the comment we examined impact of other plant hormones including gibberellins methyljasmonate, abscisic acid and brassinosteroids (Fig. S1h). None of the tested hormones stimulated pSYAC1:GUS expression after six hours of treatment, indicating that cytokinin and auxin are major hormonal regulators of the SYAC1 expression.

2- A role of SYAC1 is described for apical hook development, hypocotyl growth and root growth. In apical hook, expression of SYAC1 is correlated to the place where auxin accumulates and cell elongation is inhibited. Given other data suggesting that SYAC1 is a negative regulator of ECH/YIP-mediated secretion pathway, the results convincingly argue for a role of SYAC1 as negative regulator of cell elongation where auxin accumulates. However, during hypocotyl phototropism auxin accumulates at the opposite side of the illuminated hypocotyl to trigger cell elongation positively. Does SYAC1 also co-accumulates with auxin during hypocotyl phototropism?

Response: We examined expression of SYAC1 in hypocotyls after photo-stimulation by unilateral light as previously described in (Ding et al., 2011). 3 day-old etiolated seedlings after 24 hours exposure to long-day conditions were illuminated by low-intensity unilateral white light for 2, 6 and 24h. As expected, in seedlings expressing the auxin sensitive reporter DR5:GUS, which was used as a positive control, we detected enhancement of auxin response in the outer, non-illuminated side of hypocotyls already 2 hours after photo-stimulus. Unlike the DR5:GUS reporter, no expression of pSYAC1:GUS could be detected on either side of photo-stimulated hypocotyls. Since SYAC1 expression is highly dependent on auxin – cytokinin balance, it seems that local increase of auxin activity in hypocotyls when not accompanied with simultaneous activation of the cytokinin pathway might not be sufficient to trigger SYAC1 expression. Similarly to hypocotyls, in gravi-stimulated roots a local increase of auxin activity does not activate SYAC1 expression (see also response to the reviewer 4 addressing SYAC1 expression in gravi-stimulated roots). As these results do not reveal any significant contribution of pSYAC1:GUS in photo- or root gravity responses, we decided not to expand further the manuscript by this data set.

3- The link between auxin/cytokinin and SYAC1 is provided at the gene expression level while localization and interaction data are provided at the protein level. A functional pSYAC1:gSYAC1-GFP protein fusion was produced in this study but was too weak to be detected in untreated condition. Does the pSYAC1:gSYAC1-GFP signal significantly raised upon auxin/cytokinin treatment correlatively to the

gene

expression?

Response: Expression of the pSYAC1:gSYAC1-GFP could be detected after 6 hours treatment with 10 μ M cytokinin and 1 μ M auxin and we included these data in the revised version of the manuscript (Fig. S1e).

4- Similarly, co-localization data were performed using an estradiol over-expression system that decelerate root growth. If the pSYAC1:gSYAC1-GFP signal raised upon auxin/cytokinin treatment, does SYAC1-GFP co-localize with anti-ECH or anti-SYP61 upon auxin/cytokinin activation?

Response: We agree with the reviewer that co-localization of SYAC1 with ECH and SYP61 would be relevant experiment. Unfortunately, despite our repeated efforts signal of SYAC1-GFP driven by endogenous promoter was not sufficiently strong to perform reliable quantifications, although the calculated Pearson correlation coefficient supports co-localization with both reporters. Therefore, we concentrated on the inducible expression system, which enabled us to induce SYAC1-GFP within six hours and thus to minimize potential feedback of SYAC1 on subcellular localization and interaction with ECH.

5- SYAC1 interacts with ECH/YIP complex. Is the ECH localization altered in SYAC1ox?

Response: Unfortunately, we were not able to perform these experiments. The main obstacle is a lack of suitable antibodies that could be used for co-localization in planta. All available ABs for subcellular markers and ECH are anti-rabbit and therefore do not allow to perform proposed co-immunostaining.

Reviewer #2 (Remarks to the Author):

The manuscript by Hurny et al. describes the identification of the novel gene Synergistic on auxin and cytokinin 1 regulated (SYAC1), a gene synergistically regulated at the transcriptional level by auxin and cytokinin, and that encodes a protein of unknown function. Several assays are presented to support a role for SYAC1 in the control of root growth, participation in the secretory pathway, and altered responses to the pathogen *Plasmodiophora brassicae*. However, there is no conclusion on how this gene integrates auxin and cytokinin responses in plants, and while the pathogen assays do indicate altered responses to pathogens in SYAC1 loss-of-function and gain-of-function lines, there is no evidence that SYAC1 regulates growth-defense tradeoffs. These issues, together with a general lack of details of how experiments were performed, as well as lack of controls for some important experiments, detract from the quality of the manuscript and weaken its central hypothesis. Major weaknesses and points in need of clarification are highlight below:

1) In the beginning of the manuscript, the authors indicate that the novel SYAC1 gene was found through genome-wide transcriptome profiling of Arabidopsis roots, though it is unclear where this information came from such as RNAseq data sets or by other means. There are no details about this transcriptomics study in the manuscript. The authors should provide more details, including what type of transcriptomics (RNAseq?), age and type of plant tissue was used, as well as levels of cytokinin and auxin used, type of mock controls, and timing of tissue harvest after hormone application. This basic information is necessary to assess the importance of SYAC1 up-regulation during hormonal responses. For example, fast up-regulation by both hormones is indicative of a primary transcriptional target. As it stands, there is no information on the manuscript about how fast SYAC1 is up-regulated after hormone treatment. Therefore, I cannot agree with the claims of SYAC1 being a novel target (line 124) of auxin and cytokinin pathways as no direct binding between auxin- or cytokinin-regulated transcription factors were shown to directly bind to the promoter region of the gene, though data does support SYAC1 as a downstream response in these hormonal pathways.

Response: We apologize for lack of the relevant information. In the revised manuscript we extended material and methods to provide details about the transcriptome profiling. The SYAC1 was recovered from transcriptome profiling performed on 5-day-old seedlings treated with 1 μ M NAA and 10 μ M cytokinin for 3 hours. As originally focus of the project was on the genes involved in root branching the transcriptome profiling was performed on pericycle tissue after sorting cells expressing GFP reporter in J1201 reporter (Laplaze et al., 2005). Subsequent analysis using reporter constructs revealed that SYAC1 expression is broader and the most prominent increase of the gene expression is at the root transition and elongation zones that include also pericycle cells. In addition, in a recent study by Ristova et al., 2016 focused on the interaction network of transcriptomic and phenotypic responses of Arabidopsis roots to nitrogen and hormones, the transcript corresponding to SYAC1 was among those synergistically upregulated by roughly equivalent hormone treatments; (applying 500nM IAA and 500 nM kinetin as a

cytokinin for 4 hours) (Krouk, personal communication); although this was not specifically recognized in the study by Ristova et al., 2016.

We agree with the reviewer, that three hours of hormonal treatment is not sufficiently short time to exclude some additional factors downstream of auxin and cytokinin, which might be involved in control of SYAC1 expression. To address this point we performed a time course analysis of SYAC1 transcription. We found that already 30 min after application of auxin and cytokinin SYAC1 expression in roots significantly increases when compared to mock treated roots (Fig. S1b revised manuscript). Furthermore, we tested contribution of auxin and cytokinin receptors to synergistic regulation of SYAC1 transcription. We found that lack of either cytokinin or auxin perception mediated through CRE1-12/AHK4, AHK3 and TIR1, AFB2, respectively, significantly attenuates synergistic regulation of the SYAC1 transcription by both hormones (Fig. S1c). We believe that these results further support a role of both cytokinin and auxin pathways in control of the SYAC1 expression. On the other hand, we admit that we have not identified transcription factor(s), which downstream of auxin and cytokinin receptors directly control SYAC1 transcription. Therefore, in the manuscript we avoided to use term “direct” target of auxin and cytokinin pathways. However, we believe that observations such as that SYAC1 transcription is triggered by simultaneous treatment with auxin and cytokinin and it is dependent on the activity of respective receptors, support conclusion about SYAC1 as a novel common target of auxin and cytokinin acting in roots.

2) Line 126: “throughout the lifespan of Arabidopsis”. Differently from what the authors describe, the expression of SYAC1 was not monitored throughout life span of the plant, but only from germination to 4 days old. Further, in 2-, 3- and 4-day-old plants, expression was also observed in the cotyledons, therefore the claim that its expression is limited to plant roots (Line 137) is not warranted. The authors should clarify and show what is the expression pattern of SYAC1 in older plants, including in shoots. This is important for their argument that SYAC1 expression is correlated to “processes involving the control of elongation growth” (Line 138). The establishment of the pSYAC1:GUS and pSYAC1:nlGFP reporter lines allow for visualization of expression pattern in the roots. However, the authors should include a magnified image of the quiescent center and columella initials in Figure 1B to adequately show SYAC1 expression in these cell types.

Response: We agree that interpretation of the expression analysis in course of seedling germination as a “life span” is overstated. We apologize for the inaccuracy, which we corrected in the revised manuscript. As it concerns SYAC1 expression in older plants we examined pSYAC1:GUS reporter activity in 3 and 8 week-old plants. pSYAC1:GUS in 3 week-old seedlings was under limit of detection. In adult Arabidopsis plants, we observed GUS staining in the upper part of stem under florescence, and at the abscission zone of siliques (Fig. 1h, S1i,j). Expression pattern of SYAC1 in Arabidopsis stem largely complies with recent study correlating growth kinetics and pectin levels along the stem. It has been shown that the stem segment in direct proximity to florescence exhibits fastest growth and displays high levels of pectins, when compared to segments more distant from the apex with slower growth kinetics and reduced levels of pectin (Phyo et al., 2017). Although SYAC1 expression in siliques does not directly correlate with

elongation growth, it is in line with proposed SYAC1 role in regulation of cell wall composition including pectin levels. It has been reported that during abscission of floral organs modulation of pectins is central to cell wall loosening and cell separation (Daher and Braybrook, 2015; González-Carranza et al., 2007; Lashbrook and Cai, 2008; Ogawa et al., 2009).

We would like to point to misreading of the line 137. We do not claim that “its expression is limited to plant roots”, vice versa the sentence reads “SYAC1 function seems not to be limited to plant roots...” .

Finally, we included magnified images of root tips to improve presentation of SYAC1 expression pattern at the root tip (Fig. S1d).

3) For the SYAC1 CRISPR/Cas line generated in the study: the genotyping data confirming the mutation, homozygosity and lack of CRISPR/Cas cassette should be presented as supplementary information. Furthermore, it would be beneficial if the authors could demonstrate that the introduced mutation does indeed lead to the expression of a truncated protein.

Response: As suggested by the reviewer, in the revised manuscript sequencing data confirming mutation and homozygosity were included in Fig. S2a. Lack of CRISPR/Cas cassette was confirmed by loss of resistance to BASTA, and the relevant information was added to Material and Method paragraph. As for the detection of truncated protein, we agree that this would be beneficial, however it would require generation of ABs specific to N-terminal part of the SYAC1 protein, which unfortunately we do not have available.

4) In order to address the function of SYAC1 in Arabidopsis, the authors isolated a T-DNA mutant on this gene, as well as created a CRISPR/Cas mutated line (see also comment 3). Given that the expression of this gene was identified as synergistically regulated by auxin and cytokinin in roots, the authors then proceeded to see whether root growth was altered upon auxin and cytokinin treatment in the mutant lines, in comparison to similarly treated wild type plants. Results from this experiment (figure 2D) showed no difference between wild type and mutants, leading the authors to argue that a gene similar in sequence and in pattern of expression (named here BSYCA1, Figure S2) is redundant to SYAC1, and therefore could make up for a loss of function in SYAC1. While this could be a plausible argument, it would be difficult to argue that this gene would not be redundant during transient hormone induction (Figure 2E). Further, any claims of redundancy can only be determined by double mutant analysis, which was not performed in this study. Furthermore, the increased hypocotyl growth phenotype in the *syac1*, *yip4a*, *yip4b* triple mutant compared to the *yip4a*, *yip4b* double mutant (figure S5A-B) argue against a redundant function for BSYAC1, although to be sure, the authors would have to redo these assays using wild type plants as controls, something that is missing in this experiment (see also comment 7).

*Response: Root phenotype of the loss of *syac1* mutant phenotype is indeed very weak, which we believe might be caused by redundant activity of other SYAC1 homologs. As the homologous genes are located in*

a cluster at the chromosome 1, analyses of double syac1, bsyac1 or other multiple mutant combinations were not possible by crossing mutant alleles.

As it concerns the hypocotyl phenotype of syac1 and a partial recovery of the yip4a, yip4b hypocotyl growth by reduction of the syac1 activity, we would like to mention, that when compared to roots, in hypocotyls the expression of SYAC1 is higher and does not require addition of hormones. This might explain why in the syac1 mutant hypocotyls display more pronounced growth alterations than roots. In the previous version of the manuscript we were focused on presenting the recovery of hypocotyl elongation by reduced activity of SYAC1 in yip4a, yip4b mutant. However, in the original experiments, wild-type control generated by outcrossing was included and it was now added in the revised version of manuscript (Fig. S6a).

5) For the experiments depicted in Figures 2D and 2E, what are the levels of auxin and cytokinin used? Similarly to the point raised above, what is the kinetics of the induction (how fast after hormone application)? There is some information on a 6 hour hormone treatment in the Methods section, but it is unclear whether this refers to figures 2D/E. This information is needed to properly understand the data shown (and once available it should be included in the figure legend, as well as the main text). Also with regards to Figure 2, some colors should be changed in Figures 2E and F as some are difficult to see and therefore interpret completely.

Response: We appreciate these criticisms raised by the reviewer. We performed time course analysis of the SYAC1 induction and show that its transcription in roots increases significantly already 30 minutes after hormonal treatment when compared to untreated control (Fig.S1b)

We also updated legends of all figures including information about hormone concentrations and modified colors of graphs. We hope that introduced changes are sufficiently improving clarity of presented results.

6) The yeast two-hybrid and IP experiments shown in figure 4 place SYAC1 in the YIP complex, which is informative, given the fact that SYAC1 encodes a protein of unknown function. However, the Bi-FC experiments used for interaction validation need further explanation. Were 4-days old root suspension cultures really used for protoplast isolation, or were roots of 4-day-old seedlings used? Suspension cultures are usually kept in culture for longer periods. It would be useful if this could be further explained. Also, more information on the tandem affinity purification protocol needs to be provided. What do “extract and binding” refer to (Line 716), and what were the conditions used?

Response: The suspension culture itself is of the root origin and is weekly subcultured for some years already. For protoplast isolation and transient transfection we used 4 day-old subculture. We modified original text in methods section to make this point clearer.

Tandem affinity purification protocol was extended to provide information that is more detailed.

7) Figure 5A, B, C and E show assays used to determine a function of SYAC1 in the secretory pathways. One major point of concern is the fact that these assays rely mostly on data from overexpressing lines, and lack corroborating data from loss-of-function lines. Given that overexpression may lead to gain-of-function phenotypes that do not properly reflect native function, the reliance on overexpressing lines raises the question on whether a function in the secretory pathway is indeed the real function of SYAC1 in plants. What are the phenotypes in these assays (reduced alpha amylase secretion seen in figure 5A; reduced seed mucilage in figure 5B) for the *syac1* mutants? The data in figure 5D indicates that loss-of-function *syac1* plants do not have a phenotype, at least for that assay. Furthermore, some of these assays (seed mucilage assay on 5B; pectin secretion on pollen tubes in tobacco (figure S4B)) also rely on overexpression of SYAC1 in tissues that are different from the tissue where SYAC1 was initially determined by the authors and where a function for this gene was initially proposed (hypocotyl and root cells; figure 1 D-H; see Line 143; pathogen assays in figure 6, see Line 364-367). The main piece of data indicating that SYAC1 has a role in the secretory pathway is shown in figure S5A-B, however the phenotype seems to be subtle, and as noted earlier, those assays are missing wild type plants as controls. Also, photos of seedlings grown on media are difficult to see clearly and could be improved with better lighting to aid reader interpretation.

Response: We agree with the reviewer, that phenotypes of loss-of function syac1 mutant alleles are relatively weak, however reproducible and tested using independent alleles. We would like to stress, that expression of the SYAC1 is spatio-temporally tightly controlled and limited to some specific developmental events. Therefore, the phenotype of syac1 mutant might appear rather weak, if compared for example to mutant in ECHIDNA, which is constitutively expressed (Gendre et al., 2011). In support of that, in processes where SYAC1 endogenous expression is stronger such as the apical hook and hypocotyls, syac1 mutant alleles are significantly affected when compared to wild-type. Furthermore, in the revised manuscript we show that reduced sensitivity of both syac1 mutant alleles to Plasmodiophora correlates with enhanced expression of SYAC1 in infected roots (Fig S7a,b).

In addition, enhanced SYAC1 activity results in “reversed” phenotype alterations e.g. lack of the apical hook formation, reduced hypocotyl expansion, or enhanced sensitivity to Plasmodiophora when compared to these observed in syac1 mutant alleles.

Unfortunately the assays that would allow us to detect such a local changes in secretion are technically highly challenging and currently not sufficiently sensitive. Although we put significant efforts to detect local alterations in cell wall composition and secretion in loss of function mutants, tissues such as apical hooks are less suitable for assays like FT-IR, or pectin analyses.

As concerns α -amylase assays, this is a transient technique designed to test activity of protein expressed in protoplasts, therefore we were not able to apply it to examine impact of the SYAC1 loss of function on α -amylase secretion.

We thank the reviewer for bringing to our attention poor quality of some images, which were replaced (Fig. S6b).

8) Also, Figure 5B shows reduced mucilage in the SYAC1-GFPox transgenic seed. For comparison, it would be interesting to include similar data for the syac1-5 line and if there is an increase in mucilage from mature seeds. The total number of seeds observed for the assay in 5B and S4A are not noted in the text, figure legends, or Materials sections. Use of atomic force microscopy (AFM) is indicated in the text but parameters are not included in the Methods section. Analysis of the apparent Young modulus, or the elasticity of SYAC1 transgenic plants should be expanded upon in the text (i.e., reduced modulus indicates reduced hypocotyl elasticity and therefore more susceptible to shearing and breaking) allowing for more obvious connection to biological relevance.

Response: In the revised manuscript, representative images of mucilage release from seed coat of syac1 mutant alleles were included (Fig. 5b). However, in syac1-3 or syac1-5 alleles no dramatic changes in mucilage secretion could be obtained presumably due to low expression of SYAC1 in seed coat epidermal cells (Fig. S4b). Detailed information about robustness of the assay were added in Material and Methods. Briefly, approximately 100 seeds from each genotype were stained with ruthenium red and 30 representative images were recorded. We appreciate the reviewer suggestion to discuss more in detail mechanical properties of cell walls in SYAC1 transgenic lines in context of its impact on pectin secretion and cell expansion. In addition to expanding discussion, we have also included more detailed explanations of AFM experiments in Material and Methods.

9) For the pathogen assays in figure 6: statistical analyses are needed, especially given the fact that the disease rating is not really quantitative (i.e., it is based on a visual score, rather than on measuring of plant lesions or pathogen growth). The legend on figure 6 needs to be updated as it currently is listed as figure 7. Rosette development indicated in figure S6 is not clear in the presently displayed images as individual rosettes cannot be identified from these images.

Response: For the statistical analysis of pathogen assay results, we followed the conventions of the field and included analyses and controls in the way it can be found in reports published by others. Also the disease index is an accepted method in the clubroot community, since unlike infection rate it determines the severity of the disease. In material and methods we referred to all relevant articles where the statistics applied in assays adopted in this study has been described (Siemens et al., 2002).

Comment about rosette:

- 1. The way how the plants are grown on trays has been published and is standardly used in different laboratories studying P. brassicae - host interaction.*
- 2. The size and color of the rosette and also leaf number are different, so we did not count the rosette developmental stages, rather only the number of plants that are in the rosette stage at any developmental stage have been counted.*
- 3. We believe that presenting one rosette picture would actually bias the results, because it would be an individual selection. Indeed, the overall picture of one tray covers more the diversity, but also the susceptibility of the upper plant part. We do not aim at specific developmental rosette*

stages in terms of leaf number (see point 1). To avoid misinterpretation we amended description of the rosette phenotype.

Number of replicates: This was given in the methods; however, in the revised manuscript the missing information was added to all figure legends when relevant.

We apologize for inaccuracy in listing figure number, this was amended accordingly.

10) Also with regards to the pathogen results: The altered phenotypes of SYAC1 loss-of-function and gain-of-function lines in plant growth/secretion (figures 1-5) and pathogen responses (figure 6) led the authors to argue that SYAC1 regulates auxin and cytokinin-regulated growth-defense tradeoffs. This argument does not really stand. There are several examples in the literature of auxin and cytokinin mutants with altered responses to pathogens, and most of them also have plant growth phenotypes, but not a role in growth-defense tradeoffs. A role in growth-defense tradeoffs would require SYAC1 to be involved in the regulation of the negative effect on plant growth that is observed during activated defense, something that has not been demonstrated in this manuscript.

Response: To examine a role of SYAC1 in pathogen response was motivated by the intriguing finding that expression of this gene in roots is strongly auxin-cytokinin dependent and the known function of these hormones in plant – microbe interaction. We believe, that altered sensitivity of syac1 and SYAC1ox plants to infection by Plasmodiophora supports a potential contribution of SYAC1 to plant responses to microbes that interact with plants through modulation of auxin-cytokinin balance. To gain indications for growth- defense tradeoff responses we analyzed fresh weight of plants 28 days after infection. Fresh weight was the same for wild type in all conditions. We found, however, that fresh weight of SYAC1ox plants at 10^v4 and 10^v5 spore concentrations was reduced more when compared to wild type. In contrast, infected plants of syac1-3 and syac1-5 were less affected than the overexpressors, but similar to wild type (Fig. S7d). We agree that detailed understanding of SYAC1 in pathogen response requires further investigation, including testing plants with modulated SYAC1 expression, susceptibility to other microbes, or activity of defense pathways.

11) Finally, there are some English mistakes that need to be corrected throughout the manuscript. In the abstract, the authors should consider changing “non-locomotive” to “sessile” or “non-mobile”.

Response: We thank the reviewer for bringing it to our attention. The English editing was done.

Reviewer #3 (Remarks to the Author):

Hurny et al. identified a novel gene SYNERGISTIC ON AUXIN CYTOKININ1 (SYAC1) as one of those synergistically upregulated by a simultaneous application of plant hormones auxin and cytokinin in Arabidopsis roots. The authors also identified interacting partners of SYAC1, and showed that the auxin/cytokinin-inducible SYAC1 modulates secretory pathway for cell wall components in SYAC1 overexpressing plants. SYAC1 was further indicated as crucial for cell growth, as well as pathogen resistance in the Arabidopsis roots.

I think all data are clearly presented and support their conclusions. Even though the authors did not clarify in detail how SYAC1 regulates the secretory pathway, and how the auxin/cytokinin signalling converges on it, the manuscript describes a novel component where the two major plant hormones take part in modulating plant growth. This manuscript advances our knowledge and provides new perspectives on phytohormone-mediated regulation of plant development.

I suggest the authors to consider the following:

- Treating plants with high concentrations of two hormones (auxin and cytokinin) for 6h may result in artefacts, in addition to the actual hormone responses. To investigate whether SYAC1 is a direct target of the two phytohormone signalling pathways, the authors may check the expression level of SYAC1 in hormone signalling mutants, such as *ahk234* or inducible *bdl*-expressing plants, following the hormone treatment.

Response: Pointed also by the reviewer 2, we agree that prolonged treatment with auxin and cytokinin might activate additional downstream factors involved in control of the SYAC1 expression. To address this issue we performed a time course analysis of SYAC1 transcription using RT-qPCR. We found that already 30 min after application of auxin and cytokinins SYAC1 transcription in roots is significantly increased when compared to mock treated roots (Fig. S1b). Furthermore, we tested contribution of auxin and cytokinin receptors to synergistic regulation of SYAC1 transcription. We found that loss of either cytokinin, or auxin perception mediated through CRE1-12/AHK4,AHK3 and TIR1, AFB2, respectively significantly attenuates synergistic regulation of SYAC1 transcription by both hormones (Fig. S1c). We believe that these results further support role of both cytokinin and auxin pathways in control of SYAC1 expression.

- The current data convincingly suggest that SYAC1 is involved in plant cell growth, however it would be further supportive if the authors actually measured the cell size of SYAC1 overexpressing plants.

Response: We measured size of cortex cells in roots of 5 day-old wild-type and SYAC1ox seedlings and show, that enhanced activity of SYAC1 significantly reduces cell elongation when compared to control. The results were included in Fig. S2g and technical details in the Material and Method section.

- Description about biochemical properties of SYAC1 would be informative for readers if available. (The authors may describe what kind of domain(s) it has, how big the protein is, and predicted roles based on its properties, and so on in the first section of the results.)

Response: Our efforts to extract some information on structure of SYAC were not very instructive. The SYAC1 is small protein ~30KDa, based on hydrophobicity plot encompassing 4-5 transmembrane domains. Any other information using available software like RaptorX (<http://raptorx.uchicago.edu/>), PHYRE² (www.sbg.bio.ic.ac.uk/phyre2) or PHYRESTORM (www.sbg.bio.ic.ac.uk/~phyrestorm/phyrestorm) for structure prediction based on amino acid composition or topology were not sufficiently reliable due to absence of any related protein structure. To avoid any confusions and misinterpretation in context of further studies on this protein, we limit information provided in the manuscript to the size of SYAC1 protein.

- Figure 2 legend, 4th row: I think the wording "...from O to M phase..." should read "...from M to O phase...". Please check.

Response: We corrected the mistake.

- Figure 6 legend: This is indicated as Figure 7 currently, please correct. Also, the number of replicates is not specified, and info on the statistics is incomplete.

Response: We corrected and updated figure 6 legend.

Reviewer #4 (Remarks to the Author):

The manuscript entitled "Synergistic on auxin and cytokinin 1 integrates growth and pathogen defense" describes the identification and characterization of an auxin and cytokinin cross-talk component, SYNERGISTIC ON AUXIN AND CYTOKININ1 (SYAC1). Using transcriptomic analysis, the authors identified a gene specifically up-regulated in roots exposed simultaneously to cytokinin and auxin. Transcriptional SYAC1 activity/levels were low or not detected in untreated roots becoming active by the synergic action of auxin and cytokinins. In embryonic hypocotyl and cotyledons, the transcriptional reporter is endogenously active and remained active in seedlings cotyledons hypocotyl. Interestingly, the expression in etiolated plants is concentrated in short cells at the inner (concave) side of the apical hook, whereas no signal was detected in elongating cells at outer side of the hook. This pattern of expression led the authors to propose that SYAC1 might be regulating cell elongation.

Additional evidence supporting such theory was collected using loss of function mutants and over-expression of the protein.

The authors describe the protein SYAC1 to be located at the Golgi and TGN/endosomal organelles. Additionally, SYAC1 was shown to be interacting with ECHIDA/Yip complex by yeast two hybrid, co-immunoprecipitation and by BIFC (transient expression in protoplasts). The ECHIDA/Yip complex has been previously described to be involved in secretion specially in elongating cells. Therefore, the authors

propose that SYAC1 is acting as a negative regulator of the ECHIDA/Yip complex activity and therefore restricting cell elongation and growth. Finally, the authors propose that a protist infection that increases auxin-cytokinin levels in roots activates the expression of SYAC1 inhibiting secretion and facilitating infection.

General comments:

1.Regarding figure legends, they need to be more precise. In many cases the legends are confusing (see minor issues). Is also necessary to include in the legends: how many experiments are represented in each graph? The values are product of a single experiment but representative of the replicates? How many times the experiment was replicated? Is difficult to interpret the figures without this information that in many cases is not properly detailed in materials and methods either. Additionally, standard error can sometimes hide big differences between replicates, I recommend replacing the closed bars graphs by individual plot values, like the ones in the open boxes represented in figure 2E.

Response: We carefully re-checked all figure legends and added missing information to the figure legends or material and methods. Where possible, we also replaced graph charts as recommended.

2.Regarding material and methods, is very frustrating as a reader finding statements like this one: the experiment was performed “as previously described” “reference”. There are many protocols absent in the manuscript referenced this way, I strongly recommend the authors to include, at least, a brief description of the protocol.

Response: Because of relatively large spectrum of methods applied in the manuscript, we focused mostly on the techniques, which we adapted or introduced some modifications. We respect suggestion of the reviewer and expanded the material and methods accordingly.

Major

issues:

3.SYAC1A was shown to co-localize with RAB5-like protein markers (RHA1 and ARA7). ARA7 is present in late endosomes/PVC but it can be also found at the TGN (Singh, Manoj K., et al. Current Biology 24.12 (2014): 1383-1389.). Therefore, the authors should use additional PVC markers to provide supplementary evidence to confirm PVC localization. Otherwise, is easy to propose wrong conclusions based on miss interpretation of organelle localization.

Response: We appreciate this remark. We extended experiments and tested co-localization of the SYAC1 with VSR, a PVC marker (Miao et al., 2006), which supported conclusion about localization SYAC1 at PVC.

4.Several conclusions are based in indirect evidence weakening the manuscript:

4.1Evidence is presented suggesting that decrease alpha-amylase activity in the growing media of SYAC1 overexpressing protoplasts indicates a decrease in protein secretion. Nevertheless, it could also be a

problem in transformation efficiency, protein synthesis or protein folding that will produce a concomitant decrease alpha-amylase activity in the media. As this is a transient assay, additional controls should be included to quantify total protein and the amount of protein secreted to the media. Additionally, evidence supporting SYAC1 protein over-expression in protoplasts were the alpha-amylase activity is decreased should also be provided (for example western blot).

Response: We would like to stress, that α -Amylase is a standard assay, which was established to examine impact of proteins on secretory pathway (Leborgne-Castel et al., 1999; Pimpl et al., 2000). We agree with the reviewer concerns about limitation of this approach. Transformation efficiency can vary in each well, therefore we performed assay standardly including 2 biological replicates with two technical replicates per experiment. The α -Amylase activity in medium (secreted to the apoplast) and in cells (retained in the inner part of the cell) was measured in two to three time points and Secretion Index (SI) (ratio between α -Amylase in medium and in cells) as a normalized parameter, which allows to compare the different replicates and neutralizes the variability, was calculated. Protoplasts transformed without any plasmid were used for blank measurement. In total, the assays were performed 4 times, always with highly reproducible results. To address concerns of the reviewer about α -Amylase assay we present a raw data from one representative experiment demonstrating that a total expression of α -Amylase in protoplasts was not significantly affected by SYAC1 expression when compared to negative control. Hence, it is primarily ratio between α -Amylase activity in medium and cells that is altered in protoplast transiently expressing SYAC1 when compared to the negative control. Unfortunately, we could not monitor expression of the α -Amylase due to lack of specific antibodies. However, in all experiments, background activity of α -Amylase (blank) in non-transfected protoplasts was monitored, and the values were below these detected in protoplasts transformed with α -Amylase construct. Furthermore, we performed a Western blot as a proof of the SYAC1 protein expression in protoplasts (Fig. 5a).

20min		Blank	Control	Amy +SYAC1-HAox	Control	Amy-Spo +SYAC1-HAox	Control	Amy-HDEL +SYAC1-HAox
Biological replicate 1	Cells1a	0.097	1.706	1.983	1	0.829	2.551	2.55
	Medium1a	0.214	1.12	0.576	0.226	0.335	0.693	0.471
	Total α -amylase activity	0.311	2.826	2.559	1.226	1.164	3.244	3.021
	SI1a		0.657	0.290	0.226	0.404	0.272	0.185
	Cells 1b	0.097	1.867	2.244	1.039	0.783	2.751	3.1
	Medium 1b	0.214	1.134	0.607	0.196	0.323	0.752	0.435
	Total α -amylase activity	0.311	3.001	2.851	1.235	1.106	3.503	3.535
	SI1b		0.607	0.270	0.189	0.413	0.273	0.140
Biological replicate 2	Cells2a	0.092	1.693	1.947	0.985	0.82	2.503	2.718
	Medium2a	0.193	1.174	0.983	0.246	0.34	0.783	0.536
	Total α -	0.285	2.867	2.93	1.231	1.16	3.286	3.254

	amylase activity							
	SI2a		0.693	0.505	0.250	0.415	0.313	0.197
	Cells2b	0.092	1.874	2.398	1.05	0.842	2.969	2.999
	Medium2b	0.193	1.124	0.612	0.226	0.378	0.759	0.487
	Total α -amylase activity	0.285	2.998	3.01	1.276	1.22	3.728	3.486
	SI2b		0.600	0.255	0.215	0.449	0.256	0.162
Average	SI		0.639	0.330	0.220	0.420	0.278	0.171
	standard deviation		0.044	0.117	0.025	0.020	0.024	0.025
	standard error		0.022	0.059	0.013	0.010	0.012	0.013
	t test to control			0.00262		0.00002		0.00086

4.2 Additional information supporting the role of SYAC1 inhibiting of secretion was proposed by analyzing SYAC1ox plants. Imbibed seeds exhibit decreased mucilage secretion. Additionally, lower amounts of galacturonic acid can be extracted from cell wall in seedlings. All these evidences suggest a problem in cell wall components secretion, but it represents indirect evidence. The authors need to probe retention of cell wall components in the secretory system. Is important to know if the retention is at the level of the Golgi/TGN consistently with SYAC1 localization. Without this information is difficult to discard a problem in polysaccharides synthesis that also takes place at the Golgi apparatus. Additional evidence on protein or polysaccharides retention at the Golgi or TGN will considerably strengthen the hypothesis and mechanisms proposed.

Response: We agree with the reviewer that we propose the function of SYAC1 as a negative regulator of the secretory pathway based on set of indirect evidences.

With our collaborators we have consulted potential approaches. However, to provide direct evidences would require technically sophisticated experimentation that is currently challenge for many laboratories within the field. We admit that we are not able to exclude potential impact of SYAC1 on polysaccharide synthesis and metabolism, which would require in depth analysis of factors involved in this pathway. Taking into account fair criticism of the reviewer, we tuned-down interpretation related to the SYAC1 function in the secretory pathway and modified discussion to consider potential alternative mechanisms.

4.3The authors suggest that SYAC1 over-expression in protoplasts increased secretion of Amy-Spo indicating an impairment on vacuolar transport and, therefore a role of SYAC1 in vacuolar transport. This evidence is highly indirect, additional experiments in planta are necessary to support this claim. Again, evidence supporting SYAC1 protein over-expression in protoplasts should also be provided as this is a transient assay.

Response: The α -Amylase assay was used as one of the available approaches for testing impact of proteins on secretory pathway. We are aware that the assay serves just as a proxy. Whether and how

SYAC1 interferes with trafficking to vacuoles requires further investigation that was not primary goal of the project at this stage. However, SYAC1 localization at PVC, supported by co-localization with additional PVC specific marker VSR, which we included based on the reviewer recommendation, would be in line with this function.

The Western blot is provided (Fig. 5a) to support SYAC1 expression in protoplasts.

4.4 Transient over-expression of LAT52:SYAC1 in pollen tubes is suggested to decrease pectin deposition at the pollen tube apex. Nevertheless, supplementary figure 4D is not clear. The images seem to be coded in black and white making difficult to visualize the red staining. Essentially it looks like there is more ruthenium red signal in LAT52:SYAC1 pollen tube than in the control. Providing better quality images and including immunolocalizations using JIM5/JIM7 antibodies recognizing de-esterified or partially de-esterified pectin's is also advisable. It is also necessary to provide evidence supporting the over-expression of SAC1 protein in this transient system.

Response: We agree that analysis of SYAC1 effect in pollen tubes was not sufficiently elaborated. In the revised manuscript we present additional data including – improved imaging and quantification of the pectin accumulation pattern in pollen tubes in control and SYAC1 expressing pollen tubes, expression of the SYAC1-mCherry in pollen tube cells, and phenotype of pollen tubes that express SYAC1 when compared to control (Fig.S5).

While the analysis for de-esterified pectins is doubtless of interest, the experimental setup for transient SYAC1 overexpression in tobacco pollen tubes precludes the use of immunodetection approaches. This is because only a very small proportion of pollen tube cells in a population (<0.1%) will express the transgene upon particle bombardment. While the transgenic cells can easily be identified for in vivo imaging based on the fluorescence of the expressed fusion proteins, the approach is not feasible to use the transformed cells for antibody staining. Therefore, the suggested immuno-staining experiments with the JIM5 or JIM7 antibodies were not included in our revision.

4.5 PIN2 and PIN1 accumulation in root cells is not affected by SYAC1 over-expression (stable). How about during gravitropic response? This situation could be similar to the hypocotyl hook as the cells on the upper part are elongating in opposition the cells on the lower part of the bending root. Is important to determine SYAC1 function in planta and not only in transient experiments to support the conclusions of the paper.

Response: Based on suggestions of the reviewer 1 and 4, we tested expression of SYAC1 in response to photo- and gravistimulus in hypocotyls and roots, respectively. Both photo- and gravi-stimulation are established triggers of asymmetric auxin distribution at either shaded side of hypocotyls or at lower (gravistimulated) side of roots, respectively. In none of these experimental conditions, we detected SYAC1 expression. Since, SYAC1 expression is highly dependent on auxin and cytokinin levels, it seems that local increase of auxin activity in hypocotyl when not accompanied with simultaneous activation of the cytokinin pathway might not be sufficient to trigger SYAC1 expression. Similarly to hypocotyls, a local

increase of auxin activity in gravi-stimulated roots does not activate SYAC1 expression. Unlike hypocotyls, where contribution of the cytokinin pathway to phototropic response has not been studied so far, several recent works analyzed a role of cytokinin in root response to gravity. Interestingly, it has been shown (Waidmann et al., 2019) that cytokinin activity (monitored by cytokinin sensitive reporter TCS:GFP) is enhanced at upper side of lateral roots and counterbalances auxin-dependent gravitropic bending thus acting as anti-gravitropic signal. In addition, study of Pernisova et al., 2016 demonstrates that cytokinin interferes with PIN mediated auxin transport and supports antagonistic effect of cytokinin on auxin driven gravity response. Hence, during gravity response auxin and cytokinin activity might not be enhanced in the same tissues that would be pre-requisite for SYAC1 expression.

As concerns the function of the SYAC1 in planta, we agree that phenotypes of loss-of function *syac1* mutant alleles are relatively weak, however reproducible and tested using independent alleles. We would like to stress, that expression of the SYAC1 is spatio-temporally tightly controlled and limited to some specific developmental events. Therefore, the phenotype of *syac1* mutant might appear rather weak, if compared for example to mutant in ECHIDNA, which is constitutively expressed (Gendre et al., 2011). In support of that, we see that in processes where SYAC1 endogenous expression is stronger such as the apical hook and hypocotyls, *syac1* mutant alleles are significantly affected when compared to wild-type. Furthermore, in the revised manuscript we show that reduced sensitivity of both *syac1* mutant alleles to *Plasmodiophora* correlates with enhanced expression of SYAC1 in infected roots.

On the other hand, in all cases where syac1 mutant alleles exhibit phenotype alterations, such as apical hook development, hypocotyl expansion, or sensitivity to Plasmodiophora, enhanced SYAC1 activity results in “reversed” phenotype.

4.6 Auxin+cytokinins treatments increase SYAC1 promoter activity in the root. Nevertheless, the authors do not show a decrease on protein or polysaccharide secretion in such conditions or a concomitant decrease in cell elongation. It is also necessary to analyze if Auxin+cytokinins treatments on the loss of function SYAC1 mutants to show opposite phenotypes. I realize that the suggested experiments may represent an oversimplification of the hormone cross-talk pathway but is necessary to link the activity of the protein with the increased level of expression to be able to support the conclusions proposed in the manuscript. The authors should provide stronger evidence of such link even by providing a different set of experiments that the authors consider more suitable.

Response: We agree with a reviewer that analysis of auxin and cytokinin impact on the secretory pathway would provide connection that is more straightforward to SYAC1 activity. However, we believe that SYAC1 might be only part of the more complex regulation of subcellular trafficking and cell wall by both hormones (Cosgrove, 2016; Lehman et al., 2017; McCartney et al., 2003). As summarized in response above, currently the tools that enable reliable analysis of secretory pathway activity have technical limitations and we believe that proper and reliable analysis of auxin and cytokinin effects on the secretory pathway and cell wall composition are beyond a scope of this manuscript. However, we agree with the reviewer that auxin-cytokinin /SYAC1 module might feedback onto the activity of factors involved in other processes than the delivery of components to the cell wall such as biosynthesis of polysaccharides that comprise pectin, as well as their processing by methylesterification, acetylation or degradation. Therefore, in the revised manuscript we extended discussion to consider also these regulatory mechanisms.

4.7 Plasmodio phorabraceae infection increases auxin-cytokinin levels in roots. The manuscript shows that SYAC1 overexpressing plants are sensitive to the infection and, on the contrary, the mutants are more resistant than wild type plants. The authors propose that increased auxin-cytokinin levels in infected roots will increase the expression of SYAC1. Nevertheless, no evidence is provided indicating that SYAC1 mRNA and protein levels are indeed increased in wild type infected plants. This data will help supporting the proposed hypothesis and will increase the relevance of the infection assay.

Response: In the revised manuscript we demonstrate that in roots 20 and 28 days after inoculation the SYAC1:GUS expression is enhanced, whereas no SYAC1:GUS activity could be detected in non-infected plants. As a positive control root of 34 day-old plants were treated with auxin and cytokinin that led to upregulation of SYAC1 expression as well (Fig. S7a,b).

Minor issues:

1. GUS staining in roots highlights the expression pattern of the transcriptional reporter in the QC and elongation/differentiation zone. Nevertheless, even do they have the promoter fuse to a nuclear localization signal tag with GFP, the authors fail to show a more precise cellular pattern of expression that will certainly be more informative. Having that information will provide the readers a more precise idea on how restricted in the expression at the QC/columella, elongating cells vs dividing cells among others.

Response: In the revised version the magnified image demonstrating expression of pSYAC1:GUS as well as pSYAC1:GFP at root tips (Fig.S1d)

2. qPCR analysis showing the remaining SYAC1 mRNA levels in the insertional mutants is only provided for syac1-1 and syac1-3 lines. What about syac1-2, syac1-4 and the CRISP line syac1-5?

Response: Actually, although we indicated syac1-1, syac1-2 and syac1-4 alleles in the scheme, in our experiments we standardly used syac1-3 and syac1-5 allele generated by CRISPR/Cas approach. As concerns CRISPR syac1-5 allele we extended the revised manuscript for sequencing data confirming mutation and homozygosity of the syac1-5 (Fig S2a). Lack of CRISPR/Cas cassette was confirmed by loss of resistance to BASTA, and the relevant information was added in Material and Method paragraph.

3. The over-expression lines (SYAC1-HAox, HA-SYAC1ox, SYAC1-GFPox and GFP-SYAC1ox) were shown to have higher mRNA level than wild type plants. However, the concomitant increase in protein levels is not shown. Western blots are recommended.

Response: Confirmation of SYAC1 expression by Western blot analysis was included (Fig. S2c,d).

4. The authors propose that in syac1-3 and syac1-5 mutant backgrounds the protein BSYAC1 might be compensating for the SYAC1 loss of function. Is either the BSYAC1 mRNA or protein levels altered in the syac1-3 and syac1-5 mutant backgrounds?

Response: We tested expression of BSYAC1 in syac1-3 and syac1-5 alleles by qRT-PCR. Although its expression in the syac1 mutant does not significantly alter, we cannot exclude possible redundant activity. Furthermore, as also mentioned in reply to the reviewer 2, SYAC1 is member of small family of 7 genes located as a cluster at chromosome 1, which might contribute to redundancy.

5. The co-immunoprecipitation analysis in Figure 4C shows in line 1 the control (that I assume is the untransformed protoplast, needs to be stated in the legend), the ECG-HA input, GFP (alone?) and, ECH-HA and protoplast transfected with both SYAC1-GFP and ECH-HA. The Co-IP was performed using anti GFP antibody. However, background signal observed in protoplast extracts expressing only ECH-HA (and no SYAC1-GFP). Is the signal due to poor washing or unspecific background? It needs to be address in the text. Due to that unspecific signal and the transient nature of the experiment is necessary to include the ECH-HA protein level in the input when both proteins, SYAC1-GFP+ECH-HA, are expressed in the protoplast. In this way, is possible to unequivocally distinguish co-IP ECH-HA together with SYAC1-GFP from unspecific binding due to stronger expression of ECH-HA by itself. Is also remarkable that ECH-HA detected after the immunoprecipitation presents a higher molecular weight

compared to the input signal. Is my understanding from materials and methods that this is an SDS-PAGE so, why there is a molecular weight shift? The text needs to address this.

Response: We apologize if the legend/description for Fig. 4c was not clear enough. As the reviewer supposed, control means non-transfected protoplast and GFP means protoplast transfected with a 35S:GFP construct. We have corrected this in the description. On the left side of Figure 4c immunoprecipitation was performed with an anti-GFP antibody to show protein-protein interaction between ECHIDNA (ECH-HA) and SYAC1 (SYAC1-GFP). As the reviewer pointed out there is an unspecific band in the control and the ECH-HA lanes, although the unspecific band is separable from the specific. The reviewer also noticed that the GFP antibody did pull down some of the ECH-HA (a ghost band), though the difference in the intensity between the GFP-antibody-immunoprecipitated ECH-HA and the ECH-HA + SYAC1-GFP bands is clearly visible (the difference is several magnitude). We would like to mention here that the sample, which was used as ECH-HA input was the 1/6th of the immunoprecipitated samples. ECH-HA always gave us high levels of protein expression in the protoplast system. Additionally, we would also like to indicate that we used the same protoplast assay which was used for detecting protein-protein interaction between ECHIDNA and YIP4a in the publication of Gendre et al., 2013, where they also detected ghost bands and a shift in their SDS-PAGE blots (please see the image below).

Gendre et al. *Plant Cell* 2013;25:2633-2646

The reason for the presence of the ghost bands could be the high transient protein expression level. Regarding the shifts in protein migration, there can be several explanations e.g., immunoprecipitated (i.e. “clean”) proteins may run slightly differently than in a crude extract; or ECH may be post-translationally modified (see the smear above the main signal) and ECH interactors may not interact with each form.

Together with the presented yeast two hybrid results, the BiFC experiments, the above discussed Co-IP experiments and the cited Gendre article we are positive that the protein interactions between SYAC1 and ECH or YIP4a are solid.

6. Supplemental figure 4D needs a clearer legend, is difficult for this reviewer to understand what the authors are intending to show.

Response: *The figure legend was extended to provide more informative description of the presented graph.*

“(D) Multivariate curve resolution-alternating least squares MCR-ALS resolved spectral profiles of chemical components used for k-means clustering in Figure 5C. None of the resolved components are “pure” chemical compounds, but each spectrum is dominated by bands from different classes of compounds. The colors correspond to the colors of Figure 5C. Blue component spectrum: dominated by mainly carbohydrate related bands in the region of 900-1150 cm^{-1} ; Red component spectrum: a mixture with clear contribution from protein bands (amide II at 1550 cm^{-1} and amide I at 1650 cm^{-1}); Yellow component spectrum: mixture of carbohydrate and protein related bands (potentially glycoproteins); Purple component spectrum: mixture with high contribution from extractives (bands at 1420 and 1630 cm^{-1})”

References:

Cosgrove, D.J. (2016). Catalysts of plant cell wall loosening. *F1000Research* 5.

Daher, F.B., and Braybrook, S.A. (2015). How to let go: pectin and plant cell adhesion. *Front. Plant Sci.* 6.

Ding, Z., Galván-Ampudia, C.S., Demarsy, E., Łangowski, Ł., Kleine-Vehn, J., Fan, Y., Morita, M.T., Tasaka, M., Fankhauser, C., Offringa, R., et al. (2011). Light-mediated polarization of the PIN3 auxin transporter for the phototropic response in Arabidopsis. *Nat. Cell Biol.* 13, 447–452.

Frühholz, S., and Pimpl, P. (2017). Analysis of Nanobody–Epitope Interactions in Living Cells via Quantitative Protein Transport Assays. In *Plant Protein Secretion*, (Humana Press, New York, NY), pp. 171–182.

Gendre, D., Oh, J., Boutte, Y., Best, J.G., Samuels, L., Nilsson, R., Uemura, T., Marchant, A., Bennett, M.J., Grebe, M., et al. (2011). Conserved Arabidopsis ECHIDNA protein mediates trans-Golgi-network trafficking and cell elongation. *Proc. Natl. Acad. Sci.* *108*, 8048–8053.

Gendre, D., McFarlane, H.E., Johnson, E., Mouille, G., Sjodin, A., Oh, J., Levesque-Tremblay, G., Watanabe, Y., Samuels, L., and Bhalerao, R.P. (2013). Trans-Golgi Network Localized ECHIDNA/Ypt Interacting Protein Complex Is Required for the Secretion of Cell Wall Polysaccharides in Arabidopsis. *Plant Cell* *25*, 2633–2646.

González-Carranza, Z.H., Elliott, K.A., and Roberts, J.A. (2007). Expression of polygalacturonases and evidence to support their role during cell separation processes in Arabidopsis thaliana. *J. Exp. Bot.* *58*, 3719–3730.

Laplaze, L., Parizot, B., Baker, A., Ricaud, L., Martinière, A., Auguy, F., Franche, C., Nussaume, L., Bogusz, D., and Haseloff, J. (2005). GAL4-GFP enhancer trap lines for genetic manipulation of lateral root development in Arabidopsis thaliana. *J. Exp. Bot.* *56*, 2433–2442.

Lashbrook, C.C., and Cai, S. (2008). Cell wall remodeling in Arabidopsis stamen abscission zones. *Plant Signal. Behav.* *3*, 733–736.

Leborgne-Castel, N., Jelitto-Van Dooren, E.P., Crofts, A.J., and Denecke, J. (1999). Overexpression of BiP in tobacco alleviates endoplasmic reticulum stress. *Plant Cell* *11*, 459–470.

Lehman, T.A., Smertenko, A., and Sanguinet, K.A. (2017). Auxin, microtubules, and vesicle trafficking: conspirators behind the cell wall. *J. Exp. Bot.* *68*, 3321–3329.

McCartney, L., Steele-King, C.G., Jordan, E., and Knox, J.P. (2003). Cell wall pectic (1→4)-beta-d-galactan marks the acceleration of cell elongation in the Arabidopsis seedling root meristem. *Plant J. Cell Mol. Biol.* *33*, 447–454.

Miao, Y., Yan, P.K., Kim, H., Hwang, I., and Jiang, L. (2006). Localization of Green Fluorescent Protein Fusions with the Seven Arabidopsis Vacuolar Sorting Receptors to Prevacuolar Compartments in Tobacco BY-2 Cells. *Plant Physiol.* *142*, 945–962.

Ogawa, M., Kay, P., Wilson, S., and Swain, S.M. (2009). ARABIDOPSIS DEHISCENCE ZONE POLYGALACTURONASE1 (ADPG1), ADPG2, and QUARTET2 Are Polygalacturonases Required for Cell Separation during Reproductive Development in Arabidopsis. *Plant Cell* *21*, 216–233.

Pernisova, M., Prat, T., Grones, P., Harustiakova, D., Matonohova, M., Spichal, L., Nodzynski, T., Friml, J., and Hejatko, J. (2016). Cytokinins influence root gravitropism via differential regulation of auxin transporter expression and localization in Arabidopsis. *New Phytol.* *212*, 497–509.

Phyo, P., Wang, T., Kiemle, S.N., O'Neill, H., Pingali, S.V., Hong, M., and Cosgrove, D.J. (2017). Gradients in Wall Mechanics and Polysaccharides along Growing Inflorescence Stems. *Plant Physiol.* *175*, 1593–1607.

Pimpl, P., Movafeghi, A., Coughlan, S., Denecke, J., Hillmer, S., and Robinson, D.G. (2000). In situ localization and in vitro induction of plant COPI-coated vesicles. *Plant Cell Online* *12*, 2219–2235.

Ristova, D., Carré, C., Pervent, M., Medici, A., Kim, G.J., Scalia, D., Ruffel, S., Birnbaum, K.D., Lacombe, B., Busch, W., et al. (2016). Combinatorial interaction network of transcriptomic and phenotypic responses to nitrogen and hormones in the *Arabidopsis thaliana* root. *Sci. Signal.* *9*, rs13–rs13.

Siemens, J., Nagel, M., Ludwig-Müller, J., and Sacristán, M.D. (2002). The Interaction of *Plasmodiophora brassicae* and *Arabidopsis thaliana*: Parameters for Disease Quantification and Screening of Mutant Lines. *J. Phytopathol.* *150*, 592–605.

Waidmann, S., Ruiz Rosquete, M., Schöller, M., Sarkel, E., Lindner, H., LaRue, T., Petřík, I., Dünser, K., Martopawiro, S., Sasidharan, R., et al. (2019). Cytokinin functions as an asymmetric and anti-gravitropic signal in lateral roots. *Nat. Commun.* *10*, 1–14.

REVIEWERS' COMMENTS:

Reviewer #2 (Remarks to the Author):

I thank the authors for the revisions in the manuscript and for clarifying many points. Some points of concern still stand after the revision. See summary below:

1. I still think the authors need to introduce the experiment that gave rise to the identification of SYAC1 in the Results section. This was requested in the first revision, and I don't think it has been done adequately. The authors should add the information about which experiment was done, and how it was done, in the Results section. To say that "we performed genome wide transcriptome profiling of roots of *Arabidopsis thaliana* exposed to auxin, cytokinin and both hormones simultaneously" doesn't clarify which experiment was done (RNAseq?) or how it was done. Similarly, to say that "After a 3 hour of treatment with either auxin or cytokinin increased SYAC1 expression (2.47 ± 0.20 and 1.53 ± 0.19 , respectively)" is not enough, the concentration and age of plants needs to be known. The authors have provided good information about the experiment in the "Response to Reviewers", just add this to Results section of the manuscript. If the manuscript is accepted, this needs to be revised.

2. I appreciate the fact that the authors have now performed a qRT-PCR for SYAC1 to determine how fast it is induced by auxin and cytokinin. This is important information that suggests that SYAC1 is a primary target of both cytokinin and auxin. If the manuscript is accepted, I advise that these data are moved to the main figures (figure 1), rather than being presented as a supplemental figure 1.

3. A major point of concern that I mentioned in my original review is the fact *syac1* mutants have very subtle phenotypes. Therefore, many of the roles of SYAC1 are being deduced based either on very subtle phenotypes of *syac1* mutants (apical hook length fig2a, hypocotyl length fig 2b, fig 2e). Or, in the absence of any phenotypes on the mutants, the roles of SYAC1 are being deduced from the phenotypes of SYAC1 overexpression lines (fig 2c, 2d, 2f, 5a, 5b, 5c and 5d). The concern is that overexpressing lines can have gain-of-function phenotypes that are unrelated to that gene's natural function. Therefore, it is not clear what SYAC1 does. The authors argue that genetic redundancy may be the cause of the subtle phenotypes of *syac1* alleles (the authors mention BSYAC1 as a possible redundant gene compensating for SYAC1) and have now explained in this revision that double mutants, needed to establish whether redundancy is in fact happening, are hard to be obtained given the proximity of the SYAC1 and BSYAC1 genes in chromosome 1. A possible option would be to use CRISPR/Cas in the *syac1* background to mutate BSYAC1 and see whether a stronger phenotype is seen. This reviewer understands that such approach would require more work and would indeed make things harder, and therefore I do not think that such experiment is required. However, lack of a loss-of-function line with a clear phenotype does weaken any claims of a known genetic function for SYAC1, as it is just hard to determine a function of a gene without conclusive loss-of-function genetic analysis.

4. With regards to the results of figure 5: In my original review I asked about the statistics used in that figure, and that has not been explained in the figure itself, which is necessary. What are the letters a, b, c referring to, to which statistical test?

5. Another major point is that I still don't agree with the author's interpretation about SYAC1 mediating growth-defense tradeoffs. All we know given the data presented is that it's a gene regulated by growth hormones, that when knocked-out has subtle increased growth phenotypes (hypocotyl length and root length, figures 2b, 2c) and increased resistance phenotypes (figure 5b,c). Why is this enough to say that SYAC1 "integrates growth and defense"? Growth are phenotypes that have been known for decades to be associated (known as the growth-defense tradeoff). There are many known genes with similar phenotypes of altered growth and defense that are known, and only a few of them have been shown to really integrate growth and defense.

The strong claim by the authors that SYAC1 integrates growth and defense, especially in the title of the manuscript, is just not warranted. The following claim in the discussion "Hence, the tight regulation of SYAC1, which normally limits its expression in the root, might be an essential part of the immune system that has evolved in cruciferous plants to balance pathways controlling growth with mechanisms protecting roots from infection" is a stretch to the observations described in the manuscript.

Reviewer #3 (Remarks to the Author):

The authors have addressed my concerns, providing new data that further supports their conclusions. I have no additional comments at this stage.

Reviewer #4 (Remarks to the Author):

The authors have satisfactorily responded all of my questions and made the necessary changes to the manuscript.

Minor revisions:

The authors added co-localization analysis using several endomembrane markers from the Wave collection. They should detail the name of the marker besides the line denomination. For example, for the marker wave25R they should include the name of the protein, RabD1. This is more informative for the reader and would make the text easier to follow.

Please confirm that the revised version has been submitted in Microsoft Word format and that track changes have been used to make final edits

Response: We confirm that the revised manuscript is submitted in Microsoft Word format using track changes.

* Please provide a point-by-point response to these specific editorial requests in your cover letter. Please provide a point-by-point response to remaining reviewer comments as a separate file.

* We feel that appropriate text edits would be required to address the remaining concerns of reviewer #2. While we expect to be able to assess the revision editorially, I cannot exclude the possibility that we may need to seek further reviewer advice at this stage.

Response: In the revised manuscript, all remaining concerns of the reviewer 2 were addressed. See response to reviewer 2 below.

We would be satisfied if you were to simply conclude that SYAC1 positively regulates growth but negatively regulates soil pathogen resistance rather than suggesting a regulatory or integrative role in controlling tradeoffs. We would ask that a new title is chosen - we allow up to 15 words and ask that you avoid punctuation. We would also ask that corresponding edits are made throughout the text.

*Response: Throughout the text we introduced changes to avoid causal link between SYAC1 role in regulation of growth and pathogen sensitivity. Accordingly, the title of the manuscript was modified to: **“Synergistic on auxin and cytokinin 1 positively regulates growth and attenuates soil pathogen resistance”***

Regarding the concern about inferring protein function based on overexpression, we would ask that you carefully specify when this is the case and there may not be a corresponding phenotype in the mutant. For example, rather than definitively stating “This protein fine tunes the growth of organs by controlling the cell wall composition” we would suggest “This protein fine tunes the growth of organs and when overexpressed can impact cell wall composition”.

Response: The adequate modifications were introduced throughout the revised manuscript.

* The manuscript text should be arranged in the following order

Title

Author list and affiliations

Abstract

Introduction

Results

Discussion

Methods

Data Availability

References
Competing Interests
Acknowledgments
Author Contributions
Figure Legends

Response: The manuscript text was re-arranged accordingly.

Please provide headings and subheadings in bold font (non-italic unless referring to genes, species names etc.). Text after each heading should be placed on a separate line.

Response: The headings, subheadings and text were formatted accordingly.

* Please use all upper case and italic letters for gene names, lower case italics for mutant names and upper case, non-italic for protein names throughout.

Response: The manuscript was thoroughly checked to apply corresponding format.

* Please describe the current findings in the abstract in present rather than past tense.

Response: The abstract was amended accordingly.

* Please use the text “these authors contributed equally” to indicate equal contribution.

Response: The text was amended accordingly.

* We allow a maximum of 60 characters per subheading (including spaces) in the Results and Methods section. Please edit as required throughout. Please avoid punctuation such as commas in subheadings.

Response: The headings, subheadings and text were formatted accordingly.

* We do not allow subheadings in the Discussion section. Please remove.

Response: Subheadings in the discussion were removed.

* Wherever +/- values are used in the text, please make sure they are defined and the number of replicates is specified e.g. increased SYAC1 expression (2.47 fold increase \pm 0.20; n = 3, \pm indicates standard error)

Response: The manuscript was thoroughly checked to apply corresponding format.

* When citing Supplementary Figures or Tables, please use the format Supplementary Figure X, Supplementary Table X. Note the use of the word Supplementary and the lack of the S prefix.

Response: The text was amended accordingly.

* Please rename the ‘Materials and Methods’ section as ‘Methods’

Response: The text was amended accordingly.

* We do not allow further subdivision of subsections of the Methods section. In the section “Identification of SYAC1 by transcriptome profiling” please remove the “FACS:” and “Transcriptome profiling:” subheadings.

Response: Further subdivision in the Methods section was removed.

* Where software packages are used (limma, ebayes etc) please state the version number if known.

Response: The manuscript was thoroughly checked to specify software packages.

* Please ensure that sufficient information is provided such that the experiments could be reasonably reproduced by a qualified laboratory without the need to refer to external literature. In particular please ensure that you do not rely on 'as previously described' or similar statements. Please note that there are no word limits for the Methods section.

Response: The Methods section was thoroughly checked and extended to provide full protocols.

* Please include the antibody dilution used in the Co-IP and Western Blot sections of the Methods. Please also ensure that the source of all antibodies is indicated and catalog numbers are included for commercial antibodies).

Response: The source, catalog numbers and dilution of antibodies were added when missing.

* Please ensure that all primer sequences are included either in the Methods section or as a Supplementary Table cited from the Methods section. This includes any primers used for molecular cloning.

Response: The methods section was re-checked to cite all primers used for experimental work.

* Please replace the Van Leene et al., 2014 citation with a superscript number (current line 858)

Response: The citation was replaced by corresponding number (98)

* We do not publish separate accession number sections. Please include TAIR identifiers at first mention of the relevant gene in the Results or Methods section.

Response: TAIR identifiers were removed from Methods section and introduced throughout the text.

* Please deposit the raw RNA-seq data into the NCBI Sequence Read Archive. This must be made available before publication.

Response: Transcriptome profile data associated with this study has been deposited in the NCBI Sequence Read Archive under accession number <https://www.ncbi.nlm.nih.gov/geo/query/acc.cgi?acc=GSE146778>.

* Please deposit the mass spectrometry data in the PRIDE repository

<https://www.ebi.ac.uk/pride/archive/>

Response: The mass spectrometry proteomics data have been deposited to the ProteomeXchange Consortium via the PRIDE partner repository with the dataset identifier PXD018159 and 10.6019/PXD018159. Reviewer account details:

Username: reviewer92817@ebi.ac.uk Password: KMcws3KQ.

* Please include a Data Availability statement immediately after the Methods section. This should include an accession code for the RNA-seq and proteomic data, a reference to a Source Data file and confirmation that any other underlying data is available on request.

Response: Data Availability statement including accession code for the transcriptome and proteomic data, a reference to a Source Data file and confirmation that any other underlying data is available on request were included

E.g. The RNA-seq data associated with this study has been deposited in the NCBI Sequence Read Archive under accession number SRX????

[<https://www.ncbi.nlm.nih.gov/sra/SRX????>]. Proteomic data has been deposited to the PRIDE repository under accession code ???

Response: Both the transcriptome and proteomic data were deposited to corresponding archives and are publically available. The mass spectrometry proteomics data have been deposited to the ProteomeXchange Consortium via the PRIDE partner repository with the dataset identifier PXD018159 and 10.6019/PXD018159. Reviewer account details:

Username: reviewer92817@ebi.ac.uk Password: KMcws3KQ.

[<https://www.ebi.ac.uk/pride/archive/projects/PXD????>]. Data underlying Fig 1a, 2b and 3c are available as a separate Source Data file. Any other data supporting the findings of this study are available from the corresponding author upon request,

Response: We provide source data for all experiments presented in the manuscript, including results of experimental repetitions.

* Please ensure that a Source Data file is uploaded using the file type ‘Supplementary Dataset’ on our manuscript submission portal. Instructions on how to format a Source Data file appear below.

Response: The Source Data file was uploaded accordingly.

* Please provide an author contributions statement. The author contributions section should refer to all authors using initials. If any two authors have the same initials please use the full last name to distinguish between them.

Response: An author contribution statement is included in the revised manuscript.

* Please provide a Competing Interests statement. If there are no competing interests please state "The authors declare no competing interests."

Response: A Competing Interests is included in the revised manuscript.

* Please rename external dataset S1 and S2 as Supplementary Data 1 and Supplementary Data 2. These should be uploaded as individual Excel workbooks (i.e. two individual .xlsx files rather than two sheets within the same workbook. Please provide a brief title/legend for each Supplementary Data file in your cover letter when you resubmit.

Response: Corresponding changes were introduced.

* Please provide the main figures as individual files (without embedded legends). Figure legends should remain part of the main manuscript file.

Response: Corresponding changes were introduced.

* Please confirm that every panel of every figure is cited in the text in the correct order. If a figure needs to be cited out of order, please instead use the text ‘see below’

Response: The text was re-checked for figure citation in the correct order. To maintain the correct order of figures cited in the text Supplementary figure 4 and 5 were rearranged.

* Please double check that where error bars, symbols, colours, p-values and scale bars are used in figures that they are defined in the corresponding figure legend. Please indicate the number of experimental replicates and the name of any statistical test used where statistical significance is indicated. Symbol definitions in figure legends should be written out in words (blue circles, red dashed line, etc.) as symbols will not appear properly in the HTML text.

Response: The figure legends were amended accordingly.

* Please indicate the position of a MW marker in Figure 4c.

Response: The position of a MW marker in Figure 4c was added.

* Please confirm that each display item is no larger than a pdf page size (260x179 mm).

Response: Size of figures was re-checked.

* Please ensure that where data is represented as bar charts corresponding dot plots are overlaid on the figure (please see the following editorial for the rationale behind this request and an example <https://www.nature.com/articles/s41551-017-0079>).

Response: All graphs, with exception Fig. 2b-d; and Supplementary Figure 4k, which demonstrate relative change in % of control were transformed to the requested format.

* All Supplementary Figures and Supplementary Tables should be provided as a single PDF file. Each Supplementary Figure or Supplementary Table legend should appear on the same page as the corresponding figure/table and not in the main manuscript file.

Response: Supplementary material provided according to recommendation.

* Please provide updated Editorial Policy and Reporting Summary checklists and ensure they are uploaded with the revised manuscript. These files are available via the links below.

Reporting summary:

Editorial policy checklist:

<https://www.nature.com/documents/nr-editorial-policy-checklist.pdf>

Please note that these forms are in dynamic ‘smart pdf’ format and must therefore be downloaded and completed in Adobe Reader.

Please note that the Reporting Summary will appear online as a Supplementary file when the manuscript is published.

* We encourage increased transparency in peer review by publishing the reviewer comments and author rebuttal letters of our research articles, if the authors agree. Such peer review material is made available as a supplementary peer review file. **Please state in the cover letter ‘I wish to participate in transparent peer review’ if you want to opt in, or ‘I do not wish to participate in transparent peer review’ if you don’t.** Failure to state your preference will result in delays in accepting your paper for publication.

Please note: we allow redactions to authors’ rebuttal and reviewer comments in the interest of confidentiality. If you are concerned about the release of confidential data, please let us know specifically what information you would like to have removed. Please note that we cannot

incorporate redactions for any other reasons. Reviewer names will be published in the peer review files if the reviewer signed the comments to authors, or if reviewers explicitly agree to release their name. For more information, please refer to our FAQ page at:

<https://www.nature.com/documents/ncomms-transparent-peer-review.pdf>

* Please check whether your manuscript or Supplementary Information contain third-party images, such as figures from the literature, stock photos, clip art or commercial satellite and map data. We strongly discourage the use or adaptation of previously published images, but if this is unavoidable, please request the necessary rights documentation to re-use such material from the relevant copyright holders and return this to us when you submit your revised manuscript.

* Your paper will be accompanied by a two-sentence editor's summary, of between 250-300 characters, when it is published on our homepage. Could you please approve the draft summary below or provide us with a suitably edited version.

Response: Cytokinin and auxin are two major hormonal regulators of plant growth. Here the authors identify *SYAC1*, a gene that is synergistically activated by the two hormones being applied together, and show that it is required for normal growth while negatively impacting pathogen resistance.

REVIEWERS' COMMENTS:

Reviewer #2 (Remarks to the Author):

I thank the authors for the revisions in the manuscript and for clarifying many points. Some points of concern still stand after the revision. See summary below:

1. I still think the authors need to introduce the experiment that gave rise to the identification of *SYAC1* in the Results section. This was requested in the first revision, and I don't think it has been done adequately. The authors should add the information about which experiment was done, and how it was done, in the Results section. To say that "we performed genome wide transcriptome profiling of roots of *Arabidopsis thaliana* exposed to auxin, cytokinin and both hormones simultaneously" doesn't clarify which experiment was done (RNAseq?) or how it was done. Similarly, to say that "After a 3 hour of treatment with either auxin or cytokinin increased *SYAC1* expression (2.47 ± 0.20 and 1.53 ± 0.19 , respectively)" is not enough, the concentration and age of plants needs to be known. The authors have provided good information about the experiment in the "Response to Reviewers", just add this to Results section of the manuscript. If the manuscript is accepted, this needs to be revised.

Response: Thank you the reviewer for the useful comment. Although we provided detailed information in methods section, we agree with the suggestion and added additional information also in the result section.

2. I appreciate the fact that the authors have now performed a qRT-PCR for *SYAC1* to determine how fast it is induced by auxin and cytokinin. This is important information that suggests that *SYAC1* is a primary target of both cytokinin and auxin.

If the manuscript is accepted, I advise that these data are moved to the main figures (figure 1), rather than being presented as a supplemental figure 1.

Response: Following the suggestion of the reviewer the data presented at Figure S1b were transferred to Figure 1.

3. A major point of concern that I mentioned in my original review is the fact *syac1* mutants have very subtle phenotypes. Therefore, many of the roles of SYAC1 are being deduced based either on very subtle phenotypes of *syac1* mutants (apical hook length fig 2a, hypocotyl length fig 2b, fig 2e). Or, in the absence of any phenotypes on the mutants, the roles of SYAC1 are being deduced from the phenotypes of SYAC1 overexpression lines (fig 2c, 2d, 2f, 5a, 5b, 5c and 5d). The concern is that overexpressing lines can have gain-of-function phenotypes that are unrelated to that gene's natural function. Therefore, it is not clear what SYAC1 does. The authors argue that genetic redundancy may be the cause of the subtle phenotypes of *syac1* alleles (the authors mention BSYAC1 as a possible redundant gene compensating for SYAC1) and have now explained in this revision that double mutants, needed to establish whether redundancy is in fact happening, are hard to be obtained given the proximity of the SYAC1 and BSYAC1 genes in chromosome 1. A possible option would be to use CRISPR/Cas in the *syac1* background to mutate BSYAC1 and see whether a stronger phenotype is seen. This reviewer understands that such approach would require more work and would indeed make things harder, and therefore I do not think that such experiment is required. However, lack of a loss-of-function line with a clear phenotype does weaken any claims of a known genetic function for SYAC1, as it is just hard to determine a function of a gene without conclusive loss-of-function genetic analysis.

Response: We agree with the reviewer criticism. We continue in further investigation and functional analyses of other family members. This includes efforts to generate CRISPR line to knock-out function of multiple homologous genes.

4. With regards to the results of figure 5: In my original review I asked about the statistics used in that figure, and that has not been explained in the figure itself, which is necessary. What are the letters a, b, c referring to, to which statistical test?

Response: We apologize for inaccuracy. The figure legend was amended accordingly and the letters refer to statistical analyzes using the Kruskal-Wallis-test.

5. Another major point is that I still don't agree with the author's interpretation about SYAC1 mediating growth-defense tradeoffs. All we know given the data presented is that it's a gene regulated by growth hormones, that when knocked-out has subtle increased growth phenotypes (hypocotyl length and root length, figures 2b, 2c) and increased resistance phenotypes (figure 5b,c). Why is this enough to say that SYAC1 "integrates growth and defense"? Growth are phenotypes that have been known for decades to be associated (known as the growth-defense tradeoff). There are many known genes with similar phenotypes of altered growth and defense that are known, and only a few of them have been shown to really integrate growth and defense. The strong claim by the authors that SYAC1 integrates growth and defense, especially in the title of the manuscript, is just not warranted. The following claim in the discussion "Hence, the tight regulation of SYAC1, which normally limits its expression in the root, might be an essential part of the immune system that has evolved in cruciferous plants to balance pathways controlling growth with mechanisms protecting roots from infection" is a stretch to the observations described

in the manuscript.

Response: Throughout the text we introduced changes to avoid causal link between SYAC1 role in regulation of growth and pathogen sensitivity. Accordingly, the title of the manuscript was modified to: “Synergistic on auxin and cytokinin 1 positively regulates growth and attenuates soil pathogen resistance“

Reviewer #3 (Remarks to the Author):

The authors have addressed my concerns, providing new data that further supports their conclusions. I have no additional comments at this stage.

Reviewer #4 (Remarks to the Author):

The authors have satisfactorily responded all of my questions and made the necessary changes to the manuscript.

Minor revisions:

The authors added co-localization analysis using several endomembrane markers from the Wave collection. They should detail the name of the marker besides the line denomination. For example, for the marker wave25R they should include the name of the protein, RabD1. This is more informative for the reader and would make the text easier to follow.

Response: We amended the figure according to the reviewer suggestion.